# First Order Methods with Markovian Noise: from Acceleration to Variational Inequalities

**Aleksandr Beznosikov**
Innopolis University, Skoltech, MIPT, Yandex

**Sergey Samsonov**
HSE University

**Marina Sheshukova**
HSE University

**Alexander Gasnikov**
MIPT, Skoltech, IITP RAS

**Alexey Naumov**
HSE University

**Eric Moulines**
Ecole polytechnique

## Abstract

This paper delves into stochastic optimization problems that involve Markovian noise. We present a unified approach for the theoretical analysis of first-order gradient methods for stochastic optimization and variational inequalities. Our approach covers scenarios for both non-convex and strongly convex minimization problems. To achieve an optimal (linear) dependence on the mixing time of the underlying noise sequence, we use the randomized batching scheme, which is based on the multilevel Monte Carlo method. Moreover, our technique allows us to eliminate the limiting assumptions of previous research on Markov noise, such as the need for a bounded domain and uniformly bounded stochastic gradients. Our extension to variational inequalities under Markovian noise is original. Additionally, we provide lower bounds that match the oracle complexity of our method in the case of strongly convex optimization problems.

## 1 Introduction

Stochastic gradient methods are an essential ingredient for solving various optimization problems, with a wide range of applications in various fields such as machine learning [36, 37], empirical risk minimization problems [96], and reinforcement learning [93, 85, 69]. Various stochastic gradient descent methods (SGD) and their accelerated versions [75, 31] have been extensively studied under different statistical frameworks [17, 97]. The standard assumption for stochastic optimization algorithms is to consider independent and identically distributed noise variables. However, the growing usage of stochastic optimization methods in reinforcement learning [10, 87, 25] and distributed optimization [63, 18, 65] has led to increased interest in problems with Markovian noise. Despite this, existing theoretical works that consider Markov noise have significant limitations, and their analysis often results in suboptimal finite-time error bounds.

Our research aims to fill the gap in the existing literature on the first-order Markovian setting. By focusing on uniformly geometrically ergodic Markov chains, we obtain finite-time complexity bounds for achieving $\varepsilon$-accurate solutions that scale linearly with the mixing time of the underlying Markov chain. Our approach is based on careful applications of randomized batch size schemes and provides a unified view on both non-convex and strongly convex minimization problems, as well as variational inequalities.

**Our contributions.** Our main contributions are the following:

◇ **Accelerated SGD.** We provide the first analysis of SGD, including the Nesterov accelerated SGD method, with Markov noise without the assumption of bounded domain and uniformly bounded stochastic gradient estimates. Our results are summarised in Table 1 and Section 2.1 and cover both strongly convex and non-convex scenarios. Our findings for non-convex minimization problems complement the results obtained in [21].

37th Conference on Neural Information Processing Systems (NeurIPS 2023).

◇ **Lower bounds.** In Section 2.2 we give the lower bounds showing that the presence of mixing time in the upper complexity bounds is not an artefact of the proof. This is consistent with the results reported in [71].

◇ **Extensions.** In Section 2.4 we provide, as far as we know, the first analysis for variational inequalities with general stochastic Markov oracle, arbitrary optimization set, and arbitrary composite term. Our finite-time performance analysis provides complexity bounds in terms of oracle calls that scale linearly with the mixing time of the underlying chain, which is an improvement over the bounds obtained in [99] for the Markov setting.

**Related works.** Next, we briefly summarize the related works.

◇ **Stochastic gradient methods.** Numerous research papers have reported significant improvements achieved by accelerated methods for stochastic optimization with stochastic gradient oracles involving independent and identically distributed (i.i.d.) noise. These methods have been extensively studied in theory [44, 14, 16, 58, 61, 26, 30, 97, 94, 3, 39, 102] and have shown practical success [55, 91]. The finite-time analysis of first-order methods in i.i.d. noise settings has been extensively studied by many authors, as discussed in [59] and references therein. In Table 1 we include only some important results because i.i.d. setting is not in the interest of this paper.

While the literature on i.i.d. noise is extensive, existing research on the first-order Markovian setting is relatively sparse. In this study, we focus on Markov chains that are uniformly geometrically ergodic, and we refer the reader to Section 2 for detailed definitions. We note that the complexity bounds which scale linearly with the mixing time of the underlying general Markov chain are currently available only for general convex and non-convex minimization problems. Namely, [23] has investigated a version of the ergodic mirror descent algorithm that yields optimal convergence rates for Lipschitz, general convex and non-convex problems. Recently, [21] proposed a random batch size algorithm that adapts to the mixing time of the underlying chain for non-convex optimization with a compact domain. In particular, [21, Theorem 4.3] yields optimal complexity rates in terms of the number of oracle calls required for non-convex problems, which is consistent with the results obtained in [23]. Unlike previous studies, this method is insensitive to the mixing time of the noise sequence.

For the general case of Markovian noise the finite-time analysis of non-accelerated SGD-type algorithms was carried out in [90] and [19]. However, [90] heavily relies on the bounded domain assumption and uniformly bounded stochastic gradient oracles, while its bound in [90, Theorem 5] has a suboptimal dependence on the mixing time of the underlying chain, see Table 1. Additionally, [90] does not cover the strongly convex setting. On the other hand, [19] covers both non-convex and strongly convex settings, but the bounds of [19, Theorem 1] has terms that are *exponential* in the mixing time, and a careful examination reveals suboptimal dependence on the initial condition for strongly convex problems when SGD is applied.

In the study of Nesterov-accelerated SGD with Markovian noise, the authors of [20] considered the use of a batch size of 1 and achieved a rate of forgetting the initial condition that matches that of the i.i.d. noise setting. However, their result is suboptimal in terms of the variance terms in both non-convex and strongly convex settings, as detailed in Table 1. We emphasize that the case of unbounded gradient oracles with Markov noise is not treated in contrast to the i.i.d. setup [97, 62].

The above papers deal with general Markovian noise optimization. But there are also results that deal with Markovian stochasticity with a finite state space. Here we can highlight the work [28], where the author gives quite extensive results and achieves linear scaling by mixing time in the non-convex as well as strongly convex cases. Recently, numerous papers have appeared dealing with the special scenario of distributed optimization [89]. [99] investigates the generalization and stability of Markov SGD with special attention to the excess variance guarantees. We note that first, these algorithms only need to deal with a very special case of Markov gradients, and second, the corresponding dependence on the mixing time of the Markov chain is again quadratic. At the same time, there exist particular results, e.g. [71], which provide a lower bound for the particular finite sum problems in the Markovian setting.

◇ **Variational inequalities.** Variational inequalities [29] have been an active area of research in applied mathematics for more than half a century [78, 41, 86]. VI cover important special cases, e.g., minimization over a convex domain, saddle point or min-max and fixed point problems. computational game theory [29], robust [7] and nonsmooth [73, 72] optimization, supervised [51, 4] and unsupervised [103, 5] learning, image denoising [27, 11]. In the last 5 years, variational inequalities and their special cases have attracted much interest in the machine learning community

due to new connections to reinforcement learning [79, 50], adversarial training [64], and GANs [15, 33, 66, 12, 60, 82].

Variational inequalities (VI) and saddle point problems have their own well-established theory and methods. Unlike minimization problems, solving variational inequalities doesn't rely on (accelerated) gradient descent. Instead, the extragradient method [57], various modified versions [72, 42], or similar techniques [95] are recommended as the basic and theoretically optimal methods. While deterministic methods have long been used for solving variational inequalities, stochastic methods have gained importance only in the last 15 years, following pioneering works by [49, 52]. We summarise the results on methods for stochastic variational inequalities with the Lipschitz operator and smooth stochastic saddle point problems in Table 2. The number of papers dealing with stochastic VIs and saddle point problems is small compared to those dealing with stochastic optimization, we include in Table 2 papers with the i.i.d. noise (which we do not do for stochastic optimization). The only competing work dealing with Markovian noise in saddle point problems consider the finite sum problem and thus the finite Markov chain [99], therefore we do not include it in Table 2. Moreover, the results from [99] has much worse oracle complexity guarantees $\mathcal{O}(\tau^2/\varepsilon^2)$ in terms of $\tau$. There are more papers dealing with stochastic finite-sum variational inequalities or saddle point problems, but in the i.i.d. setting [12, 80, 104, 2, 8]. We also do not consider in Table 2 because of the difference in the stochastic oracle structure. It is important to note that, unlike most previous works, we consider the most general formulation of VI itself for an arbitrary optimization set and composite term.

**Notations and definitions.** Let $(\mathsf{Z}, \mathsf{d}_\mathsf{Z})$ be a complete separable metric space endowed with its Borel $\sigma$-field $\mathcal{Z}$. Let $(\mathsf{Z}^\mathbb{N}, \mathcal{Z}^{\otimes\mathbb{N}})$ be the corresponding canonical process. Consider the Markov kernel Q defined on $\mathsf{Z} \times \mathcal{Z}$, and denote by $\mathbb{P}_\xi$ and $\mathbb{E}_\xi$ the corresponding probability distribution and the expected value with initial distribution $\xi$. Without loss of generality, we assume that $(Z_k)_{k\in\mathbb{N}}$ is the corresponding canonical process. By construction, for any $A \in \mathcal{Z}$, it holds that $\mathbb{P}_\xi(Z_k \in A | Z_{k-1}) = \mathrm{Q}(Z_{k-1}, A)$, $\mathbb{P}_\xi$-a.s. If $\xi = \delta_z$, $z \in \mathsf{Z}$, we write $\mathbb{P}_z$ and $\mathbb{E}_z$ instead of $\mathbb{P}_{\delta_z}$ and $\mathbb{E}_{\delta_z}$, respectively. For $x^1, \dots, x^k$ being the iterates of any stochastic first-order method, we denote $\mathcal{F}_k = \sigma(x^j, j \le k)$ and write $\mathbb{E}_k$ as an alias for $\mathbb{E}[\cdot|\mathcal{F}_k]$. We also write $\mathbb{N}^* := \mathbb{N} \setminus \{0\}$. For the sequences $(a_n)_{n\in\mathbb{N}}$ and $(b_n)_{n\in\mathbb{N}}$ we write $a_n \lesssim b_n$ if there exists a constant $c$ such that that $a_n \le c b_n$ for all $n \in \mathbb{N}$.

Table 1: This table summarizes our results on first-order method with Markovian noise. The columns of the table indicate whether the authors consider optimization over bounded domain, potentially unbounded gradients, and whether or not they assume additional restrictions on the Markovian noise (finite state space or reversibility). For ease of comparison we provide the respective results on SGD and ASGD (accelerated SGD) in the i.i.d. setting.

| | Method | Domain | Unbounded Gradient noise | General MC | Acceleration | Oracle complexity (Smooth and non-convex) | Oracle complexity (Smooth and strongly convex) |
|---|---|---|---|---|---|---|---|
| i.i.d. | SGD [84,70,42] | ✓ | ✗ | N/A | ✗ | $\tilde{\mathcal{O}}\left(L(f(x^0)-f(x^*))\left[\frac{1}{\varepsilon^2}+\frac{\sigma^2}{\varepsilon^4}\right]\right)$ | $\tilde{\mathcal{O}}\left(\frac{L}{\mu}\log\frac{\|x^0-x^*\|^2}{\varepsilon}+\frac{\sigma^2}{\mu^2\varepsilon}\right)$ |
| | ASGD [97,13] [1] | ✓ | ✓ | N/A | ✓ | $\tilde{\mathcal{O}}\left(L(f(x^0)-f(x^*))\left[\frac{1+\delta^2}{\varepsilon^2}+\frac{\sigma^2}{\varepsilon^4}\right]\right)$ | $\tilde{\mathcal{O}}\left((1+\delta^2)\sqrt{\frac{L}{\mu}}\log\frac{\|x^0-x^*\|^2}{\varepsilon}+\frac{\sigma^2}{\mu^2\varepsilon}\right)$ |
| Markovian | EMD [23] [2] | ✗ | ✗ | ✓ | ✗ | $\tilde{\mathcal{O}}\left(\frac{\tau G^2 D^2}{\varepsilon^4}\right)$ | ✗ |
| | MC SGD [90] [3] | ✓ | ✗ | ✗ | ✗ | $\tilde{\mathcal{O}}\left(h(G,L)\left(\frac{\tau}{\varepsilon^2}\right)^{1/(1-q)}\right)$ | ✗ |
| | MC SGD [19] [4] | ✓ | ✓ | ✓ | ✗ | $\tilde{\mathcal{O}}\left(\frac{\tau L^2(1+\|x^*\|^2+\|x^0-x^*\|^2)}{\varepsilon^4}\right)$ | $\tilde{\mathcal{O}}\left(e^{\tau(L/\mu)^2}\left[h(\frac{L}{\mu})\log\frac{\|x^0-x^*\|^2}{\varepsilon}+\frac{\tau^2 L^2(1+\|x^*\|^2)}{\mu^2\varepsilon}\right]\right)$ |
| | ASGD [20] [5] | ✗ | ✗ | ✗ | ✓ | $\tilde{\mathcal{O}}\left(\frac{1}{\varepsilon^4}\left[B^2+G^6(L^2\tau^2+1)\right]\right)$ | $\tilde{\mathcal{O}}\left(\sqrt{\frac{L}{\mu}}\frac{\|x^0-x^*\|^2}{\varepsilon^{1/2}}+\frac{\tau^2(G^2+\mu GD+\mu LD^2)}{\mu^2\varepsilon}\right)$ |
| | MAG [21] [6] | ✓ | ✗ | ✓ | ✗ | $\tilde{\mathcal{O}}\left(\frac{\tau(G+L+B)^2G^2}{\varepsilon^4}\right)$ | |
| | MC SGD [28] (Sec. 5.1) [7] | ✓ | ✗ | ✗ | ✗ | $\mathcal{O}\left(\frac{\tau(L(f(x^0)-f(x^*))+\sigma^2)}{\varepsilon^2}+\frac{\tau(L(f(x^0)-f(x^*))+\sigma^2)\sigma^2}{\varepsilon^4}\right)$ | $\mathcal{O}\left(\frac{\tau L}{\mu}\log\frac{(f(x^0)-f(x^*))/\mu+\sigma^2/(\mu L)}{\varepsilon}+\frac{\tau\sigma^2}{\mu^2\varepsilon}\right)$ |
| | MC SGD [28] (Sec. 5.2) [8] | ✓ | ✗ | ✗ | ✗ | ✗ | $O\left(\frac{L}{\mu}\log\frac{\|x^0-x^*\|^2}{\varepsilon}+\frac{L\tau\sigma_*^2}{\mu^3\varepsilon}\right)$ |
| | RASGD (ours) | ✓ | ✓ | ✓ | ✓ | $\tilde{\mathcal{O}}\left(\tau L(f(x^0)-f(x^*))\left[\frac{1+\delta^2}{\varepsilon^2}+\frac{\sigma^2}{\varepsilon^4}\right]\right)$ | $\tilde{\mathcal{O}}\left(\tau\left[(1+\delta^2)\sqrt{\frac{L}{\mu}}\log\frac{\|x^0-x^*\|^2}{\varepsilon}+\frac{\sigma^2}{\mu^2\varepsilon}\right]\right)$ |

*notation:* $\mu$ and $L$ are as in A.1 and A.2; $G = \sup_{x,z} \|\nabla F(x,z)\|$. Note that $G \ge L$ and $G^2 \ge \sigma^2$ under A.4. We also set $B = \sup_x |f(x)|$; $x^0$ - starting point, $x^*$ - solution, $\mathcal{D}$ - optimisation domain; $D = \sup_{x\in\mathcal{D}} \|x-x^*\|$, $\sigma$ and $\delta$ - stochasticity parameters (see A.4); $\sigma_*$ - stochasticity parameter in $x^*$; $\tau$ - mixing time of the chain (see A.3), $\varepsilon$ - accuracy of the solution, measured as $\mathbb{E}[\|\nabla f(x)\|^2] \lesssim \varepsilon^2$ for non-convex problems and $\mathbb{E}[\|x-x^*\|^2] \lesssim \varepsilon$ for the strongly convex ones. Functions $h(L/\mu)$ and $h(G,L)$ stands for an implicit dependence of the respective parameters.

[1] gives results with stepsize as a parameter, we choose it the close way as in our Corollary 1. [2] covers more general noise setting, then just Markovian. [3] for general state-space Markov noise the analysis of [90] requires reversibility. Parameter $q \in (1/2; 1)$ refers to the step size $\sim 1/k^q$. [4] The fluctuation terms in [19, Theorem 1,3] contain hidden dependence on the initial error and $\|x^*\|$ in the fluctuation terms, making the result comparison complicated. They also contain hidden factors, which are exponential in $C = \tau/\log 4$ in the notations of our paper. Moreover, the analysis of [19] requires that $F(x,z)$ is Lipschitz w.r.t. $x$ for any $z \in \mathsf{Z}$. [5] considers Markovian noise with finite state space and a specifically decreasing step size. Moreover, in the proof of [20, Theorem 3] (equations (64) − (66)) the authors lost the factor $C^2$, with $C = \tau/\log 4$. The result in the table accounts for this lost factor. [6] considers the adaptive tuning of batch size, which is oblivious to $\tau$. [7] considers Markov noise with finite state space and additionally assumes that all stochastic realization $F(\cdot, Z)$ are $L$-smooth. [8] considers Markovian noise with finite state space, $\sigma_*$ bounds noise only in $x^*$, but additionally assumes that all stochastic realization $F(\cdot, Z)$ are $L$-smooth and $\mu$-strongly convex.

## 2 Main results

**Assumptions.** In this paper we study the minimization problem

$$\min_{x\in\mathbb{R}^d} f(x) := \mathbb{E}_{Z\sim\pi}[F(x,Z)], \qquad (1)$$

where the access to the function $f$ and its gradient is available only through the (unbiased) noisy oracle $F(x,Z)$ and $\nabla F(x,Z)$, respectively. In the following presentation we impose at least one of the following regularity constraint on the underlying function $f$ itself:

Table 2: This table summarizes the findings on methods for solving stochastic (strongly) monotone variational inequalities with a Lipschitz operator and (un)bounded stochasticity. The columns of the table indicate whether the authors consider variational inequalities or only certain saddle point problems, the arbitrariness of the sets, and the use of additional composite terms. The columns on stochasticity provide information on the assumptions made with respect to the stochastic operator, such as bounded variance and the Markovian noise setting. Note that with the exception of our work, all other studies assume the independent noise.

| | Method | Statement | | | Stochasticity | | Oracle complexity |
|---|---|---|---|---|---|---|---|
| | | VI? | Any set? | Composite? | Unbounded? | Markovian? | |
| Strongly monotone | SPEG [33,42] | ✓ | ✓ | ✗ | ✗ | ✗ | $\tilde{\mathcal{O}}\left(\frac{L}{\mu}\log\frac{\|x^0-x^*\|^2}{\varepsilon}+\frac{\sigma^2}{\mu^2\varepsilon}\right)$ [1] |
| | SEG [53] | ✓ | ✓ | ✗ | ✗ | ✗ | $\tilde{\mathcal{O}}\left(\frac{\|x^0-x^*\|^2}{\varepsilon}+\frac{B^2+\sigma^2+(B+\sigma)(1+LD)}{\sigma^2\varepsilon}\right)$ |
| | SS-SEG [68,38] | ✓ | ✓ | ✓ | ✓ | ✗ | $\tilde{\mathcal{O}}\left(\frac{L}{\mu}\log\frac{\|x^0-x^*\|^2}{\varepsilon}+\frac{\sigma_*^2}{\mu^2\varepsilon}\right)$ |
| | SEG [9] | ✗ | ✓ | ✗ | ✗ | ✗ | $\tilde{\mathcal{O}}\left(\frac{L}{\mu}\log\frac{\|x^0-x^*\|^2}{\varepsilon}+\frac{\sigma^2}{\mu^2\varepsilon}\right)$ |
| | DSEG [43] | ✓ | ✗ | ✗ | ✓ | ✗ | $\mathcal{O}\left(\left[\frac{L^2\sigma^2}{\mu^4\varepsilon}\right]^3\right)$ [2] |
| | UEG [38] | ✓ | ✗ | ✗ | ✓ | ✗ | $\mathcal{O}\left(\left(\frac{L+\Delta}{\mu}+\frac{\Delta^2}{\mu^2}\right)\log\frac{\|x^0-x^*\|^2}{\varepsilon}+\frac{\sigma^2}{\mu^2\varepsilon}\right)$ |
| | SGDA [8] | ✓ | ✓ | ✓ | ✗ | ✗ | $\mathcal{O}\left(\frac{L^2}{\mu^2}\log\frac{\|x^0-x^*\|^2}{\varepsilon}+\frac{\sigma^2}{\mu^2\varepsilon}\right)$ [3] |
| | REG (ours) | ✓ | ✓ | ✓ | ✓ | ✓ | $\tilde{\mathcal{O}}\left(\tau\cdot\left[\left(\frac{L+\Delta}{\mu}+\frac{\Delta^2}{\mu^2}\right)\log\frac{\|x^0-x^*\|^2}{\varepsilon}+\frac{\sigma^2}{\mu^2\varepsilon}\right]\right)$ |
| Monotone | SMP [52] | ✓ | ✓ | ✗ | ✗ | ✗ | $\mathcal{O}\left(\frac{LD^2}{\varepsilon}+\frac{\sigma^2\Delta^2}{\varepsilon^2}\right)$ |
| | VR-SEG [45] | ✓ | ✓ | ✗ | ✓ | ✗ | $\mathcal{O}\left(\frac{(\sigma+\Delta)^8D^4}{\varepsilon^2}+\frac{D^4}{\varepsilon^2}\right)$ |
| | IPM [46] | ✓ | ✓ | ✓ | ✓ | ✗ | $\mathcal{O}\left(\tilde{\mathcal{O}}\left(\frac{L^4D^4}{\varepsilon^2}+\frac{\sigma_*^2D^4}{\varepsilon^2}\right)\right)$ |
| | SS-SEG [68] | ✓ | ✓ | ✓ | ✓ | ✗ | $\tilde{\mathcal{O}}\left(\frac{L^2D^4}{\varepsilon}+\frac{\sigma_*^4}{L^2\varepsilon^2}\right)$ |
| | SEG [9] | ✗ | ✓ | ✗ | ✗ | ✗ | $\mathcal{O}\left(\frac{LD^2}{\varepsilon}+\frac{\sigma^2\Delta^2}{\varepsilon^2}\right)$ |
| | REG (ours) | ✓ | ✓ | ✓ | ✓ | ✓ | $\tilde{\mathcal{O}}\left(\tau\cdot\left[\frac{LD^2}{\varepsilon}+\frac{\sigma^2D^2}{\varepsilon^2}+\frac{\Delta^2D^4}{\varepsilon^2}\right]\right)$ |

*notation:* $\mu$ = constant of strong monotonicity of operator $F$, $L$ = Lipschitz constant of $F$, $B$ = uniform bound of $F$, $D$ = uniform bound of iterations $x^k$, $x^0$ = starting point, $x^*$ = solution, $\Delta$ and $\sigma$ = stochasticity parameters (see A 7, [52,33,42,53,9,8] take $\Delta=0$), $\sigma_*$ = stochasticity parameter in $x^*$ (see [68]), $\tau$ = mixing time of the chain (see A 3), $\varepsilon$ = accuracy of the solution. [1] give results with stepsize as a parameter, we choose it according to Section 3 from [88]. [2] consider A 7, but do not provide explicit rates if $\Delta\neq0$ (see also [38, Table 1]). [3] consider the cocoercive case, for which in general $\ell=L^2/\mu$.

**A 1.** *The function $f$ is $L$-smooth on $\mathbb{R}^d$ with $L>0$, i.e., it is differentiable and there is a constant $L>0$ such that the following inequality holds for all $x,y\in\mathbb{R}^d$:*

$$\|\nabla f(x)-\nabla f(y)\|\leq L\|x-y\|.$$

**A 2.** *The function $f$ is $\mu$-strongly convex on $\mathbb{R}^d$, i.e., it is continuously differentiable and there is a constant $\mu>0$ such that the following inequality holds for all $x,y\in\mathbb{R}^d$:*

$$(\mu/2)\|x-y\|^2\leq f(x)-f(y)-\langle\nabla f(y),x-y\rangle. \qquad (2)$$

Next we specify our assumptions on the sequence of noise variables $\{Z_i\}_{i=0}^\infty$. We consider here the general setting of $\{Z_i\}_{i=0}^\infty$ being a time-homogeneous Markov chain. Such problems naturally arise in stochastic optimization. In the empirical risk minimization problems it naturally appears in the context of non-random minibatch choice. Indeed, a random choice of a batch number may lose to a non-random one, see [67, 56]. A wide range of problems dealing with Markovian noise is spawned by the reinforcement learning methods. The usual MDP setting falls naturally inside this paradigm, moreover, the analysis of non-tabular RL problems requires to deal with the general state-space Markov noise. Here the potential range of applications include the policy evaluation methods, such as the temporal difference methods [92], and policy optimization algorithms, such as policy gradient family, e.g. the celebrated REINFORCE algorithm [100].

We denote by Q the Markov kernel corresponding to the sequence $\{Z_i\}_{i=0}^\infty$ and impose the following assumption on the mixing properties of Q:

**A 3.** *$\{Z_i\}_{i=0}^\infty$ is a stationary Markov chain on $(\mathsf{Z},\mathcal{Z})$ with Markov kernel Q and unique invariant distribution $\pi$. Moreover, Q is uniformly geometrically ergodic with mixing time $\tau\in\mathbb{N}$, i.e., for every $k\in\mathbb{N}$,*

$$\Delta(\mathrm{Q}^k)=\sup_{z,z'\in\mathsf{Z}}(1/2)\|\mathrm{Q}^k(z,\cdot)-\mathrm{Q}^k(z',\cdot)\|_{\mathsf{TV}}\leq(1/4)^{\lfloor k/\tau\rfloor}. \qquad (3)$$

The assumption A 3 is classical in the literature on optimization methods with Markovian noise and has been considered in particular in recent works [90, 21, 20]. In particular, this assumption covers finite state-space Markov chains with irreducible and aperiodic transition matrix considered in

[28]. Yet our definition of the mixing time $\tau$ is more classical in the probability literature [81], and is slightly different from the one considered e.g. in [28, 65]. Next we specify our assumptions on stochastic gradient:

**A 4.** *For all $x \in \mathbb{R}^d$ it holds that $\mathbb{E}_\pi[\nabla F(x, Z)] = \nabla f(x)$. Moreover, for all $z \in \mathsf{Z}$ and $x \in \mathbb{R}^d$ it holds that*

$$\|\nabla F(x, z) - \nabla f(x)\|^2 \le \sigma^2 + \delta^2 \|\nabla f(x)\|^2. \tag{4}$$

The assumption A 4 resembles the strong growth condition [97], which is classical for the over-parametrized learning setup [97, 98]. The main difference is that A 4 concerns the almost sure bound in (4), which is unavoidable when dealing with uniformly geometrically ergodic Markovian noise A 3. Note that it is possible that the quantity $\delta^2$ in (4) is not instance-independent and scales with the ratio $L/\mu$ from A 1-A 2 in the particular problems. With the assumptions A 3 and A 4 we can prove the result on the mean squred error of the stochastic gradient estimate computed over batch size $n$ under arbitrary initial distribution. This result is summarized below in Lemma 1:

**Lemma 1.** *Assume A 3 and A 4. Then, for any $n \ge 1$ and $x \in \mathbb{R}^d$, it holds that*

$$\mathbb{E}_\pi[\|n^{-1} \textstyle\sum_{i=1}^n \nabla F(x, Z_i) - \nabla f(x)\|^2] \le \tfrac{8\tau}{n} \left(\sigma^2 + \delta^2 \|\nabla f(x)\|^2\right). \tag{5}$$

*Moreover, for any initial distribution $\xi$ on $(\mathsf{Z}, \mathcal{Z})$, that*

$$\mathbb{E}_\xi[\|n^{-1} \textstyle\sum_{i=1}^n \nabla F(x, Z_i) - \nabla f(x)\|^2] \le \tfrac{C_1 \tau}{n} \left(\sigma^2 + \delta^2 \|\nabla f(x)\|^2\right), \tag{6}$$

*where $C_1 = 16(1 + \frac{1}{\ln^2 4})$.*

*Proof.* We first prove (5). Note that due to [81, Proposition 3.4] the Markov kernel Q under A 3 admits a positive pseudospectral gap $\gamma_{ps} > 0$ such that $1/\gamma_{ps} \le 2\tau$. Thus, applying the statement of [81, Theorem 3.2], we get under A 4 that

$$\mathbb{E}_\pi[\|n^{-1} \textstyle\sum_{i=1}^n \nabla F(x, Z_i) - \nabla f(x)\|^2] \le \tfrac{4\mathbb{E}_\pi[\|\nabla F(x, Z_1) - \nabla f(x)\|^2]}{n\gamma_{ps}} \le \tfrac{8\tau}{n} \left(\sigma^2 + \delta^2 \|\nabla f(x)\|^2\right).$$

To prove the second part we use the maximal exact coupling construction and follow, e.g., [24, Theorem 1]. The complete proof is given in Appendix B.1. □

The proof of Lemma 1 simplifies the arguments in [21, Lemma 4] and allows us to obtain tighter values for the constants when dealing with the randomized batch size. Note that it is especially important to have the result for MSE under arbitrary initial distribution $\xi$, since in the proofs of our main results we will inevitably deal with the conditional expectations w.r.t. the previous iterate. We provide more details on the bias and variance of the Markov SGD gradients in the next section.

## 2.1 Accelerated method

We begin with a version of Nesterov accelerated SGD with randomized batch size, described in Algorithm 1. Due to the unboundedness of the stochastic gradient variance (see A 4), using of the classical Nesterov accelerated method [76, Section 2.2.] does not give the desired result, it is necessary to introduce an additional momentum [74, 97]. We use our own version, but partially similar to [74, 97]. The main feature of Algorithm 1 is that the number of samples used during the $k$-th gradient computation scales as $2^{J_k}$, where $J_k$ comes from a truncated geometric distribution. The truncation parameter needs to be adopted (see Theorem 1) in order to control the computational complexity of the algorithm.

Randomized batch size allows for efficient *bias* reduction in the stochastic gradient estimates and can be seen as a particular case of the so called multilevel MCMC [35, 34]. In the optimization context this approach was successfully used by [21] for the non-convex problems. Indeed, this bias naturally appears under the Markovian stochastic gradients oracles. It is easy to see that, with the counter $T^k$ defined in Line 9, we have

$$\mathbb{E}_k[\nabla F(x^k, Z_{T^k+i})] \ne \nabla f(x^k).$$

Below we show how the bias of the gradient estimate scales with the truncation parameter $M$. The statement of Lemma 2 yields that the gradient estimates $g_k$ introduced above have the bias, which decreases *quadratically* with $M$.

**Lemma 2.** *Assume A 3 and A 4. Then for the gradient estimates $g^k$ from Algorithm 1 it holds that $\mathbb{E}_k[g^k] = \mathbb{E}_k[g^k_{\lfloor \log_2 M \rfloor}]$. Moreover,*

$$\mathbb{E}_k[\|\nabla f(x^k) - g^k\|^2] \lesssim \left(\tau B^{-1} \log_2 M + \tau^2 B^{-2}\right)(\sigma^2 + \delta^2 \|\nabla f(x^k)\|^2),$$
$$\|\nabla f(x^k) - \mathbb{E}_k[g^k]\|^2 \lesssim \tau^2 M^{-2} B^{-2}(\sigma^2 + \delta^2 \|\nabla f(x^k)\|^2).$$

---

**Algorithm 1** `Randomized Accelerated GD`

---

1: **Parameters:** stepsize $\gamma > 0$, momentums $\theta, \eta, \beta, p$, number of iterations $N$, batchsize limit $M$
2: **Initialization:** choose $x^0 = x_f^0$
3: **for** $k = 0, 1, 2, \ldots, N-1$ **do**
4: $\quad x_g^k = \theta x_f^k + (1-\theta)x^k$
5: $\quad$ Sample $J_k \sim \text{Geom}(1/2)$
6: $\quad g^k = g_0^k + \begin{cases} 2^{J_k}\left(g_{J_k}^k - g_{J_k-1}^k\right), & \text{if } 2^{J_k} \leq M \\ 0, & \text{otherwise} \end{cases} \quad \text{with} \quad g_j^k = 2^{-j}B^{-1}\sum_{i=1}^{2^j B} \nabla f(x_g^k, Z_{T^k+i})$
7: $\quad x_f^{k+1} = x_g^k - p\gamma g^k$
8: $\quad x^{k+1} = \eta x_f^{k+1} + (p-\eta)x_f^k + (1-p)(1-\beta)x^k + (1-p)\beta x_g^k$
9: $\quad T^{k+1} = T^k + 2^{J_k}B$
10: **end for**

---

The proof and the statement with explicit constants are given in Appendix B.2. Note that the Lemma 2 is a natural counterpart of the deterministic bound Lemma 1. Moreover, it gives the idea of the trade-off between the parameters $B$ and $M$. Namely, the expected number of oracle calls to compute $g_k$ is $\mathcal{O}(B\log_2(M))$ with the bias scaling as $M^{-2}$. Thus the increase of $M$ drastically reduced the bias with only a logarithmic payment in variance. At the same time, gradient variance scales as $(\tau/B)^2$, but the increase of $B$ is much more expensive for the computational cost of the whole procedure. Taking into account the considerations above, we can prove the following result:

**Theorem 1.** *Assume A 1 – A 4. Let problem (1) be solved by Algorithm 1. Then for any $b \in \mathbb{N}^*$, $\gamma \in (0; \frac{3}{4L}]$, and $\beta, \theta, \eta, p, M, B$ satisfying*

$$p \simeq (1 + (1+\gamma L)[\delta^2\tau b^{-1} + \delta^2\tau^2 b^{-2}])^{-1}, \quad \beta \simeq \sqrt{p^2\mu\gamma}, \quad \eta \simeq \sqrt{\frac{1}{\mu\gamma}},$$

$$\theta \simeq \frac{p\eta^{-1}-1}{\beta p\eta^{-1}-1}, \quad M \simeq \max\{2; \sqrt{p^{-1}(1+p/\beta)}\}, \quad B = \lceil b\log_2 M \rceil,$$

*it holds that*

$$\mathbb{E}\left[\|x^N - x^*\|^2 + \frac{6}{\mu}(f(x_f^N) - f(x^*))\right] \lesssim \exp\left(-N\sqrt{\frac{p^2\mu\gamma}{3}}\right)\left[\|x^0 - x^*\|^2 + \frac{6}{\mu}(f(x^0) - f(x^*))\right]$$
$$+ \frac{p\sqrt{\gamma}}{\mu^{3/2}}\left(\sigma^2\tau b^{-1} + \sigma^2\tau^2 b^{-2}\right). \quad (7)$$

The proof is provided in Appendix B.3. The result of Theorem 1 can be rewritten as an upper complexity bound under an appropriate choice of the remaining free parameter $b$:

**Corollary 1.** *Under the conditions of Theorem 1, choosing $b = \tau$ and $\gamma$ as*

$$\gamma \simeq \min\left\{\frac{1}{L}; \frac{1}{p^2\mu N^2}\ln\left(\max\left\{2; \frac{\mu^2 N[\|x^0-x^*\|^2+6\mu^{-1}(f(x_f^0)-f(x^*))]}{\sigma^2}\right\}\right)\right\},$$

*in order to achieve $\varepsilon$-approximate solution (in terms of $\mathbb{E}[\|x - x^*\|^2] \lesssim \varepsilon$) it takes*

$$\tilde{\mathcal{O}}\left(\tau\left[(1+\delta^2)\sqrt{\frac{L}{\mu}}\log\frac{1}{\varepsilon} + \frac{\sigma^2}{\mu^2\varepsilon}\right]\right) \quad \text{oracle calls}. \quad (8)$$

The results of Corollary 1 are obtained with fixed parameters of the method. In Corollary 1 these parameters are selected a bit artificially, e.g., the stepsize $\gamma$ depends on the iteration horizon $N$. In Appendix B.4 we show how one can similar results, but with a decreasing stepsize.

**Comparison.** Running the procedure above requires to know the mixing time $\tau$. Estimating the mixing time from a single trajectory of the running Markov chain is known to be computationally hard problem, see e.g. [101] and references therein. At the same time, methods, which share the same (optimal) linear scaling of the sample complexity w.r.t. the mixing time also share the same drawback as our method. In particular, it holds true for the EMD algorithm [23], SGD-DD algorithm [71], and usual SGD with Markovian data [28]. At the same time, in the non-convex scanario the paper [21] is truly oblivious to mixing time, allowing to obtain sample complexity rates for non–convex problems, which are homogeneous w.r.t. $\tau$ with AdaGrad-type learning rate. An interesting direction for the future work to suggest a procedure that would allow to generalize the results of [21] to accelerated SGD setting.

It is possible that the sample complexity bound (8) is worse than the respective bounds for non-accelerated SGD with Markov data, provided that $\delta^2$ grows quickly with $L/\mu$. At the same time, this drawback is shared by the classical results on learning under the strong growth condition, see e.g. [97]. As it is shown in [62], the respective rates can be worse than the ones obtained by usual SGD even under the i.i.d. noise setting, see Appendix F.3 in [62]. Making the analysis of accelerated SGD 'backward compatible' w.r.t. the rates of usual SGD requires to perform analysis in terms of additional problem-specific quantities, see [47, 62].

The closest equivalent of the result Corollary 1 is given by [20, Theorem 3]. However, the corresponding bound of [20, Theorem 3] is incomplete, since the factor $\tau^2$ is lost in the proof (see equations $(64 - 66)$). With this completion, the bound of [20, Theorem 3] yields a variance term of order $\tilde{\mathcal{O}}\left(\frac{\sigma^2 \tau^2}{\mu^2 \varepsilon}\right)$, which is suboptimal with respect to $\tau$. Moreover, the corresponding analysis relies heavily on the assumption of a bounded domain. In [28], the author considers Markovian noise with a finite number of states and manages to obtain a rather interesting result of the form $O\left(\frac{L}{\mu} \log \frac{1}{\varepsilon} + \frac{L\tau\sigma_*^2}{\mu^3\varepsilon}\right)$. Here the first term does not depend on $\tau$, and the second consists only $\sigma^*$ (stochasticity in $x^*$), but the price for this is an additional factor $L/\mu$ in the second term and more strict assumption that all realizations $F(\cdot, z)$ are smooth and strongly convex. In the context of overparameterized learning, our results are almost consistent with the bound of [97, Theorem 1] under i.i.d. sampling. The difference is that the term $\delta^2$ in A 4 can be more pessimistic than the expectation bound in [97].

## 2.2 Lower bounds

We start with a lower bound for the complexity of Markovian stochastic optimization under the assumptions A 1–A 4. Below we provide a result that highlights that the bound of Theorem 1 is tight provided that $\delta$ does not scale with the instance-dependent quantities, e.g., condition number $L/\mu$.

**Theorem 2.** *There exists an instance of the optimization problem satisfying assumptions A 1–A 4 with $\delta = 1$ and arbitrary $\sigma \geq 0, L, \mu > 0, \tau \in \mathbb{N}^*$, such that for any first-order gradient method it takes at least*

$$N = \Omega\left(\tau\sqrt{\frac{L}{\mu}} \log \frac{1}{\varepsilon} + \frac{\tau\sigma^2}{\mu^2\varepsilon}\right)$$

*oracle calls in order to achieve $\mathbb{E}[\|x^N - x^*\|^2] \leq \varepsilon$.*

The proof is provided in Appendix B.5. The idea of the constructed lower for deterministic part bound $\Omega\left(\tau\sqrt{\frac{L}{\mu}} \log \frac{1}{\varepsilon}\right)$ goes back to [76, Theorem 2.1.13]. The stochastic part lower bound goes back to the classical statistical reasoning, and is well explained for i.i.d. noise in [59, Chapter 4.1]. Our adaptation for Markovian setting is based on Le Cam's theory, see [1, Theorem 8], and also [105]. For the case of Markov noise this lower bound is, to the best of our knowledge, original. The closest result to ours is the stochastic term lower bound in [28, Proposition 1], but it is valid only for the vanilla stochastic gradient methods. Below we provide another lower bound showing that the dependence of the sample complexity Corollary 1 on $\delta$ is not an artefact of the proof.

**Proposition 1.** *There exists an instance of the optimization problem satisfying assumptions A 1–A 4 with arbitrary $L, \mu > 0, \tau \in \mathbb{N}^*, \delta = \frac{L}{\mu}$, and $\sigma = 0$, such that for any first-order gradient method it takes at least*

$$N = \Omega\left(\tau\frac{L}{\mu} \log \frac{1}{\varepsilon}\right)$$

*gradient calls in order to achieve $\mathbb{E}[\|x^N - x^*\|^2] \leq \varepsilon$.*

This lower bound is adapted from the information-theoretic lower bound [71]. The detailed proof can be found in Appendix B.5. Recent studies [54, 71, 13] have revealed the impossibility of accelerating stochastic gradient descent (SGD) for online linear regression problems with specific noise structures. To address this issue, researchers have proposed various solutions, such as the MaSS algorithm [62] and the approach presented in [48]. However, these methods rely heavily on the particular structure of the online regression setup. Another question that naturally arises is whether one can get rid of the dependence on $\tau$ in the deterministic part of (8) if $\delta = 0$. The following counterexample shows that this is not the case in general.

**Proposition 2.** *There exists an instance of the optimisation problem satisfying assumptions A 1–A 4 with with arbitrary $L, \mu > 0, \tau \in \mathbb{N}^*, \sigma = 1, \delta = 0$, such that for any first-order gradient method it takes at least*

$$N = \Omega\left(\left(\tau + \sqrt{\frac{L}{\mu}}\right) \log \frac{1}{\varepsilon}\right)$$

*oracle calls in order to achieve $\mathbb{E}[\|x^N - x^*\|^2] \leq \varepsilon$.*

The proof is provided in Appendix B.5.

## 2.3 Non-convex problems

Now we proceed with a randomized batch size version of the simple SGD algorithm. It is summarized in Algorithm 2 and can be shown to achieve optimal rates of convergence for smooth non-convex problems. For the case of non-convex problems with Markov noise similar analysis appeared in [21, Theorem 4].

---

**Algorithm 2** Randomized GD

---

1: **Parameters:** stepsize $\gamma > 0$, number of iterations $K$, bound on batchsize $B$, mixing time $\tau$;
2: **Initialization:** choose $x^0 \in \mathcal{X}$
3: **for** $k = 0, 1, 2, \ldots, N-1$ **do**
4: $\quad$ Sample $J_k \sim \text{Geom}\left(\frac{1}{2}\right)$
5: $\quad g^k = g_0^k + \begin{cases} 2^{J_k}\left(g_{J_k}^k - g_{J_k-1}^k\right), & \text{if } 2^{J_k} \leq M \\ 0, & \text{otherwise} \end{cases}$ $\quad$ with $\quad g_j^k = 2^{-j}B^{-1}\sum_{i=1}^{2^j B}\nabla f(x^k, Z_{T^k+i})$
6: $\quad x^{k+1} = x^k - \gamma g^k$
7: $\quad T^{k+1} = T^k + 2^{J_k}B$
8: **end for**

---

By balancing the values of $B$ and $M$ with Lemma 2, we establish the following result:

**Theorem 3.** *Assume A 1, A 3, A 4. Let problem (1) be solved by Algorithm 2. Let $f^*$ be a global (maybe not unique) minimum of $f$. Then for any $b \in \mathbb{N}^*$, and $\gamma$, $M$ satisfying*

$$\gamma \lesssim \left(L[1 + \delta^2\tau b^{-1} + \delta^2\tau^2 b^{-2}]\right)^{-1}, \quad M \simeq \max\{2; \sqrt{\gamma^{-1}L^{-1}}\}, \quad B = \lceil b\log_2 M\rceil,$$

*it holds that*

$$\mathbb{E}\left[\frac{1}{N}\sum_{k=0}^{N-1}\|\nabla f(x^k)\|^2\right] \lesssim \frac{f(x^0) - f^*}{\gamma N} + L\gamma \cdot \left[\sigma^2\tau b^{-1} + \sigma^2\tau^2 b^{-2}\right].$$

The proof is provided in Appendix B.6. The next corollary immediately follows from the theorem.

**Corollary 2.** *Under the conditions of Theorem 3, if we choose $b = \tau$ and $\gamma$ given by*

$$\gamma \simeq \min\left\{\frac{1}{L(1+\delta^2)}; \sqrt{\frac{f(x^0) - f^*}{LN\sigma^2}}\right\},$$

*then to achieve $\varepsilon$-solution (in terms of $\mathbb{E}[\|\nabla f(x)\|^2] \lesssim \varepsilon^2$) we need*

$$\tilde{\mathcal{O}}\left(\tau \cdot \left[\frac{(1+\delta^2)L(f(x^0) - f^*)}{\varepsilon^2} + \frac{L(f(x^0) - f^*)\sigma^2}{\varepsilon^4}\right]\right) \quad \text{oracle calls.}$$

**Comparison.** The respective bound for the non-convex setting provided in [20, Theorem 1] yields the sample complexity of order $\tilde{\mathcal{O}}\left(\frac{\tau^2 L(f(x^0) - f(x^*))\sigma^2}{\varepsilon^4}\right)$. Also we can note the results of [28, Theorem 2] with the following estimate $O\left(\frac{\tau(L(f(x^0) - f(x^*)) + \sigma^2)}{\varepsilon^2} + \frac{\tau(L(f(x^0) - f(x^*)) + \sigma^2)\sigma^2}{\varepsilon^4}\right)$.

To achieve linear convergence rates in the non-convex setting we can use the Polyak-Lojasiewicz (PL) condition [83]. The respective result is provided in Appendix B.7.

## 2.4 Variational inequalities

In this section, we are interested in the following problem:

$$\text{Find } x^* \in \mathcal{X} \text{ such that } \langle F(x^*), x - x^*\rangle + r(x) - r(x^*) \geq 0 \text{ for all } x \in \mathcal{X}. \tag{9}$$

Here $F : \mathbb{R}^d \to \mathbb{R}^d$ an operator, $\mathcal{X}$ a convex set, and $r : \mathbb{R}^d \to \mathbb{R}$ is a regularization term (a suitable lower semicontinuous convex function) which is assumed to have a simple structure. As mentioned earlier, this problem is quite general and covers a wide range of possible problem formulations. For example, if the operator $F$ is the gradient of a convex function $f$, then the problem (9) is equivalent to the composite minimization problem [6], i.e., minimization of $f(x) + r(x)$. In the meantime, (9) is also a reformulation of the min-max problem

$$\min_{x_1 \in \mathcal{X}_1} \max_{x_2 \in \mathcal{X}_2} r_1(x_1) + g(x_1, x_2) - r_2(x_2), \tag{10}$$

with convex-concave continuously differentiable $g$, convex sets $\mathcal{X}_1$, $\mathcal{X}_2$ and convex functions $r_1$, $r_2$. Using the first-order optimality conditions, it is easy to verify that (10) is equivalent to (9) with $x = (x_1^T, x_2^T)^T$, $F(x) = \left(\nabla_{x_1}f(x_1, x_2)^T, -\nabla_{x_2}f(x_1, x_2)^T\right)^T$, and $r(x) = r_1(x_1) + r_2(x_2)$.

**A 5.** *The operator $F$ is $L$-Lipschitz continuous on $\mathcal{X}$ with $L > 0$, i.e., the following inequality holds for all $x, y \in \mathcal{X}$:*

$$\|F(x) - F(y)\| \leq L\|x - y\|, \qquad \forall x, y \in \mathcal{X}.$$

**A 6.** *The operator $F$ is $\mu_F$-strongly monotone on $\mathcal{X}$, i.e., the following inequality holds for all $x, y \in \mathcal{X}$:*

$$\langle F(x) - F(y), x - y \rangle \geq \mu_F \|x - y\|^2. \tag{11}$$

*The function $r$ is $\mu_r$-strongly convex on $\mathcal{X}$, i.e. for all $x, y \in \mathcal{X}$ and any $r'(x) \in \partial r(x)$ we have*

$$r(y) \geq r(x) + \langle r'(x), y - x \rangle + (\mu_r/2)\|x - y\|^2. \tag{12}$$

These two assumptions are more than standard for the study of variational inequalities and are found in all the papers from Table 2. We consider two cases: strongly monotone/convex with $\mu_F + \mu_r > 0$ and monotone/convex with $\mu_F + \mu_r = 0$.

**A 7.** *For all $x \in \mathbb{R}^d$ it holds that $\mathbb{E}_\pi[F(x, Z)] = F(x)$. Moreover, for all $z \in Z$ and $x \in \mathcal{X}$ it holds that*

$$\|F(x, z) - F(x)\|^2 \leq \sigma^2 + \Delta^2\|x - x^*\|^2, \tag{13}$$

*where $x^*$ is some point from the solution set.*

A 7 is found in the literature on variational inequalities [43, 45, 38] and is considered to be analog to A 4 on overparametrized learning.

Just as the Nesterov accelerated method is optimal for smooth convex minimization problems, the ExtraGradient method [57, 72, 52] is optimal for monotone variational inequalities. Therefore, we take it as a base. On the extrapolation step (Line 4) of Algorithm 3, we simply collect a batch of size $B$, but on the main step (Line 8) we use the randomization as in Algorithm 1. The next theorem gives the convergence of our method.

**Theorem 4.** *Assume A 5, A 6 with $\mu_F + \mu_r > 0$, A 3, A 7. Let problem (9) be solved by Algorithm 3. Then for any $b \in \mathbb{N}^*$, and $\gamma, M$ satisfying*

$$\gamma \lesssim \min\left\{ (\mu_F + \mu_r)^{-1}; L^{-1}; (\mu_F + \mu_r)(\Delta^2 \tau b^{-1} + \Delta^2 \tau^2 b^{-2})^{-1}; \sqrt{\Delta^{-2} \tau^{-1} b} \right\},$$

$$M \simeq \max\{2; \sqrt{\gamma^{-1}(\mu_F + \mu_r)^{-1}}\}, \quad B = \lceil b \log_2 M \rceil,$$

*it holds that*

$$\mathbb{E}\left[\|x^N - x^*\|^2\right] \lesssim \exp\left(-\frac{N(\mu_F + \mu_r)\gamma}{2}\right)\|x^0 - x^*\|^2 + \frac{\gamma}{\mu}(\sigma^2 \tau b^{-1} + \sigma^2 \tau^2 b^{-2}).$$

The proof is postponed to Appendix B.8. One can get an estimate on oracle complexity.

---

**Algorithm 3** `Randomized ExtraGradient`

---

1: **Parameters:** stepsize $\gamma > 0$, number of iterations $N$
2: **Initialization:** choose $x^0 \in \mathcal{X}$
3: **for** $k = 0, 1, 2, \ldots, N - 1$ **do**
4: $\quad x^{k+1/2} = \text{prox}_{\gamma r}\left(x^k - \gamma B^{-1} \sum_{i=1}^{B} F(x^k, Z_{T^k + i})\right)$
5: $\quad T^{k+1/2} = T^k + B$
6: $\quad$ Sample $J_k \sim \text{Geom}\left(\frac{1}{2}\right)$
7: $\quad g^k = g_0^k + \begin{cases} 2^{J_k}\left(g_{J_k}^k - g_{J_k-1}^k\right), & \text{if } 2^{J_k} \leq M \\ 0, & \text{otherwise} \end{cases}$ with $\quad g_j^k = 2^{-j}B^{-1}\sum_{i=1}^{2^j \cdot B} F(x^{k+1/2}, Z_{T^{k+1/2} + i})$
8: $\quad x^{k+1} = \text{prox}_{\gamma r}\left(x^k - \gamma g^k\right)$
9: $\quad T^{k+1} = T^{k+1/2} + 2^{J_k}B$
10: **end for**

---

**Corollary 3.** *Under the conditions of Theorem 4, if we choose $b = \tau$ and $\gamma$ as follows*

$$\gamma \simeq \min\left\{ \frac{1}{\mu_F + \mu_r}; \frac{1}{L}; \frac{\mu_F + \mu_r}{\Delta^2}; \frac{1}{\Delta}; \frac{1}{N(\mu_F + \mu_r)} \ln\left(\max\left\{2; \frac{\mu N\|x^0 - x^*\|^2}{\sigma^2}\right\}\right)\right\},$$

*then to achieve $\varepsilon$-solution (in terms of $\mathbb{E}[\|x - x^*\|^2] \lesssim \varepsilon$) we need*

$$\tilde{\mathcal{O}}\left(\tau \cdot \left[\left(1 + \frac{L}{\mu_F + \mu_r} + \frac{\Delta}{\mu_F + \mu_r} + \frac{\Delta^2}{(\mu_F + \mu_r)^2}\right)\log\frac{1}{\varepsilon} + \frac{\sigma^2}{\mu^2 \varepsilon}\right]\right) \quad \text{oracle calls.}$$

Note that one provide an (almost) matching lower complexity bounds for variational inequalities via lower bounds for saddle point problems, which are a special case of variational inequalities. The method for obtaining lower bounds for saddle point problems is reduced to obtaining estimates for the strongly convex minimization problem (see [106, 40] for respective deterministic lower bounds), which we provide in Section 2.2. Similarly, the question of constructing a lower bound which is tight w.r.t. $\Delta$ remains open.

For the monotone case, we use the *gap function* as a convergence criterion:

$$\text{Gap}(x) = \sup_{y \in \mathcal{X}} \left[ \langle F(y), x - y \rangle + r(x) - r(y) \right] . \tag{14}$$

Such a criterion is standard and classical for monotone variational inequalities [72, 52]. An important assumption for the gap function is the boundedness of the set $\mathcal{X}$.

**A 8.** *The set $\mathcal{X}$ is bounded and has a diameter D, i.e., for all $x, y \in \mathcal{X}$: $\|x - y\|^2 \leq D^2$.*

A 8 can be slightly relaxed. We need to use a simple trick from [77]. In particular, we need to consider $\mathcal{C}$ – a compact subset of $\mathcal{X}$ and change $\mathcal{X}$ to $\mathcal{C}$ in (14). But such a technique is rather technical and does not change the essence. Finally, the following result holds.

**Theorem 5.** *Assume A 5, A 6 with $\mu_F + \mu_r = 0$, A 8, A 3, A 7. Let problem (9) be solved by Algorithm 3. Then for any $B \in \mathbb{N}^*$, and $\gamma, M$ satisfying $\gamma \lesssim L^{-1}$, $M = \sqrt{N}$, it holds that*

$$\mathbb{E}\left[ Gap(\bar{x}^N) \right] \lesssim \frac{D^2}{\gamma N} + \gamma (\tau B^{-1} \log_2 N + \tau^2 B^{-2})(\sigma^2 + \Delta^2 D^2) \text{ where } \bar{x}^N = \frac{1}{N} \sum_{k=0}^{N-1} x^{k+1/2} .$$

The proof is postponed to Appendix B.9. The following corollary holds.

**Corollary 4.** *Under the conditions of Theorem 5, if we choose $B = \tau$ and $\gamma$ as follows*

$$\gamma \simeq \min \left\{ \frac{1}{L}; \sqrt{\frac{D^2}{(\sigma^2 + \Delta^2 D^2)N}} \right\} ,$$

*then to achieve $\varepsilon$-solution (in terms of $\mathbb{E}[Gap(x)] \lesssim \varepsilon$) we need*

$$\tilde{\mathcal{O}}\left( \tau \left[ \frac{LD^2}{\varepsilon} + \frac{\sigma^2 D^2 + \Delta^2 D^4}{\varepsilon^2} \right] \right) \quad \text{oracle calls.}$$

**Comparison.** These results is the first for variational inequalities with Markovian stochasticity, either in the strongly monotone or monotone cases. The only close work is [99]. The authors work with convex-concave saddle point problems and provide the following estimate on the oracle complexity $\mathcal{O}\left( \tau^2 \cdot \frac{G^4}{\varepsilon^2} + \frac{D^2}{\varepsilon^2} \right)$ (with $G$ – the uniform bound of the operator), which is worse than ours at least in terms of $\tau$. Moreover, the authors consider the case of a finite Markov chain, which is a special case of our setup.

## 3 Conclusion

In this paper, we present a unified random batch size framework that achieves optimal finite-time performance for non-convex and strongly convex optimization problems with Markov noise, as well as for variational inequalities. Unlike existing methods, our framework relaxes the assumptions typically imposed on the domain and stochastic gradient oracle. We also provide a variety of lower bounds, which are to the best of our knowledge original in the Markov setting.

## Acknowledgments

This research of A. Beznosikov has been supported by The Analytical Center for the Government of the Russian Federation (Agreement No. 70-2021-00143 dd. 01.11.2021, IGK 000000D730321P5Q0002). E. Moulines received support from the grant ANR-19-CHIA-002 SCAI and parts of his work has been done under the auspices of Lagrange Center for maths and computing.

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
