# B Proofs of Section 2.1, Section 2.3

## B.1 Proof of Lemma 1

**Lemma 3** (Lemma 1). *Assume A 3 and A 4. Then, for any $n \geq 1$ and $x \in \mathbb{R}^d$, it holds that*

$$\mathbb{E}_{\pi}[\|n^{-1} \sum_{i=1}^{n} \nabla F(x, Z_i) - \nabla f(x)\|^2] \leq \tfrac{8\tau}{n} \left(\sigma^2 + \delta^2 \|\nabla f(x)\|^2\right). \tag{15}$$

*Moreover, for any initial distribution $\xi$ on $(\mathsf{Z}, \mathcal{Z})$, that*

$$\mathbb{E}_{\xi}[\|n^{-1} \sum_{i=1}^{n} \nabla F(x, Z_i) - \nabla f(x)\|^2] \leq \tfrac{C_1 \tau}{n} \left(\sigma^2 + \delta^2 \|\nabla f(x)\|^2\right), \tag{16}$$

*where $C_1 = 16(1 + \frac{1}{\ln^2 4})$.*

By [22, Lemma 19.3.6 and Theorem 19.3.9 ], for any two probabilities $\xi, \xi'$ on $(\mathsf{Z}, \mathcal{Z})$ there is a *maximal exact coupling* $(\Omega, \mathcal{F}, \tilde{\mathbb{P}}_{\xi,\xi'}, Z, Z', T)$ of $\mathbb{P}_{\xi}^{\mathsf{Q}}$ and $\mathbb{P}_{\xi'}^{\mathsf{Q}}$, that is,

$$\|\xi \mathsf{Q}^n - \xi' \mathsf{Q}^n\|_{\mathrm{TV}} = 2\tilde{\mathbb{P}}_{\xi,\xi'}(T > n). \tag{17}$$

We write $\tilde{\mathbb{E}}_{\xi,\xi'}$ for the expectation with respect to $\tilde{\mathbb{P}}_{\xi,\xi'}$. Using the coupling construction (17),

$$\mathbb{E}_{\xi}^{1/2}[\| \sum_{i=1}^{n} \{\nabla f(x, Z_i) - \nabla f(x)\}\|^2] \leq \mathbb{E}_{\pi}^{1/2}[\| \sum_{i=0}^{n-1} \nabla f(x, Z_i) - \nabla f(x)\|^2] +$$
$$\tilde{\mathbb{E}}_{\xi,\pi}^{1/2}[\| \sum_{i=0}^{n-1} \{\nabla f(x, Z_i) - \nabla f(x, Z_i')\}\|^2].$$

The first term is bounded with (15). Moreover, with (17) and A 4, we get

$$\| \sum_{i=0}^{n-1} \{\nabla f(x, Z_i) - \nabla f(x, Z_i')\}\|^2 \leq 8 \left(\sigma^2 + \delta^2 \|\nabla f(x)\|^2\right) \left(\sum_{i=0}^{n-1} \mathbb{1}_{\{Z_i \neq Z_i'\}}\right)^2$$

$$= 8 \left(\sigma^2 + \delta^2 \|\nabla f(x)\|^2\right) \left(\sum_{i=0}^{n-1} \mathbb{1}_{\{T > i\}}\right)^2$$

$$\leq 16 \left(\sigma^2 + \delta^2 \|\nabla f(x)\|^2\right) \sum_{i=1}^{\infty} i \, \mathbb{1}_{\{T > i\}}.$$

Thus, using the assumption A 3, we bound

$$\tilde{\mathbb{E}}_{\xi,\pi}[\sum_{i=1}^{\infty} i \, \mathbb{1}_{\{T > i\}}] = \sum_{i=1}^{\infty} i \tilde{\mathbb{P}}_{\xi,\xi'}(T > i) = \sum_{i=1}^{\infty} i(1/4)^{\lfloor i/\tau \rfloor} \leq 4 \sum_{i=1}^{\infty} i(1/4)^{i/\tau}.$$

Now we set $\rho = (1/4)^{1/\tau}$ and use an upper bound

$$\sum_{k=1}^{\infty} k\rho^k \leq \rho^{-1} \int_{0}^{+\infty} x^p \rho^x \, dx \leq \rho^{-1} \left(\ln \rho^{-1}\right)^{-2} \Gamma(2) = \rho^{-1} \left(\ln \rho^{-1}\right)^{-2} = \frac{\tau^2}{(1/4)^{1/\tau} \ln^2 4}.$$

Combining the bounds above yields

$$\mathbb{E}_{\xi}[\|n^{-1} \sum_{i=1}^{n} \nabla f(x, Z_i) - \nabla f(x)\|^2] \leq \frac{c_1 \tau}{n} \left(\sigma^2 + \delta^2 \|\nabla f(x)\|^2\right) + \frac{c_2 \tau^2}{n^2} \left(\sigma^2 + \delta^2 \|\nabla f(x)\|^2\right),$$

where $c_1 = 16$, $c_2 = \frac{128(1/4)^{-1/\tau}}{\ln^2 4}$. Now we consider the two cases. If $n < c_1\tau$, we get from Minkowski's inequality that

$$\mathbb{E}_\xi[\|n^{-1}\sum_{i=1}^{n}\nabla f(x, Z_i) - \nabla f(x)\|^2] \leq 2\sigma^2 + 2\delta^2\|\nabla f(x)\|^2,$$

and (16) holds. If $n > c_1\tau$, it holds that

$$\frac{c_2\tau^2}{n^2}\left(\sigma^2 + \delta^2\|\nabla f(x)\|^2\right) \leq \frac{c_2\tau^2}{nc_1\tau}\left(\sigma^2 + \delta^2\|\nabla f(x)\|^2\right),$$

and we also get (16).

## B.2 Proof of Lemma 2

Before we proceed to the proof, we give a statement of Lemma 2 with exact constants.

**Lemma 4** (Lemma 2). *Assume A 3 and A 4. Then for the gradient estimates $g^k$ from Algorithm 1 it holds that $\mathbb{E}_k[g^k] = \mathbb{E}_k[g^k_{\lfloor\log_2 M\rfloor}]$. Moreover,*

$$\mathbb{E}_k[\|\nabla f(x_g^k) - g^k\|^2] \leq \left(4C_1\tau B^{-1}\log_2 M + (4C_1 + 2)\tau^2 B^{-2}\right)\left(\sigma^2 + \delta^2\|\nabla f(x_g^k)\|^2\right), \quad (18)$$
$$\|\nabla f(x_g^k) - \mathbb{E}_k[g^k]\|^2 \leq C_2\tau^2 M^{-2}B^{-2}(\sigma^2 + \delta^2\|\nabla f(x_g^k)\|^2),$$

*where $C_1$ is defined in (16) and $C_2 = 256/3$.*

*Proof.* To show that $\mathbb{E}_k[g^k] = \mathbb{E}_k[g^k_{\lfloor\log_2 M\rfloor}]$ we simply compute conditional expectation w.r.t. $J_k$:

$$\mathbb{E}_k[g^k] = \mathbb{E}_k\left[\mathbb{E}_{J_k}[g^k]\right] = \mathbb{E}_k[g_0^k] + \sum_{i=1}^{\lfloor\log_2 M\rfloor}\mathbb{P}\{J_k = i\} \cdot 2^i\mathbb{E}_k[g_i^k - g_{i-1}^k]$$

$$= \mathbb{E}_k[g_0^k] + \sum_{i=1}^{\lfloor\log_2 M\rfloor}\mathbb{E}_k[g_i^k - g_{i-1}^k] = \mathbb{E}_k[g^k_{\lfloor\log_2 M\rfloor}].$$

We start with the proof of the first statement of (18) by taking the conditional expectation for $J_k$:

$$\mathbb{E}_k[\|\nabla f(x_g^k) - g^k\|^2] \leq 2\mathbb{E}_k[\|\nabla f(x_g^k) - g_0^k\|^2] + 2\mathbb{E}_k[\|g^k - g_0^k\|^2]$$

$$= 2\mathbb{E}_k[\|\nabla f(x_g^k) - g_0^k\|^2] + 2\sum_{i=1}^{\lfloor\log_2 M\rfloor}\mathbb{P}\{J_k = i\} \cdot 4^i\mathbb{E}_k[\|g_i^k - g_{i-1}^k\|^2]$$

$$= 2\mathbb{E}_k[\|\nabla f(x_g^k) - g_0^k\|^2] + 2\sum_{i=1}^{\lfloor\log_2 M\rfloor}2^i\mathbb{E}_k[\|g_i^k - g_{i-1}^k\|^2]$$

$$\leq 2\mathbb{E}_k[\|\nabla f(x_g^k) - g_0^k\|^2] + 4\sum_{i=1}^{\lfloor\log_2 M\rfloor}2^i\left(\mathbb{E}_k[\|\nabla f(x_g^k) - g_{i-1}^k\|^2] + \mathbb{E}_k[\|g_i^k - \nabla f(x_g^k)\|^2]\right).$$

To bound $\mathbb{E}_k[\|\nabla f(x_g^k) - g_0^k\|^2]$, $\mathbb{E}_k[\|\nabla f(x_g^k) - g_{i-1}^k\|^2]$, $\mathbb{E}_k[\|g_i^k - \nabla f(x_g^k)\|^2]$, we apply Lemma 1 and get

$$\mathbb{E}_k[\|\nabla f(x_g^k) - g^k\|^2] \leq 2\sigma^2 + 4\sum_{i=1}^{\lfloor\log_2 M\rfloor}2^i\left(\frac{C_1\tau}{2^i B}(\sigma^2 + \delta^2\|\nabla f(x_g^k)\|^2) + \frac{C_1\tau^2}{2^{2i}B^2}(\sigma^2 + \delta^2\|\nabla f(x_g^k)\|^2)\right)$$

$$\leq \frac{4C_1(\sigma^2 + \delta^2\|\nabla f(x_g^k)\|^2)\tau\log_2 M}{B} + \frac{(4C_1 + 2)(\sigma^2 + \delta^2\|\nabla f(x_g^k)\|^2)\tau^2}{B^2}.$$

To show the second part of the statement, we use Lemma 2 and get

$$\|\nabla f(x_g^k) - \mathbb{E}_k[g^k]\|^2 = \|\nabla f(x_g^k) - \mathbb{E}_k[g^k_{\lfloor\log_2 M\rfloor}]\|^2.$$

The remaining proof once again uses Lemma 1 and is omitted. To conclude we use that $2^{\lfloor\log_2 M\rfloor} \geq M/2$. □

## B.3 Proof of Theorem 1.

We preface the proof by two technical Lemmas.

**Lemma 5.** *Assume A 1 and A 2. Then for the iterates of Algorithm 1 with $\theta = (p\eta^{-1}-1)/(\beta p\eta^{-1}-1)$,
$\theta > 0$, $\eta \geq 1$, $p > 0$, it holds that*

$$
\begin{aligned}
\mathbb{E}_k[\|x^{k+1} - x^*\|^2] \leq & (1 + \alpha p\gamma\eta)(1-\beta)\|x^k - x^*\|^2 + (1 + \alpha p\gamma\eta)\beta\|x_g^k - x^*\|^2 \\
& + (1 + \alpha p\gamma\eta)(\beta^2 - \beta)\|x^k - x_g^k\|^2 + p^2\eta^2\gamma^2\mathbb{E}_k[\|g^k\|^2] \\
& - 2\eta^2\gamma\langle\nabla f(x_g^k), x_g^k + (p\eta^{-1} - 1)x_f^k - \eta^{-1}px^*\rangle \\
& + \frac{p\eta\gamma}{\alpha}\|\mathbb{E}_k[g^k] - \nabla f(x_g^k)\|^2,
\end{aligned}
\tag{19}
$$

*where $\alpha > 0$ is any positive constant.*

*Proof.* We start with lines 8 and 7 of Algorithm 1:

$$
\begin{aligned}
\|x^{k+1} - x^*\|^2 = & \|\eta x_f^{k+1} + (p-\eta)x_f^k + (1-p)(1-\beta)x^k + (1-p)\beta x_g^k - x^*\|^2 \\
= & \|\eta x_g^k - p\eta\gamma g^k + (p-\eta)x_f^k + (1-p)(1-\beta)x^k + (1-p)\beta x_g^k - x^*\|^2 \\
= & \|\eta x_g^k + (p-\eta)x_f^k + (1-p)(1-\beta)x^k + (1-p)\beta x_g^k - x^*\|^2 + p^2\gamma^2\eta^2\|g^k\|^2 \\
& - 2p\gamma\eta\langle g^k, \eta x_g^k + (p-\eta)x_f^k + (1-p)(1-\beta)x^k + (1-p)\beta x_g^k - x^*\rangle.
\end{aligned}
$$

Using straightforward algebra, we get

$$
\begin{aligned}
\|x^{k+1} - x^*\|^2 = & \|\eta x_g^k + (p-\eta)x_f^k + (1-p)(1-\beta)x^k + (1-p)\beta x_g^k - x^*\|^2 + p^2\gamma^2\eta^2\|g^k\|^2 \\
& - 2p\gamma\eta\langle\nabla f(x_g^k), \eta x_g^k + (p-\eta)x_f^k + (1-p)(1-\beta)x^k + (1-p)\beta x_g^k - x^*\rangle \\
& - 2p\gamma\eta\langle\mathbb{E}_k[g^k] - \nabla f(x_g^k), \eta x_g^k + (p-\eta)x_f^k + (1-p)(1-\beta)x^k + (1-p)\beta x_g^k - x^*\rangle \\
& - 2p\gamma\eta\langle g^k - \mathbb{E}_k[g^k], \eta x_g^k + (p-\eta)x_f^k + (1-p)(1-\beta)x^k + (1-p)\beta x_g^k - x^*\rangle \\
\leq & (1 + \alpha p\eta\gamma)\|\eta x_g^k + (p-\eta)x_f^k + (1-p)(1-\beta)x^k + (1-p)\beta x_g^k - x^*\|^2 \\
& - 2p\gamma\eta\langle\nabla f(x_g^k), \eta x_g^k + (p-\eta)x_f^k + (1-p)(1-\beta)x^k + (1-p)\beta x_g^k - x^*\rangle \\
& - 2p\gamma\eta\langle g^k - \mathbb{E}_k[g^k], \eta x_g^k + (p-\eta)x_f^k + (1-p)(1-\beta)x^k + (1-p)\beta x_g^k - x^*\rangle \\
& + p^2\gamma^2\eta^2\|g^k\|^2 + \frac{p\gamma\eta}{\alpha}\|\mathbb{E}_k[g^k] - \nabla f(x_g^k)\|^2.
\end{aligned}
$$

In the last step we also applied Cauchy-Schwartz inequality in the form (43) with $\alpha > 0$. Taking the conditional expectation, we get

$$
\begin{aligned}
\mathbb{E}_k[\|x^{k+1} - x^*\|^2] \leq & (1 + \alpha p\eta\gamma)\|\eta x_g^k + (p-\eta)x_f^k + (1-p)(1-\beta)x^k + (1-p)\beta x_g^k - x^*\|^2 \\
& - 2p\gamma\eta\langle\nabla f(x_g^k), \eta x_g^k + (p-\eta)x_f^k + (1-p)(1-\beta)x^k + (1-p)\beta x_g^k - x^*\rangle \\
& + p^2\gamma^2\eta^2\mathbb{E}_k[\|g^k\|^2] + \frac{p\gamma\eta}{\alpha}\|\mathbb{E}_k[g^k] - \nabla f(x_g^k)\|^2.
\end{aligned}
\tag{20}
$$

Now let us handle expression $\|\eta x_g^k + (p-\eta)x_f^k + (1-p)(1-\beta)x^k + (1-p)\beta x_g^k - x^*\|^2$ for a while. Taking into account line 4 and the choice of $\theta$ such that $\theta = (p\eta^{-1} - 1)/(\beta p\eta^{-1} - 1)$ (in particular, $(p\eta^{-1} - 1) = (\beta p\eta^{-1} - 1)\theta$ and $\eta(1 - \beta p\eta^{-1})(1 - \theta) = p(1 - \beta)$), we get

$$
\begin{aligned}
\eta x_g^k & + (p-\eta)x_f^k + (1-p)(1-\beta)x^k + (1-p)\beta x_g^k \\
& = (\eta + (1-p)\beta)x_g^k + (p-\eta)x_f^k + (1-p)(1-\beta)x^k \\
& = (\eta + (1-p)\beta)x_g^k + \eta(p\eta^{-1} - 1)x_f^k + (1-p)(1-\beta)x^k \\
& = (\eta + (1-p)\beta)x_g^k + \eta(\beta p\eta^{-1} - 1)\theta x_f^k + (1-p)(1-\beta)x^k \\
& = (\eta + (1-p)\beta)x_g^k + \eta(\beta p\eta^{-1} - 1)(x_g^k - (1-\theta)x^k) + (1-p)(1-\beta)x^k \\
& = \beta x_g^k - \eta(\beta p\eta^{-1} - 1)(1-\theta)x^k + (1-p)(1-\beta)x^k \\
& = \beta x_g^k + p(1-\beta)x^k + (1-p)(1-\beta)x^k
\end{aligned}
$$

$$= \beta x_g^k + (1-\beta)x^k .$$

Substituting into $\|\eta x_g^k + (p-\eta)x_f^k + (1-p)(1-\beta)x^k + (1-p)\beta x_g^k - x^*\|^2$, we get

$$\|\eta x_g^k + (p-\eta)x_f^k + (1-p)(1-\beta)x^k + (1-p)\beta x_g^k - x^*\|^2$$
$$= \|\beta x_g^k + (1-\beta)x^k - x^*\|^2$$
$$= \|x^k - x^* + \beta(x_g^k - x^k)\|^2$$
$$= \|x^k - x^*\|^2 + 2\beta\langle x^k - x^*, x_g^k - x^k\rangle + \beta^2\|x^k - x_g^k\|^2$$
$$= \|x^k - x^*\|^2 + \beta\left(\|x_g^k - x^*\|^2 - \|x^k - x^*\|^2 - \|x_g^k - x^k\|^2\right) + \beta^2\|x^k - x_g^k\|^2$$
$$= (1-\beta)\|x^k - x^*\|^2 + \beta\|x_g^k - x^*\|^2 + (\beta^2 - \beta)\|x^k - x_g^k\|^2. \tag{21}$$

Again with line 4 and the choice of $\theta$ such that $\theta = (p\eta^{-1} - 1)/(\beta p\eta^{-1} - 1)$ (in particular, $\eta^{-1}p(1-\beta) = (1 - \beta p\eta^{-1})(1-\theta)$ and $(\beta p\eta^{-1} - 1)\theta = (p\eta^{-1} - 1)$), one can also note

$$\eta x_g^k + (p-\eta)x_f^k + (1-p)(1-\beta)x^k + (1-p)\beta x_g^k - x^*$$
$$= (\eta + (1-p)\beta)x_g^k + (p-\eta)x_f^k + (1-p)(1-\beta)x^k - x^*$$
$$= \eta p^{-1}\left((p + (1-p)\eta^{-1}p\beta)x_g^k + (p\eta^{-1} - 1)px_f^k + (1-p)(1-\beta)p\eta^{-1}x^k - \eta^{-1}px^*\right)$$
$$= \eta p^{-1}\left((p + (1-p)\eta^{-1}p\beta)x_g^k + (p\eta^{-1} - 1)px_f^k + (1-p)(1-\beta p\eta^{-1})(1-\theta)x^k - \eta^{-1}px^*\right)$$
$$= \eta p^{-1}\left((p + (1-p)\eta^{-1}p\beta)x_g^k + (p\eta^{-1} - 1)px_f^k + (1-p)(1-\beta p\eta^{-1})(x_g^k - \theta x_f^k) - \eta^{-1}px^*\right)$$
$$= \eta p^{-1}\left(x_g^k + (p\eta^{-1} - 1)px_f^k - (1-p)(1 - \beta p\eta^{-1})\theta x_f^k - \eta^{-1}px^*\right)$$
$$= \eta p^{-1}\left(x_g^k + (p\eta^{-1} - 1)px_f^k + (1-p)(p\eta^{-1} - 1)x_f^k - \eta^{-1}px^*\right)$$
$$= \eta p^{-1}\left(x_g^k + (p\eta^{-1} - 1)x_f^k - \eta^{-1}px^*\right). \tag{22}$$

Combining (21) and (22) with (20), we finish the proof. $\square$

**Lemma 6.** *Assume A 1-A 2. Let problem* (1) *be solved by Algorithm 1. Then for any $u \in \mathbb{R}^d$, we get*

$$\mathbb{E}_k[f(x_f^{k+1})] \leq f(u) - \langle\nabla f(x_g^k), u - x_g^k\rangle - \frac{\mu}{2}\|u - x_g^k\|^2 - \frac{\gamma}{2}\|\nabla f(x_g^k)\|^2$$
$$+ \frac{\gamma}{2}\|\mathbb{E}_k[g^k] - \nabla f(x_g^k)\|^2 + \frac{L\gamma^2}{2}\mathbb{E}_k[\|g^k\|^2].$$

*Proof.* Using A 1 in the form (42) with $x = x_f^{k+1}$, $y = x_g^k$ and line 7 of Algorithm 1, we get

$$f(x_f^{k+1}) \leq f(x_g^k) + \langle\nabla f(x_g^k), x_f^{k+1} - x_g^k\rangle + \frac{L}{2}\|x_f^{k+1} - x_g^k\|^2$$
$$= f(x_g^k) - p\gamma\langle\nabla f(x_g^k), g^k\rangle + \frac{Lp^2\gamma^2}{2}\|g^k\|^2$$
$$= f(x_g^k) - p\gamma\langle\nabla f(x_g^k), \nabla f(x_g^k)\rangle - p\gamma\langle\nabla f(x_g^k), \mathbb{E}_k[g^k] - \nabla f(x_g^k)\rangle$$
$$\quad - p\gamma\langle\nabla f(x_g^k), g^k - \mathbb{E}_k[g^k]\rangle + \frac{Lp^2\gamma^2}{2}\|g^k\|^2$$
$$\leq f(x_g^k) - p\gamma\|\nabla f(x_g^k)\|^2 + \frac{p\gamma}{2}\|\nabla f(x_g^k)\|^2 + \frac{p\gamma}{2}\|\mathbb{E}_k[g^k] - \nabla f(x_g^k)\|^2$$
$$\quad - p\gamma\langle\nabla f(x_g^k), g^k - \mathbb{E}_k[g^k]\rangle + \frac{Lp^2\gamma^2}{2}\|g^k\|^2.$$

Here we also used Cauchy Schwartz inequality (43) with $a = \nabla f(x_g^k)$, $b = \nabla f(x_g^k) - \mathbb{E}_k[g^k]$ and $c = 1$. Taking the conditional expectation, we get

$$\mathbb{E}_k[f(x_f^{k+1})] \leq f(x_g^k) - \frac{p\gamma}{2}\|\nabla f(x_g^k)\|^2 + \frac{p\gamma}{2}\|\mathbb{E}_k[g^k] - \nabla f(x_g^k)\|^2 + \frac{Lp^2\gamma^2}{2}\mathbb{E}_k[\|g^k\|^2].$$

Using A 2 with $x = u$ and $y = x_g^k$, one can conclude that for any $u \in \mathbb{R}^d$ it holds

$$\mathbb{E}_k[f(x_f^{k+1})] \leq f(u) - \langle\nabla f(x_g^k), u - x_g^k\rangle - \frac{\mu}{2}\|u - x_g^k\|^2 - \frac{p\gamma}{2}\|\nabla f(x_g^k)\|^2$$

$$+ \frac{p\gamma}{2}\|\mathbb{E}_k[g^k] - \nabla f(x_g^k)\|^2 + \frac{Lp^2\gamma^2}{2}\mathbb{E}_k[\|g^k\|^2].$$

$\square$

**Theorem 6** (Theorem 1). *Assume A 1 – A 4. Let problem* (1) *be solved by Algorithm 1. Then for any* $b \in \mathbb{N}^*$, $\gamma \in (0; \frac{3}{4L}]$, *and* $\beta, \theta, \eta, p, M, B$ *satisfying*

$$p = \left[1 + 2\left(1 + \gamma L\right)\left(1 + 4\left[C_1\tau b^{-1} + (C_1 + 1)\tau^2 b^{-2}\right]\delta^2\right)\right]^{-1},$$

$$\beta = \sqrt{\frac{4p^2\mu\gamma}{3}}, \quad \eta = \frac{3\beta}{2p\mu\gamma} = \sqrt{\frac{3}{\mu\gamma}}, \quad \theta = \frac{p\eta^{-1}-1}{\beta p\eta^{-1}-1},$$

$$M = \max\{2; \sqrt{C_2 p^{-1}(1 + 2p/\beta)}\}, \quad B = \lceil b\log_2 M\rceil.$$

*it holds that*

$$\mathbb{E}\left[\|x^N - x^*\|^2 + \frac{6}{\mu}(f(x_f^N) - f(x^*))\right]$$

$$\lesssim \exp\left(-N\sqrt{\frac{p^2\mu\gamma}{3}}\right)\left[\|x^0 - x^*\|^2 + \frac{6}{\mu}(f(x^0) - f(x^*))\right] + \frac{p\sqrt{\gamma}}{\mu^{3/2}}\left(\sigma^2\tau b^{-1} + \sigma^2\tau^2 b^{-2}\right).$$

*Proof.* Using Lemma 6 with $u = x^*$ and $u = x_f^k$, we get

$$\mathbb{E}_k[f(x_f^{k+1})] \leq f(x^*) - \langle\nabla f(x_g^k), x^* - x_g^k\rangle - \frac{\mu}{2}\|x^* - x_g^k\|^2 - \frac{p\gamma}{2}\|\nabla f(x_g^k)\|^2$$

$$+ \frac{p\gamma}{2}\|\mathbb{E}_k[g^k] - \nabla f(x_g^k)\|^2 + \frac{Lp^2\gamma^2}{2}\mathbb{E}_k[\|g^k\|^2],$$

$$\mathbb{E}_k[f(x_f^{k+1})] \leq f(x_f^k) - \langle\nabla f(x_g^k), x_f^k - x_g^k\rangle - \frac{\mu}{2}\|x_f^k - x_g^k\|^2 - \frac{p\gamma}{2}\|\nabla f(x_g^k)\|^2$$

$$+ \frac{p\gamma}{2}\|\mathbb{E}_k[g^k] - \nabla f(x_g^k)\|^2 + \frac{Lp^2\gamma^2}{2}\mathbb{E}_k[\|g^k\|^2].$$

Summing the first inequality with coefficient $2p\gamma\eta$, the second with coefficient $2\gamma\eta(\eta - p)$ and (19), we obtain

$$\mathbb{E}_k[\|x^{k+1} - x^*\|^2 + 2\gamma\eta^2 f(x_f^{k+1})]$$

$$\leq (1 + \alpha p\gamma\eta)(1 - \beta)\|x^k - x^*\|^2 + (1 + \alpha p\gamma\eta)\beta\|x_g^k - x^*\|^2$$

$$+ (1 + \alpha p\gamma\eta)(\beta^2 - \beta)\|x^k - x_g^k\|^2 - 2\eta^2\gamma\langle\nabla f(x_g^k), x_g^k + (p\eta^{-1} - 1)x_f^k - \eta^{-1}px^*\rangle$$

$$+ p^2\eta^2\gamma^2\mathbb{E}_k[\|g^k\|^2] + \frac{p\eta\gamma}{\alpha}\|\mathbb{E}_k[g^k] - \nabla f(x_g^k)\|^2$$

$$+ 2p\gamma\eta\left(f(x^*) - \langle\nabla f(x_g^k), x^* - x_g^k\rangle - \frac{\mu}{2}\|x^* - x_g^k\|^2 - \frac{p\gamma}{2}\|\nabla f(x_g^k)\|^2\right.$$

$$\left.+ \frac{p\gamma}{2}\|\mathbb{E}_k[g^k] - \nabla f(x_g^k)\|^2 + \frac{Lp^2\gamma^2}{2}\mathbb{E}_k[\|g^k\|^2]\right)$$

$$+ 2\gamma\eta(\eta - p)\left(f(x_f^k) - \langle\nabla f(x_g^k), x_f^k - x_g^k\rangle - \frac{\mu}{2}\|x_f^k - x_g^k\|^2 - \frac{p\gamma}{2}\|\nabla f(x_g^k)\|^2\right.$$

$$\left.+ \frac{p\gamma}{2}\|\mathbb{E}_k[g^k] - \nabla f(x_g^k)\|^2 + \frac{Lp^2\gamma^2}{2}\mathbb{E}_k[\|g^k\|^2]\right)$$

$$= (1 + \alpha p\gamma\eta)(1 - \beta)\|x^k - x^*\|^2 + 2\gamma\eta\left(\eta - p\right)f(x_f^k) + 2p\gamma\eta f(x^*)$$

$$+ ((1 + \alpha p\gamma\eta)\beta - p\gamma\eta\mu)\|x_g^k - x^*\|^2$$

$$+ (1 + \alpha p\gamma\eta)(\beta^2 - \beta)\|x^k - x_g^k\|^2 - p\gamma^2\eta^2\|\nabla f(x_g^k)\|^2$$

$$+ \left(\frac{p\eta\gamma}{\alpha} + p\gamma^2\eta^2\right)\|\mathbb{E}_k[g^k] - \nabla f(x_g^k)\|^2 + \left(p^2\eta^2\gamma^2 + p^2\gamma^3\eta^2 L\right)\mathbb{E}_k[\|g^k\|^2]$$

$$\leq (1 + \alpha p\gamma\eta)(1 - \beta)\|x^k - x^*\|^2 + 2\gamma\eta\left(\eta - p\right)f(x_f^k) + 2p\gamma\eta f(x^*)$$

$$+ ((1 + \alpha p\gamma\eta)\beta - p\gamma\eta\mu)\|x_g^k - x^*\|^2$$

$$+ (1 + \alpha p\gamma\eta)(\beta^2 - \beta)\|x^k - x_g^k\|^2 - p\gamma^2\eta^2\|\nabla f(x_g^k)\|^2$$

$$+ p\eta\gamma\left(\frac{1}{\alpha} + \gamma\eta\right)\|\mathbb{E}_k[g^k] - \nabla f(x_g^k)\|^2 + 2p^2\eta^2\gamma^2\left(1 + \gamma L\right)\mathbb{E}_k[\|g^k - \nabla f(x_g^k)\|^2]$$

$$+ 2p^2\eta^2\gamma^2\left(1 + \gamma L\right)\mathbb{E}_k[\|\nabla f(x_g^k)\|^2].$$

In the last step we also used (44) with $c = 1$. Since $\gamma \leq \frac{3}{4L}$, the choice of $\alpha = \frac{\beta}{2p\eta\gamma}$, $\beta = \sqrt{4p^2\mu\gamma/3}$, and $p\mu\gamma\eta = 3\beta/2$ gives

$$\beta = \sqrt{4p^2\mu\gamma/3} \leq \sqrt{p^2\mu/L} \leq 1,$$

$$(1 + \alpha p\eta\gamma)(1 - \beta) = \left(1 + \frac{\beta}{2}\right)(1 - \beta) \leq \left(1 - \frac{\beta}{2}\right),$$

$$((1 + \alpha p\eta\gamma)\beta - p\mu\gamma\eta) = \left(\beta + \frac{\beta^2}{2} - p\mu\gamma\eta\right) \leq \left(\frac{3\beta}{2} - p\mu\gamma\eta\right) \leq 0,$$

and, therefore,

$$\mathbb{E}_k\left[\|x^{k+1} - x^*\|^2 + 2\gamma\eta^2 f(x_f^{k+1})\right]$$
$$\leq (1 - \beta/2)\|x^k - x^*\|^2 + 2\gamma\eta\left(\eta - p\right)f(x_f^k) + 2p\gamma\eta f(x^*)$$
$$+ p\eta^2\gamma^2\left(1 + 2p/\beta\right)\|\mathbb{E}_k[g^k] - \nabla f(x_g^k)\|^2$$
$$+ 2p^2\eta^2\gamma^2\left(1 + \gamma L\right)\mathbb{E}_k[\|g^k - \nabla f(x_g^k)\|^2]$$
$$- p\gamma^2\eta^2(1 - 2p(1 + \gamma L))\|\nabla f(x_g^k)\|^2.$$

Subtracting $2\gamma\eta^2 f(x^*)$ from both sides, we get

$$\mathbb{E}_k\left[\|x^{k+1} - x^*\|^2 + 2\gamma\eta^2(f(x_f^{k+1}) - f(x^*))\right]$$
$$\leq (1 - \beta/2)\|x^k - x^*\|^2 + (1 - p/\eta) \cdot 2\gamma\eta^2(f(x_f^k) - f(x^*))$$
$$+ p\eta^2\gamma^2\left(1 + 2p/\beta\right)\|\mathbb{E}_k[g^k] - \nabla f(x_g^k)\|^2$$
$$+ 2p^2\eta^2\gamma^2\left(1 + \gamma L\right)\mathbb{E}_k[\|g^k - \nabla f(x_g^k)\|^2]$$
$$- p\gamma^2\eta^2(1 - 2p(1 + \gamma L))\|\nabla f(x_g^k)\|^2.$$

Applying Lemma 4, one can obtain

$$\mathbb{E}_k\left[\|x^{k+1} - x^*\|^2 + 2\gamma\eta^2(f(x_f^{k+1}) - f(x^*))\right]$$
$$\leq (1 - \beta/2)\|x^k - x^*\|^2 + (1 - p/\eta) \cdot 2\gamma\eta^2(f(x_f^k) - f(x^*))$$
$$+ p\eta^2\gamma^2\left(1 + 2p/\beta\right) \cdot C_2\tau^2 M^{-2}B^{-2}(\sigma^2 + \delta^2\|\nabla f(x_g^k)\|^2)$$
$$+ 2p^2\eta^2\gamma^2\left(1 + \gamma L\right) \cdot \left(4C_1\tau B^{-1}\log_2 M + (4C_1 + 2)\tau^2 B^{-2}\right)(\sigma^2 + \delta^2\|\nabla f(x_g^k)\|^2)$$
$$- p\gamma^2\eta^2(1 - 2p(1 + \gamma L))\|\nabla f(x_g^k)\|^2.$$

With $M \geq \sqrt{C_2 p^{-1}(1 + 2p/\beta)}$, we have

$$\mathbb{E}_k\left[\|x^{k+1} - x^*\|^2 + 2\gamma\eta^2(f(x_f^{k+1}) - f(x^*))\right]$$
$$\leq (1 - \beta/2)\|x^k - x^*\|^2 + (1 - p/\eta) \cdot 2\gamma\eta^2(f(x_f^k) - f(x^*))$$
$$+ p^2\eta^2\gamma^2\tau^2 B^{-2}(\sigma^2 + \delta^2\|\nabla f(x_g^k)\|^2)$$
$$+ 2p^2\eta^2\gamma^2\left(1 + \gamma L\right) \cdot \left(4C_1\tau B^{-1}\log_2 M + (4C_1 + 2)\tau^2 B^{-2}\right)(\sigma^2 + \delta^2\|\nabla f(x_g^k)\|^2)$$
$$- p\gamma^2\eta^2(1 - 2p(1 + \gamma L))\|\nabla f(x_g^k)\|^2$$
$$\leq (1 - \beta/2)\|x^k - x^*\|^2 + (1 - p/\eta) \cdot 2\gamma\eta^2(f(x_f^k) - f(x^*))$$
$$+ 8p^2\eta^2\gamma^2\left(1 + \gamma L\right) \cdot \left(C_1\tau B^{-1}\log_2 M + (C_1 + 1)\tau^2 B^{-2}\right)\sigma^2$$
$$- p\gamma^2\eta^2\left[1 - 2p\left(1 + \gamma L\right)\left(1 + 4\left[C_1\tau B^{-1}\log_2 M + (C_1 + 1)\tau^2 B^{-2}\right]\delta^2\right)\right]\|\nabla f(x_g^k)\|^2.$$

Since $p = \left[1 + 2\left(1 + \gamma L\right)\left(1 + 4\left[C_1\tau b^{-1} + (C_1 + 1)\tau^2 b^{-2}\right]\delta^2\right)\right]^{-1}$, $B = \lceil b\log_2 M\rceil$ and $M \geq 2$, we obtain

$$p = \left[1 + 2\left(1 + \gamma L\right)\left(1 + 4\left[C_1\tau b^{-1} + (C_1 + 1)\tau^2 b^{-2}\right]\delta^2\right)\right]^{-1}$$
$$\leq \left[1 + 2\left(1 + \gamma L\right)\left(1 + 4\left[C_1\tau B^{-1}\log_2 M + (C_1 + 1)\tau^2 B^{-2}\right]\delta^2\right)\right]^{-1},$$

and then,

$$\mathbb{E}_k\left[\|x^{k+1} - x^*\|^2 + 2\gamma\eta^2(f(x_f^{k+1}) - f(x^*))\right]$$
$$\leq (1 - \beta/2)\,\|x^k - x^*\|^2 + (1 - p/\eta)\cdot 2\gamma\eta^2(f(x_f^k) - f(x^*))$$
$$+ 8p^2\eta^2\gamma^2\left(1 + \gamma L\right)\cdot\left(C_1\tau B^{-1}\log_2 M + (C_1 + 1)\tau^2 B^{-2}\right)\sigma^2$$
$$\leq \max\left\{(1 - \beta/2), (1 - p/\eta)\right\}\left[\|x^k - x^*\|^2 + 2\gamma\eta^2(f(x_f^k) - f(x^*))\right]$$
$$+ 8p^2\eta^2\gamma^2\left(1 + \gamma L\right)\cdot\left(C_1\tau B^{-1}\log_2 M + (C_1 + 1)\tau^2 B^{-2}\right)\sigma^2.$$

Using that $p\eta\gamma = 3\beta/(2\mu)$, $\beta/2 = p/\eta$, $B = \lceil b\log_2 M\rceil$ and $\gamma \leq L^{-1}$, we have

$$\mathbb{E}_k\left[\|x^{k+1} - x^*\|^2 + 2\gamma\eta^2(f(x_f^{k+1}) - f(x^*))\right]$$
$$\leq (1 - \beta/2)\left[\|x^k - x^*\|^2 + 2\gamma\eta^2(f(x_f^k) - f(x^*))\right]$$
$$+ 36\beta^2\mu^{-2}\left(C_1\tau b^{-1} + (C_1 + 1)\tau^2 b^{-2}\right)\sigma^2. \tag{23}$$

Here we also took into account that $M \geq 2$. Finally, we perform the recursion and substitute $\beta = \sqrt{4p^2\mu\gamma/3}$

$$\mathbb{E}\left[\|x^N - x^*\|^2 + 2\gamma\eta^2(f(x_f^N) - f(x^*))\right]$$
$$\leq \left(1 - \sqrt{\frac{p^2\mu\gamma}{3}}\right)^N\left[\|x^0 - x^*\|^2 + 2\gamma\eta^2(f(x_f^0) - f(x^*))\right]$$
$$+ 72\beta\mu^{-2}\left(C_1\tau b^{-1} + (C_1 + 1)\tau^2 b^{-2}\right)\sigma^2$$
$$\leq \exp\left(-\sqrt{\frac{p^2\mu\gamma N^2}{3}}\right)\left[\|x^0 - x^*\|^2 + 2\gamma\eta^2(f(x_f^0) - f(x^*))\right]$$
$$+ \frac{144p\sqrt{\gamma}}{\sqrt{3}\mu^{3/2}}\left(C_1\sigma^2\tau b^{-1} + (C_1 + 1)\sigma^2\tau^2 b^{-2}\right).$$

Substituting of $\eta = \sqrt{\frac{3}{\mu\gamma}}$ concludes the proof. $\qquad\square$

### B.4 Results of Section 2.1 with decreasing stepsize

The first thing we need to change is to make the parameters of Algorithm 1 depend on the iteration number $k$: $\gamma, p, \beta, \eta, M, B \to \gamma_k, p_k, \beta_k, \eta_k, M_k, B_k$. For this new version of Algorithm 1 one can reprove Theorem 1.

**Theorem 7.** *Assume A 1 − A 4. Let problem (1) be solved by Algorithm 1. Then for any $b \in \mathbb{N}^*$, $\gamma_k \in (0; \frac{3}{4L}]$, and $\beta_k, \theta_k, \eta_k, p_k, M_k, B_k$ satisfying*

$$p_k \simeq (1 + (1 + \gamma_k L)[\delta^2\tau b^{-1} + \delta^2\tau^2 b^{-2}])^{-1}, \quad \beta_k \simeq \sqrt{p_k^2\mu\gamma_k}, \quad \eta_k \simeq \sqrt{\frac{1}{\mu\gamma_k}},$$

$$\theta_k \simeq \frac{p_k\eta_k^{-1} - 1}{\beta_k p_k\eta^{-1} - 1}, \quad M_k \simeq \max\{2; \sqrt{p_k^{-1}(1 + p_k/\beta_k)}\}, \quad B_k = \lceil b\log_2 M_k\rceil,$$

*it holds that*

$$\mathbb{E}\left[\|x^{k+1} - x^*\|^2 + \frac{6}{\mu}(f(x_f^{k+1}) - f(x^*))\right]$$
$$\leq \left(1 - \sqrt{\frac{p_k^2\mu\gamma_k}{3}}\right)\left[\|x^k - x^*\|^2 + \frac{6}{\mu}(f(x_f^k) - f(x^*))\right]$$
$$+ \frac{48p_k^2\gamma_k}{\mu}\left(C_1\tau b^{-1} + (C_1 + 1)\tau^2 b^{-2}\right)\sigma^2.$$

*Proof.* All steps of the proof remain the same with of Theorem 1 and we get (23):

$$\mathbb{E}\left[\|x^{k+1} - x^*\|^2 + 2\gamma_k\eta_k^2(f(x_f^{k+1}) - f(x^*))\right]$$

$$\leq (1 - \beta_k/2)\left[\|x^k - x^*\|^2 + 2\gamma_k\eta_k^2(f(x_f^k) - f(x^*))\right]$$
$$+ 36\beta_k^2\mu^{-2}\left(C_1\tau b^{-1} + (C_1 + 1)\tau^2 b^{-2}\right)\sigma^2.$$

By substituting $\beta_k = \sqrt{4p_k^2\mu\gamma_k/3}$ and $\eta_k = \sqrt{\frac{3}{\mu\gamma_k}}$, we finishes the proof. $\square$

Since $p_k = \left[1 + 2\left(1 + \gamma_k L\right)\left(1 + 4\left[C_1\tau b^{-1} + (C_1 + 1)\tau^2 b^{-2}\right]\delta^2\right)\right]^{-1}$ and $\gamma_k \in (0; \frac{3}{4L})$, then $p_k \in [p_l; p_u]$, where $p_l, p_u \sim (1 + (1 + \tau b^{-1} + \tau b^{-2})\delta^2)^{-1}$. It means that we can rewrite the results of the theorem as follows:

$$\mathbb{E}\left[\|x^{k+1} - x^*\|^2 + \frac{6}{\mu}(f(x_f^{k+1}) - f(x^*))\right]$$

$$\leq \left(1 - \sqrt{\frac{p_l^2\mu\gamma_k}{3}}\right)\left[\|x^k - x^*\|^2 + \frac{6}{\mu}(f(x_f^k) - f(x^*))\right]$$
$$+ \frac{48p_u^2\gamma_k}{\mu}\left(C_1\tau b^{-1} + (C_1 + 1)\tau^2 b^{-2}\right)\sigma^2.$$

With notation $r_k = \mathbb{E}\left[\|x^k - x^*\|^2 + \frac{6}{\mu}(f(x_f^k) - f(x^*))\right]$, $a = \sqrt{p_l^2\mu/3}$, $\omega_k = \sqrt{\gamma_k}$ and $C = \frac{48p_u^2}{\mu}\left(C_1\tau b^{-1} + (C_1 + 1)\tau^2 b^{-2}\right)\sigma^2$, one can rewrite the previous estimate:

$$r_{k+1} \leq (1 - a\omega_k)r_k + \omega_k^2 C,$$

where $0 < \omega_k \leq d = \sqrt{3/(4L)}$. For this kind of recursion, we can use the results of Lemma 3 of [88]. In particular, we can choose $\gamma_k$ as follows

$$\text{if } N \leq \frac{d}{a}, \qquad\qquad \gamma_k = \frac{1}{d},$$
$$\text{if } N > \frac{d}{a} \text{ and } k < \left\lceil\frac{N}{2}\right\rceil, \qquad\qquad \gamma_k = \frac{1}{d},$$
$$\text{if } N > \frac{d}{a} \text{ and } k \geq \left\lceil\frac{N}{2}\right\rceil, \qquad\qquad \gamma_k = \frac{2}{a(k + \frac{2d}{a} + \lceil\frac{N}{2}\rceil)},$$

and get

$$r_N = \mathcal{O}\left(\frac{dr_0}{a}\exp\left(-\frac{aN}{2d}\right) + \frac{C}{a^2 N}\right).$$

But the stepsize still depends on the horizon of iterations $N$. To fix it, we can apply the following restart procedure. We construct a sequence of the iteration number $N_t = 2^t$ for $t \geq 0$. For each restart $t$ we set the stepsize $\gamma(N_t)$ according to Lemma 3 of [88], run the algorithm for $N_t$ basic iterations. If we do not achieve the unknown horizon of the total iteration number $N$, then we use the obtained point as a warm-start for the next restart. For simplicity, we can also use the same starting $x^0$ point for all the restarts. Let us now assume that the algorithm made $N$ iterations. This means that it made at least $T = \lfloor\log_2(N + 1)\rfloor$ finished restarts. Since at the end of the last restart it made $N_T$ basic iterations with the stepsize $\gamma(N_T)$, we can guarantee that

$$r_{N_T} = \mathcal{O}\left(\frac{dr_0}{a}\exp\left(-\frac{aN_T}{2d}\right) + \frac{C}{a^2 N_T}\right).$$

One can note that $N_T \sim N$, then

$$r_{N_T} = \mathcal{O}\left(\frac{dr_0}{a}\exp\left(-\frac{aN}{2d}\right) + \frac{C}{a^2 N}\right).$$

This algorithm does not require to fix the number of basic steps $N$ in advance, but if we want to have $\varepsilon$-solution in terms of $r_N$, then we have the following estimate on the number of iterations:

$$N = \mathcal{O}\left(\frac{d}{a}\log\frac{1}{\varepsilon} + \frac{C}{a^2\varepsilon}\right) = \mathcal{O}\left(\left[1 + (1 + \tau b^{-1} + \tau^2 b^{-2})\delta^2\right]\sqrt{\frac{L}{\mu}}\log\frac{1}{\varepsilon} + \frac{\sigma^2}{\mu^2\varepsilon}\left(\tau b^{-1} + \tau^2 b^{-2}\right)\right).$$

To get the close to Corollary 1 results on the oracle complexity one need to take $b = \tau$ and note that now $B_k = b \log_2 M_k = b \log_2 M_k \sim b \log_2 N \sim b \log_2 \varepsilon^{-1}$. Finally, it gives additional logarithmic factor in the estimate for the oracle complexity. But this factor does not really change the bound and it means that we obtain the result of Corollary 1.

## B.5 Lower bounds proofs

**Proof of Theorem 2.**

We begin the proof with two lemmas, showing the lower bounds for deterministic and stochastic components of the error separately. Then we combine the two in Theorem 8 and complete the proof of Theorem 2. First, we consider the lower bound for the deterministic part of the error, and construct a problem with $\delta = 1$ and $\sigma = 0$.

**Lemma 7.** *There exists an instance of the optimization problem satisfying assumptions A 1 –A 4 with $\delta = 1$ and $\sigma = 0$, such that for any first-order gradient method it takes at least*

$$N = \Omega\left(\tau \sqrt{\frac{L}{\mu}} \log \frac{1}{\varepsilon}\right)$$

*oracle calls in order to achieve $\mathbb{E}[\|x^N - x^*\|^2] \leq \varepsilon$.*

*Proof.* Consider the optimization problem

$$f_1(x) = \frac{\mu(Q-1)}{4}\left(\frac{x^\top A x}{2} - e_1^\top x\right) + \frac{\mu}{2}\|x\|^2 \to \min_{x \in \mathbb{R}^d}, \tag{24}$$

where $x \in \mathbb{R}^d$, $\mu > 0, Q > 1$, dimension $d$ is even, $d = 2u$, $e_1 = (1, 0, \ldots, 0) \in \mathbb{R}^d$ is the first coordinate vector, and $A \in \mathbb{R}^{d \times d}$ is a symmetric nonnegative-definite matrix given by

$$A = \begin{pmatrix} 2 & -1 & 0 & 0 & \ldots & 0 & 0 \\ -1 & 2 & -1 & 0 & \ldots & 0 & 0 \\ 0 & -1 & 2 & -1 & \ldots & 0 & 0 \\ & & & \ldots & & & \\ 0 & 0 & 0 & 0 & \ldots & 2 & -1 \\ 0 & 0 & 0 & 0 & \ldots & -1 & \alpha, \end{pmatrix} \tag{25}$$

where $\alpha = \frac{\sqrt{Q}+3}{\sqrt{Q}+1}$. Straightforward calculations (see e.g. [59, Chapter 5.1.4] for more details) yield

$$0 \preceq A \preceq 4I, \nabla f_1(x) = \frac{\mu(Q-1)}{4} A x - e_1 + \mu x.$$

Thus the problem (24) is $L-$smooth with $L = \mu Q$ and $\mu$-strongly convex, i.e., the assumptions A 1 and A 2 are satisfied, and the corresponding condition number is equal to $L/\mu = Q$. For $\epsilon \in (0; 1/2)$ we now consider the two-state Markov transition matrix (or kernel)

$$\mathrm{P}_1 = \begin{pmatrix} 1 - \epsilon & \epsilon \\ \epsilon & 1 - \epsilon \end{pmatrix} \tag{26}$$

and denote by $(Z_i)_{i=1}^\infty$ the corresponding Markov Chain, $Z_i \in \{-1, 1\}$. It is easy to see that the Markov kernel P is uniformly geometrically ergodic and satisfies A 4 with $\tau \leq \epsilon^{-1} \log 4$. It is easy to check that the corresponding invariant distribution is $\pi = (1/2, 1/2)$. For $Z \in \{-1, 1\}$ we now consider the noise matrix

$$W(Z) = 2 \operatorname{diag}\{\mathbb{1}_{\{Z=1\}}, \mathbb{1}_{\{Z=-1\}}, \mathbb{1}_{\{Z=1\}}, \ldots, \mathbb{1}_{\{Z=-1\}}\} \in \mathbb{R}^{d \times d}.$$

Now for $x \in \mathbb{R}^d$ and $Z \in \{-1, 1\}$ we define the stochastic gradient oracle as

$$\nabla F_1(x, Z) = W(Z)\nabla f(x). \tag{27}$$

It is easy to check that $\mathbb{E}_\pi[W(Z)] = I$, and the direct calculations imply $\|\nabla F_1(x, Z) - \nabla f_1(x)\| \leq \|\nabla f_1(x)\|$, that is, the assumption A 4 holds with $\delta = 1$ and $\sigma = 0$. Following [76], [59, Chapter 5.1.4], the solution to the minimization problem (24) is given by

$$x^* = (q^1, \ldots, q^d) \in \mathbb{R}^d, q = \frac{\sqrt{Q}-1}{\sqrt{Q}+1}. \tag{28}$$

Suppose that we start from $Z_1 = 1$ and initial point $x_0 = 0 \in \mathbb{R}^d$. Then after 1 oracle call we observe the 1-st coordinate of $x$. At the same time, the second component can not be computed until the time moment $T_2 = \inf\{i \in \mathbb{N} : Z_i = -1\}$. Similarly, the next computation of the 3-rd component of the solution requires the chain to go back to state 1 and can not happen earlier then $T_3 = \inf\{i \geq \tau_2 : Z_i = 1\}$. Thus, after $k$ iterations of any first-order method, the respective MSE is lower bounded by

$$\mathbb{E}_\pi[\|x^k - x^*\|^2] \geq \mathbb{E}_\pi\left[\mathbb{1}_{\{Z_1=1\}} \sum_{i=N_k}^{d} q^{2i}\right] = \frac{1}{2}\mathbb{E}_{\delta_1}\left[\frac{q^{2N_k} - q^{2d}}{1 - q^2}\right].$$

In the formula above we denoted by $N_k$ the number of state changes in the sequence $(Z_i)_{i=1}^{k}$. Using Jensen's inequality and the explicit construction of the Markov kernel $P_1$ in (26), we deduce that

$$\mathbb{E}_\pi[\|x^k - x^*\|^2] \geq \frac{1}{2}\frac{q^{2\mathbb{E}_{\delta_1}[N_k]} - q^{2d}}{1 - q^2} = \frac{1}{2}\frac{q^{2(k-1)\epsilon} - q^{2d}}{1 - q^2} \geq (1/2)(1 - q^2)^{-1}q^{(2/\log 4)k/\tau} =$$

$$= (1/2)(1 - q^2)^{-1}\left(1 - \frac{2}{\sqrt{Q} + 1}\right)^{(2/\log 4)k/\tau}$$

$$\geq (1/2)(1 - q^2)^{-1}\exp\left(-\frac{8k}{(\sqrt{Q} + 1)\tau \log 4}\right),$$

provided that $d$ is large enough. In the last inequality we also used that $1 - x \geq e^{-2x}$ for $x \in [0; 1/2]$. Hence, taking into account that $Q$ is the condition number of the problem (24), we get the desired lower bound. $\square$

Now we consider an instance of the problem with $\delta = 0$, arbitrary $\sigma \geq 0$, and construct the respective lower bound for the stochastic part of the error.

**Lemma 8.** *There exists an instance of the optimization problem satisfying assumptions A 1 –A 4 with $\delta = 0$ and arbitrary $\sigma \geq 0$, such that for any first-order gradient method it takes at least*

$$N = \Omega\left(\frac{\tau\sigma^2}{\mu^2\varepsilon}\right)$$

*oracle calls in order to achieve $\mathbb{E}[\|x^N - x^*\|^2] \leq \varepsilon$.*

*Proof.* Our proof is based on a simple 1-dimensional optimization problem and Le Cam's lemma [1, Theorem 8], see also [105]. Consider the following minimization problem

$$f_2(x) = \frac{\mu}{2}(x - x^*)^2 \mapsto \min_{x \in \mathbb{R}}. \tag{29}$$

Obviously this problem satisfies A 2 with strong convexity constant $\mu$ and A 1 with $L = \mu$. Consider the noisy gradient oracle

$$\nabla F_2(x, Y) = \mu(x - x^*) + \frac{\sigma}{2}Y, \tag{30}$$

where $Y$ is a noise variable taking values $Y \in \{-1, 1\}$. For now we do not specify the distribution of $Y$, yet we easily note that for any distribution $\pi$ on $\{-1, 1\}$ we have

$$\|\nabla F_2(x, Y) - \mathbb{E}_\pi\nabla F_2(x, Y)\|^2 \leq \sigma^2.$$

Consider the sequence of noise variables $(Y_i)_{i=1}^{n}$ with the joint distribution to the specified later, and any sequence of design points $(x_i)_{i=1}^{n}$, where the resulting gradients are evaluated. At this point the statistician observes the gradients

$$\left(\mu(x_i - x^*) + \frac{\sigma}{2}Y_i\right), i = 1\ldots, n,$$

and, since $x_i$ and $\mu$ are known, this is equivalent to observing

$$x^* - \frac{\sigma}{2\mu}Y_i, i = 1\ldots, n.$$

Now we aim to construct to "almost indistinguishable" models for the noise variables $Y_i$. Namely, we consider the parametric family of Markov kernels

$$P_\varphi = \begin{pmatrix} 1 - \epsilon & \epsilon \\ \epsilon + \varphi & 1 - \epsilon - \varphi \end{pmatrix}, \tag{31}$$

where the parameters $\varphi, \epsilon \in (0; 1/4)$, and $\varphi \in [0; \alpha]$, and the parameter $\alpha$ will be set depending on $\epsilon$ and $n$ later. It is easy to check that the invariant distribution of the Markov kernel $P_\varphi$ is given by

$$\pi^\varphi = \left( \frac{\epsilon + \varphi}{2\epsilon + \varphi}, \frac{\epsilon}{2\epsilon + \varphi} \right).$$

Now we consider the setting of Le Cam's lemma [1, Theorem 8]. Namely, for a fixed sample size $n$ we consider the family of Markov kernels $(P_\varphi)_{\varphi \in [0;\alpha]}$, and family of corresponding joint $n-$step distributions under stationarity, that is, $\pi^\varphi P_\varphi^{\otimes n}$. The reader not familiar with the respective notation could find it, in particular, in [22, Chapter 1]. As a parameter of interest we consider the expectation

$$\theta(\varphi) := \theta(\pi^\varphi P_\varphi^{\otimes n}) := \mathbb{E}_{\pi^\varphi}\left[x^* - \frac{\sigma}{2\mu} Z_i\right] = x^* - \frac{\sigma\varphi}{2\mu(2\epsilon + \varphi)}. \tag{32}$$

Now we consider the 2 representatives of the above class, that is, the $n$-step distributions corresponding the parameters $\varphi = 0$ and $\varphi = \alpha$. Then the direct application of Le Cam's lemma [1, Theorem 8] yields

$$\inf_{\widehat{\theta}} \sup_{\varphi \in [0;\alpha]} \mathbb{E}^{1/2}_{\pi^\varphi P_\varphi^{\otimes n}}[|\widehat{\theta} - \theta(\varphi)|^2] \geq \frac{1}{2}|\theta(0) - \theta(\alpha)|(1 - \|\pi^0 P_0^{\otimes n} - \pi^\alpha P_\alpha^{\otimes n}\|_{TV}), \tag{33}$$

where $\widehat{\theta} = \widehat{\theta}(Y_1, \ldots, Y_n)$ is any measurable function. Thus, taking square and using the definition of $\theta(\varphi)$ in (32), we obtain that

$$\inf_{\widehat{\theta}} \sup_{\varphi \in [0;\alpha]} \mathbb{E}_\pi[|\theta - \theta(\varphi)|^2] \geq \frac{\sigma^2 \alpha^2}{16\mu^2(2\epsilon + \alpha)^2}(1 - \|\pi^0 P_0^{\otimes n} - \pi^\alpha P_\alpha^{\otimes n}\|_{TV}). \tag{34}$$

Now we set $\alpha = \sqrt{\frac{\epsilon}{n}}$ and apply the statement of Lemma 9 with this choice of $\alpha$. Note that we impose at this point the regularity condition $n \geq \epsilon^{-1}$ in order to have $\alpha \leq \epsilon$. Thus we get

$$\inf_{\widehat{\theta}} \sup_{\varphi \in [0;\sqrt{\frac{\epsilon}{n}}]} \mathbb{E}_\pi[|\widehat{\theta} - \theta(\varphi)|^2] \geq \frac{\sigma^2 \epsilon}{32\mu^2 n(2\epsilon + \sqrt{\frac{\epsilon}{n}})^2} \geq \frac{\sigma^2}{288\mu^2 n\epsilon},$$

and the statement follows by noticing that the corresponding mixing time $\tau \leq c\epsilon^{-1}$ for some $c > 0$ (see e.g. [71, Proposition 1]). $\square$

**Lemma 9.** *Consider the family of Markov kernels*

$$P_\varphi = \begin{pmatrix} 1 - \epsilon & \epsilon \\ \epsilon + \varphi & 1 - \epsilon - \varphi \end{pmatrix}$$

*and the corresponding invariant distributions* $\pi^\varphi = \left( \frac{\epsilon+\varphi}{2\epsilon+\varphi}, \frac{\epsilon}{2\epsilon+\varphi} \right)$ *for* $\varphi \in \{0, \alpha\}$. *Then it holds that*

$$\|\pi^0 P_0^{\otimes n} - \pi^\alpha P_\alpha^{\otimes n}\|_{TV} \leq \frac{1}{2}\sqrt{\frac{n\alpha^2}{\epsilon}}.$$

*Proof.* Note first that an application of Pinsker's inequality yields

$$\|\pi^0 P_0^{\otimes n} - \pi^\alpha P_\alpha^{\otimes n}\|_{TV} \leq \sqrt{(1/2)\,\mathrm{KL}(\pi^0 P_0^{\otimes n}\|\pi^\alpha P_\alpha^{\otimes n})}.$$

Using the chain rule for KL-divergence, we get

$$\mathrm{KL}(\pi^0 P_0^{\otimes n}\|\pi^\alpha P_\alpha^{\otimes n}) = \mathrm{KL}(\pi^0\|\pi^\alpha) + \sum_{i=1}^{n-1} \sum_{y \in \{-1,1\}} P_{\pi^0 P_0^{\otimes n}}(Y_i = y)\,\mathrm{KL}(P_0(\cdot|y)\|P_\alpha(\cdot|y)). \tag{35}$$

In the notation above for $y \in \{-1, 1\}$ we have set $\mathrm{KL}(\mathrm{P}_0(\cdot|y)||\mathrm{P}_\alpha(\cdot|y))$ for the $1-$step conditional distribution

$$\mathrm{KL}(\mathrm{P}_0(\cdot|y)||\mathrm{P}_\alpha(\cdot|y)) = \sum_{x \in \{-1,1\}} \mathrm{P}_0(x|y) \log \frac{\mathrm{P}_0(x|y)}{\mathrm{P}_\alpha(x|y)} \,.$$

Now an application of reversed Pinsker's inequality together with $\alpha \leq \epsilon$ yields that

$$\mathrm{KL}(\mathrm{P}_0(\cdot|y)||\mathrm{P}_\alpha(\cdot|y)) \leq \frac{\alpha^2}{2\epsilon} \,,$$

and the bound (35) implies that

$$\mathrm{KL}(\pi^0 \mathrm{P}_0^{\otimes n}||\pi^\alpha \mathrm{P}_\alpha^{\otimes n}) \leq \frac{n\alpha^2}{2\epsilon} \,.$$

Combining the bounds above yields the statement. $\qquad\square$

Now we are ready to combine the bounds above and prove Theorem 2.

**Theorem 8** (Theorem 2). *There exists an instance of the optimization problem satisfying assumptions A 1 –A 4 with $\delta = 1$ and arbitrary $\sigma \geq 0, L, \mu > 0, \tau \in \mathbb{N}^*$, such that for any first-order gradient method it takes at least*

$$N = \Omega\big(\tau\sqrt{\tfrac{L}{\mu}} \log \tfrac{1}{\varepsilon} + \tfrac{\tau\sigma^2}{\mu^2\varepsilon}\big)$$

*oracle calls in order to achieve $\mathbb{E}[\|x^N - x^*\|^2] \leq \varepsilon$.*

*Proof.* We split the original problem into two parts. Indeed, for any $d \in \mathbb{N}^*$ we consider $x = (x_{det}, x_{stoch}) \in \mathbb{R}^{d+1}$, where $x_{det} \in \mathbb{R}^d$ and $x_{stoch} \in \mathbb{R}$. Now we consider the minimization problem

$$f(x) = f(x_{det}, x_{stoch}) = f_1(x_{det}) + f_2(x_{stoch}) \to \min_{x \in \mathbb{R}^{d+1}} \,, \tag{36}$$

where the functions $f_1 : \mathbb{R}^d \to \mathbb{R}$ and $f_2 : \mathbb{R} \to \mathbb{R}$ are defined in (24) and (29), respectively. We fix the respective parameters $\mu, Q$, and $\sigma$. Applying Lemma 7 and Lemma 8, we get that the respective problem (36) is $L$-smooth and $\mu$-strongly convex with $L = \mu Q$ and parameter $Q > 1$ defined in (24). For $Z, Y \in \{-1, 1\}$ we define the stochastic gradient oracle as

$$\nabla F(x, Z, Y) = (\nabla F_1(x_{det}, Z), \nabla F_2(x_{stoch}, Y)) \in \mathbb{R}^{d+1} \,.$$

The oracles $\nabla F_1(x_{det}, Z)$ and $\nabla F_2(x_{stoch}, Y)$ are defined in (27) and (30), respectively. Lemma 7 and Lemma 8 imply that A 4 holds with $\delta = 1$ and $\sigma > 0$ defined in (30). Consider now the Markov chains $(Z_i)_{i=1}^\infty$ with the transition kernel $\mathrm{P}_1$ defined in (26) and $(Y_i)_{i=1}^\infty$ with the transition kernel $\mathrm{P}_\varphi$ of the form (31). As in the proof of Lemma 8, we take $\varphi \in [0; \sqrt{\epsilon/n}]$ and assume that $n \geq \epsilon^{-1}$. Consider the joint process $(X_i, Y_i)_{i=1}^\infty$ of independently evolving Markov chains $(Z_i)_{i=1}^\infty$ and $(Y_i)_{i=1}^\infty$. It is easy to see that such a process is a Markov chain on $\{-1, 1\}^2$ with the transition kernel

$$\mathrm{P} = \mathrm{P}_1 \otimes \mathrm{P}_\varphi \,,$$

where $\otimes$ stands for the Kronecker's product. In is clear that $\mathrm{P}$ is irreducible and aperiodic, hence the assumption A 3 holds. Note that both $\mathrm{P}_1$ and $\mathrm{P}_\varphi$ are reversible (see e.g. [81][Section 3.1] for the respective definitions). Thus their Kronecker's product is also reversible, with the spectrum given by the pairwise products of eigenvalues of $\mathrm{P}_1$ and $\mathrm{P}_\varphi$. Hence, with the direct calculations, we compute the eigenvalues of $\mathrm{P}$: $\{1, 1 - 2\epsilon - \varphi, 1 - 2\epsilon, (1 - 2\epsilon)(1 - 2\epsilon - \varphi)\}$. Thus the corresponding spectral gap $\gamma = 2\epsilon$, and the mixing time $\tau$ of $\mathrm{P}$ is bounded by

$$\frac{1}{2\epsilon(1 + 1/\log 2)} \leq \tau \leq \frac{2\log 2 + \log 6}{4\epsilon} \,,$$

see [81][Proposition 3.3]. Hence, the mixing time of the corresponding joint chain scales as $\epsilon^{-1}$, as for $(Z_i)_{i=1}^\infty$ and $(Y_i)_{i=1}^\infty$ separately. On the $k$-th step of the stochastic gradient computations we rely on the stochastic gradient

$$\nabla F(x_k, Z_k, Y_k) \,,$$

computed using the pair $(Z_k, Y_k)$. To complete the proof it remains to apply the complexity results of Lemma 7 and Lemma 8 to the parts $x_{det}$ and $x_{stoch}$, respectively. $\qquad\square$

**Proposition 3** (Proposition 1). *There exists an instance of the optimization problem satisfying assumptions A 1 –A 4 with arbitrary $L, \mu > 0, \tau \in \mathbb{N}^*$, $\delta = \frac{L}{\mu}$, and $\sigma = 0$, such that for any first-order gradient method it takes at least*

$$N = \Omega\left(\tau \frac{L}{\mu} \log \frac{1}{\varepsilon}\right)$$

*gradient calls in order to achieve $\mathbb{E}[\|x^N - x^*\|^2] \leq \varepsilon$.*

*Proof.* In this part we closely follow the setting of [71]. We consider the setting of linear regression:

$$f(x) = \frac{1}{2}\mathbb{E}_{(\varphi,Y)\sim\mathcal{D}}\left[|Y - \varphi^\top x|^2\right] \to \min_x, \tag{37}$$

where $\varphi \in \mathbb{R}^d$ is a (random) feature vector, $Y \in \mathbb{R}$ is a (random) regressor, with the joint distribution $(\varphi, Y) \sim \mathcal{D}$, and $x \in \mathbb{R}^d$ is the optimized parameter. We consider the so-called realizable case, that is, we assume that

$$Y = \varphi^\top x^*$$

for some vector $x^* \in \mathbb{R}^d$. In this scenario the problem (37) reduces to

$$f(x) = \frac{1}{2}(x - x^*)^\top \Sigma^2 (x - x^*) \to \min_x,$$

where we have denoted $\Sigma^2 = \mathbb{E}_\pi[\varphi\varphi^\top]$. This means that the exact gradient is given by $\nabla f(x) = \Sigma^2(x - x^*)$. Now we consider the stochastic setting of the online regression with sequentially observed data points $(\varphi_i, Y_i)_{i=1}^N$ with $Y_i = \varphi_i^\top x^*$. In this case the $i$-th realization of stochastic gradient at point $x \in \mathbb{R}^d$ is given by

$$\nabla F(x, \varphi_i, Y_i) = \varphi_i(\varphi_i^\top x - Y_i) = \varphi_i\varphi_i^\top (x - x^*).$$

Hence, with a simple algebra we get

$$\|\nabla F(x, \varphi_i, Y_i) - \nabla f(x)\| = \|(\Sigma^2 - \varphi_i\varphi_i^\top)(x - x^*)\| = \|(\mathrm{I} - \varphi_i\varphi_i^\top \Sigma^{-2})\nabla f(x)\|,$$

where we have used the fact that $x - x^* = (\Sigma^2)^{-1}\nabla f(x)$ and used additional notation $\Sigma^{-2} := (\Sigma^2)^{-1}$. Fix now the condition number $Q > 1$, parameter $\epsilon \in (0; 1/4)$ and consider the Markov kernel

$$\mathrm{P} = \begin{pmatrix} 1 - \frac{\epsilon}{Q-1} & \frac{\epsilon}{Q-1} \\ \epsilon & 1 - \epsilon \end{pmatrix}$$

and the corresponding canonical chain $(Z_i)_{i=1}^N$. The invariant distribution of P is given by $\pi = (1 - 1/Q, 1/Q)$, and the corresponding mixing time $\tau$ is bounded by

$$\tau \leq \frac{(Q-1)\log 4}{Q\epsilon},$$

see e.g. [71, Proposition 1]. We let $\varphi = \varphi(Z)$, and w.l.o.g. we can assume that $Z \in \{-1, 1\}$. Consider

$$\varphi(1) = (1, 0), \quad \varphi(-1) = (0, 1).$$

The design matrix $\Sigma^2$ is given by

$$\Sigma^2 = \mathbb{E}_\pi[\varphi(Z_i)\varphi(Z_i)^\top] = \begin{pmatrix} 1 - 1/Q & 0 \\ 0 & 1/Q \end{pmatrix},$$

which implies that A 1 and A 2 are satisfied with $\mu = 1/Q$ and $L = 1 - 1/Q$. Then the direct calculations yield

$$\|\nabla F(x, \varphi(Z_i), Y_i) - \nabla f(x)\| \leq (Q-1)\|\nabla f(x)\|,$$

and the assumption A 4 is satisfied with $\delta = Q - 1$. Then the direct application of lower bound [71] implies the lower bound

$$\mathbb{E}_\pi[\|x^k - x^*\|^2] \geq \exp\left(-\frac{ck}{Q\tau}\right)$$

after $k$ iterations of any first-order method with Markovian sampling oracle defined above. Here $c > 0$ is some absolute positive constant not dependeing upon $\tau$ and $Q$. This means that the instance-dependent increase of $\delta$ yields to inevitably slower convergence rates. $\square$

**Proposition 4** (Proposition 2). *There exists an instance of the optimization problem satisfying assumptions A 1 –A 4 with with arbitrary $L, \mu > 0, \tau \in \mathbb{N}^*$, $\sigma = 1, \delta = 0$, such that for any first-order gradient method it takes at least*

$$N = \Omega \left( \left( \tau + \sqrt{\tfrac{L}{\mu}} \right) \log\{\tfrac{1}{\varepsilon}\} \right)$$

*oracle calls in order to achieve $\mathbb{E}[\|x^N - x^*\|^2] \leq \varepsilon$.*

*Proof.* Let us consider the same minimization problem (24) as in the proof of Theorem 2. Recall that the true gradient in this setting is given by

$$\nabla f(x) = \frac{\mu(Q-1)}{4} Ax - e_1 + \mu x \,.$$

Hence the problem (24) is $L-$smooth with $L = \mu Q$ and $\mu$-strongly convex, that is, assumptions A 1 and A 2 are satisfied, and the corresponding condition number equals $L/\mu = Q$. Now for $\epsilon \in (0; 1/2)$ we consider the discrete-state space Markov kernel

$$\mathrm{P} = \begin{pmatrix} 1 - \epsilon & \epsilon \\ \epsilon & 1 - \epsilon \end{pmatrix} \tag{38}$$

and the corresponding Markov Chain $(Z_i)_{i=1}^\infty$. It is easy to see that the Markov kernel $\mathrm{P}$ is uniformly geometrically ergodic and satisfies A 4 with $\tau \leq \epsilon^{-1} \log 4$. Each $Z_i$ takes 2 different values, and w.l.o.g. we can assume that $Z_i \in \{-1, 1\}$. It is easy to check that the corresponding invariant distribution is $\pi = (1/2, 1/2)$. For $Z \in \{-1, 1\}$ we now consider the noisy oracle

$$\nabla F(x, Z) = \frac{\mu(Q-1)}{4} Ax - (1 + \mathbb{1}_{\{Z=-1\}} - \mathbb{1}_{\{Z=1\}})e_1 + \mu x \,.$$

It is easy to check that $\mathbb{E}_\pi[\nabla F(x, Z)] = \nabla f(x)$, and the direct calculations imply $\|\nabla F(x, Z) - \nabla f(x)\| \leq 1$, that is, the assumption A 4 holds with $\delta = 0$ and $\sigma = 1$. Suppose that we start from $Z_1 = 1$ and initial point $x_0 = 0 \in \mathbb{R}^d$. Then we observe $\nabla F(x, Z) = 0 \in \mathbb{R}^d$ unless the time moment $T_2 = \inf\{i \in \mathbb{N} : Z_i = -1\}$. Thus, after $k$ iterations of any first-order method, the respective MSE is lower bounded by

$$\mathbb{E}_\pi[\|x^k - x^*\|^2] \geq \mathbb{E}_\pi \left[ \mathbb{1}_{\{Z_1=1\}} \sum_{i=k}^d q^{2i} \right] + \mathbb{E}_\pi \left[ \mathbb{1}_{\{Z_1=1, T_2 \geq k\}} (1 - q^{2d}) \right]$$

$$\geq \frac{1}{2}(1 - q^2)^{-1}(q^{2k} - q^{2d}) + \frac{1}{2}(1 - q^2)^{-1} \mathbb{P}_{\delta_1}(T_2 \geq k)$$

$$= \frac{1}{2}(1 - q^2)^{-1}(q^{2k} - q^{2d}) + \frac{1}{2}(1 - q^2)^{-1}(1 - \epsilon)^{k-1} \,.$$

Hence, with the defition of $q$ in (28), we get from the previous bound that

$$\mathbb{E}_\pi[\|x^k - x^*\|^2] \geq \frac{1}{2}(1 - q^2)^{-1} \left[ \exp\left( -\frac{4k}{(\sqrt{Q}+1)} \right) - q^{2d} \right] + \frac{1}{2}(1 - q^2)^{-1} \exp\left( -\frac{2k}{\tau \log 4} \right) \,,$$

where in the last inequality we also used that $1 - x \geq e^{-2x}$ for $x \in [0; 1/2]$. Now the statement follows from the definition of $Q = L/\mu$. □

### B.6 Proof of Theorem 3

**Theorem 9** (Theorem 3). *Assume A 1, A 3, A 4. Let problem* (1) *be solved by Algorithm 2. Let $f^*$ be a global (maybe not unique) minimum of $f$. Then for any $b \in \mathbb{N}^*$, and $\gamma, M$ satisfying*

$$\gamma \leq \left[ 4L \left( 1 + 4 \left[ C_1 \tau b^{-1} + (C_1 + 1)\tau^2 b^{-2} \right] \delta^2 \right) \right]^{-1},$$

$$M = \max\{2; \sqrt{C_2 \gamma^{-1} L^{-1}}\}, \quad B = \lceil b \log_2 M \rceil,$$

*it holds that*

$$\mathbb{E} \left[ \frac{1}{N} \sum_{k=0}^{N-1} \|\nabla f(x^k)\|^2 \right] \lesssim \frac{f(x^0) - f^*}{\gamma N} + L\gamma \cdot \left[ \sigma^2 \tau b^{-1} + \sigma^2 \tau^2 b^{-2} \right] \,.$$

*Proof.* We start from A 1 (in the form (42) with $x = x^{k+1}$ and $y = x^k$) and line 6 of Algorithm 2:

$$f(x^{k+1}) \leq f(x^k) + \langle \nabla f(x^k), x^{k+1} - x^k \rangle + \frac{L}{2}\|x^{k+1} - x^k\|^2$$

$$\leq f(x^k) - \gamma\langle \nabla f(x^k), g^k \rangle + \frac{\gamma^2 L}{2}\|g^k\|^2$$

$$= f(x^k) - \gamma\langle \nabla f(x^k), \nabla f(x^k) \rangle - \gamma\langle \nabla f(x^k), \mathbb{E}_k[g^k] - \nabla f(x^k) \rangle$$

$$- \gamma\langle \nabla f(x^k), g^k - \mathbb{E}_k[g^k] \rangle + \frac{L\gamma^2}{2}\mathbb{E}_k[\|g^k\|^2]$$

Subtracting $f^*$ from both sides, using Cauchy Schwartz inequality (43) and taking the conditional expectation, we get

$$\mathbb{E}_k[f(x^{k+1}) - f^*] \leq f(x^k) - f^* - \gamma\|\nabla f(x^k)\|^2 + \frac{\gamma}{2}\|\nabla f(x^k)\|^2$$

$$+ \frac{\gamma}{2}\|\mathbb{E}_k[g^k] - \nabla f(x_g^k)\|^2 + \frac{L\gamma^2}{2}\mathbb{E}_k[\|g^k\|^2]$$

$$= f(x^k) - f^* - \frac{\gamma}{2}\|\nabla f(x^k)\|^2 + \frac{\gamma}{2}\|\mathbb{E}_k[g^k] - \nabla f(x^k)\|^2 + \frac{L\gamma^2}{2}\mathbb{E}_k[\|g^k\|^2].$$

Reapplying Cauchy Schwartz inequality (44) one more time, we have

$$\mathbb{E}_k[f(x^{k+1}) - f^*] \leq f(x^k) - f^* - \frac{\gamma}{2}(1 - 2\gamma L)\|\nabla f(x^k)\|^2$$

$$+ \frac{\gamma}{2}\|\mathbb{E}_k[g^k] - \nabla f(x^k)\|^2 + L\gamma^2\mathbb{E}_k[\|g^k - \nabla f(x^k)\|^2].$$

Lemma 4 with $x_g^k$ replaced by $x^k$ gives

$$\mathbb{E}_k[f(x^{k+1}) - f^*] \leq f(x^k) - f^* - \frac{\gamma}{2}(1 - 2\gamma L)\|\nabla f(x^k)\|^2$$

$$+ \frac{\gamma}{2} \cdot C_2\tau^2 M^{-2}B^{-2}(\sigma^2 + \delta^2\|\nabla f(x^k)\|^2)$$

$$+ L\gamma^2 \cdot \left(4C_1\tau B^{-1}\log_2 M + (4C_1 + 2)\tau^2 B^{-2}\right)(\sigma^2 + \delta^2\|\nabla f(x^k)\|^2).$$

With $M \geq \sqrt{C_2\gamma^{-1}L^{-1}}$, we have

$$\mathbb{E}_k[f(x^{k+1}) - f^*] \leq f(x^k) - f^* - \frac{\gamma}{2}(1 - 2\gamma L)\|\nabla f(x^k)\|^2$$

$$+ \frac{L\gamma^2}{2} \cdot \tau^2 B^{-2}(\sigma^2 + \delta^2\|\nabla f(x^k)\|^2)$$

$$+ L\gamma^2 \cdot \left(4C_1\tau B^{-1}\log_2 M + (4C_1 + 2)\tau^2 B^{-2}\right)(\sigma^2 + \delta^2\|\nabla f(x^k)\|^2)$$

$$\leq f(x^k) - f^*$$

$$- \frac{\gamma}{2}\left[1 - 2\gamma L\left(1 + 4\left[C_1\tau B^{-1}\log_2 M + (C_1 + 1)\tau^2 B^{-2}\right]\delta^2\right)\right]\|\nabla f(x^k)\|^2$$

$$+ 4L\gamma^2 \cdot \left(C_1\tau B^{-1}\log_2 M + (C_1 + 1)\tau^2 B^{-2}\right)\sigma^2.$$

Since $\gamma \leq \left[4L\left(1 + 4\left[C_1\tau b^{-1} + (C_1 + 1)\tau^2 b^{-2}\right]\delta^2\right)\right]^{-1}$, $B = \lceil b\log_2 M \rceil$ and $M \geq 2$, one can obtain

$$\gamma \leq \left[4L\left(1 + 4\left[C_1\tau b^{-1} + (C_1 + 1)\tau^2 b^{-2}\right]\delta^2\right)\right]^{-1}$$

$$\leq \left[4L\left(1 + 4\left[C_1\tau B^{-1}\log_2 M + (C_1 + 1)\tau^2 B^{-2}\right]\delta^2\right)\right]^{-1},$$

and then,

$$\mathbb{E}_k[f(x^{k+1}) - f^*] \leq f(x^k) - f^* - \frac{\gamma}{4}\|\nabla f(x^k)\|^2$$

$$+ 4L\gamma^2 \cdot \left(C_1\tau B^{-1}\log_2 M + (C_1 + 1)\tau^2 B^{-2}\right)\sigma^2. \quad (39)$$

By doing a small rearrangements, summing over all $k$ from 0 to $N - 1$, averaging over $N$ iterations, taking the full expectation of both sides, we get

$$\mathbb{E}\left[\frac{1}{N}\sum_{k=0}^{N-1}\|\nabla f(x^k)\|^2\right] \leq \frac{4(f(x^0) - f^*)}{\gamma N} + 16L\gamma \cdot \left[C_1\sigma^2\tau B^{-1}\log_2 M + (C_2 + 1)\sigma^2\tau^2 B^{-2}\right].$$

Substituting $B = \lceil b\log_2 M \rceil$ and using $M \geq 2$ finish the proof. $\qquad\square$

## B.7 Result for Polyak-Loiasyewitch condition

**A 9.** *The function $f$ satisfies PL condition on $\mathbb{R}^d$ with $\mu > 0$, i.e. the following inequality holds for all $x \in \mathbb{R}^d$:*

$$\|\nabla f(x)\| \geq 2\mu(f(x) - f^*),$$

*where $f^*$ is a global (potentially not unique) minimum of $f$.*

**Corollary 5.** *Under the conditions of Theorem 3 and A 9, if we choose $b = \tau$ and $\gamma$ given by*

$$\gamma \simeq \min\left\{\frac{1}{(1+\delta^2)L}; \frac{1}{\mu N}\ln\left(\max\left\{2; \frac{\mu^2 N(f(x^0) - f^*)}{L\sigma^2}\right\}\right)\right\}, \tag{40}$$

*then to achieve $\varepsilon$-solution (in terms of $\mathbb{E}[f(x) - f^*] \lesssim \varepsilon$) we need*

$$\tilde{\mathcal{O}}\left(\tau \cdot \left[(1+\delta^2)\frac{L}{\mu}\log\frac{1}{\varepsilon} + \frac{L\sigma^2}{\mu^2\varepsilon}\right]\right) \quad \text{oracle calls.}$$

*Proof.* We start from (39) and apply A 9.

$$\begin{aligned}
\mathbb{E}_k[f(x^{k+1}) - f^*] \leq &(1 - \mu\gamma/2)(f(x^k) - f^*)\\
&+ 4L\gamma^2 \cdot \left(C_1\tau B^{-1}\log_2 M + (C_1+1)\tau^2 B^{-2}\right)\sigma^2.
\end{aligned}$$

Next, we perform the recursion

$$\begin{aligned}
\mathbb{E}[f(x^N) - f^*] \leq &(1 - \mu\gamma/2)^N(f(x^k) - f^*)\\
&+ 8L\mu^{-1}\gamma \cdot \left(C_1\tau B^{-1}\log_2 M + (C_1+1)\tau^2 B^{-2}\right)\sigma^2\\
\leq &\exp(-\mu\gamma N/2)(f(x^0) - f^*)\\
&+ 8L\mu^{-1}\gamma \cdot \left(C_1\tau B^{-1}\log_2 M + (C_1+1)\tau^2 B^{-2}\right)\sigma^2.
\end{aligned}$$

It remains to substitute $\gamma$ from (40), $B = \lceil b\log_2 M\rceil$ and $b = \tau$. $\qquad\square$

## B.8 Proof of Theorem 4

We preface the proof by technical Lemma.

**Lemma 10.** *Let $r$ be $\mu_r$-strongly convex and $x^+ = \text{prox}_{\gamma r}(x)$. Then for all $u \in \mathcal{X}$ the following iniquity hold:*

$$\langle x^+ - x, u - x^+\rangle \geq \gamma\left(r(x^+) - r(u) + \frac{\mu_r}{2}\|x^+ - u\|^2\right).$$

*Proof.* The optimality condition for $x^+ = \text{prox}_{\gamma r}(x) = \arg\min_{y\in\mathcal{X}}(\gamma r(y) + \frac{1}{2}\|x^+ - y\|^2)$ gives that $(x - x^+) \in \partial r(x^+)$. Therefore, using strong convexity (see A 6) for $r'(x^+) = (x - x^+) \in \partial r(x^+)$, we get

$$\gamma(r(u) - r(x^+)) \geq \langle x - x^+, u - x^+\rangle + \frac{\gamma\mu_r}{2}\|x^+ - u\|^2.$$

After small rearrangements we have what we need to prove. $\qquad\square$

**Theorem 10** (Theorem 4). *Assume A 5, A 6 with $\mu_F + \mu_r > 0$, A 3, A 7. Let problem (9) be solved by Algorithm 3. Then for any $b \in \mathbb{N}^*$, and $\gamma$, $M$ satisfying*

$$\gamma \leq \min\{(3\mu_F + 3\mu_r)^{-1}; (3L)^{-1}; (6\mu_F + \mu_r) \cdot [120(C_1\tau b^{-1} + (C_1+1)\tau^2 b^{-2})\Delta^2]^{-1}; \sqrt{(18C_1)^{-1}\Delta^{-2}\tau^{-1}b}\},$$
$$M = \max\{2; \sqrt{C_2\gamma^{-1}(\mu_F + \mu_r)^{-1}}\}, \quad B = \lceil b\log_2 M\rceil,$$

*it holds that*

$$\mathbb{E}\left[\|x^N - x^*\|^2\right] \lesssim \exp\left(-\frac{(\mu_F + \mu_r)\gamma N}{16}\right)\|x^0 - x^*\|^2 + \frac{\gamma}{\mu}(\sigma^2\tau b^{-1} + \sigma^2\tau^2 b^{-2}).$$

*Proof.* We start from Lemma 10 for $x^{k+1} = \text{prox}_{\gamma r}\left(x^k - \gamma g^k\right)$ with $x^+ = x^{k+1}$, $x = x^k - \gamma g^k$, $u = x^*$ and for $x^{k+1/2} = \text{prox}_{\gamma r}\left(x^k - \gamma B^{-1}\sum_{i=1}^{B} F(x^k, z_i^k)\right)$ with $x^+ = x^{k+1/2}$, $x = x^k - \gamma B^{-1}\sum_{i=1}^{B} F(x^k, z_i^k)$, $u = x^{k+1}$:

$$\langle x^{k+1} - x^k + \gamma g^k, x^* - x^{k+1}\rangle \geq \gamma\left(r(x^{k+1}) - r(x^*) + \frac{\mu_r}{2}\|x^{k+1} - x^*\|^2\right),$$

and

$$\langle x^{k+1/2} - x^k + \gamma B^{-1}\sum_{i=1}^{B} F(x^k, z_i^k), x^{k+1} - x^{k+1/2}\rangle$$
$$\geq \gamma\left(r(x^{k+1/2}) - r(x^{k+1}) + \frac{\mu_r}{2}\|x^{k+1} - x^{k+1/2}\|^2\right).$$

Summing up these two inequalities, we get

$$\langle x^{k+1} - x^k + \gamma g^k, x^* - x^{k+1}\rangle + \langle x^{k+1/2} - x^k + \gamma F(x^k, z^k), x^{k+1} - x^{k+1/2}\rangle$$
$$\geq \gamma\left(r(x^{k+1/2}) - r(x^*) + \frac{\mu_r}{2}\|x^{k+1} - x^*\|^2 + \frac{\mu_r}{2}\|x^{k+1} - x^{k+1/2}\|^2\right).$$

After some rearrangements, we have

$$\langle x^{k+1} - x^k, x^* - x^{k+1}\rangle + \langle x^{k+1/2} - x^k, x^{k+1} - x^{k+1/2}\rangle$$
$$+ \gamma\langle g^k - B^{-1}\sum_{i=1}^{B} F(x^k, z_i^k), x^{k+1/2} - x^{k+1}\rangle + \gamma\langle g^k, x^* - x^{k+1/2}\rangle$$
$$\geq \gamma\left(r(x^{k+1/2}) - r(x^*) + \frac{\mu_r}{2}\|x^{k+1} - x^*\|^2 + \frac{\mu_r}{2}\|x^{k+1} - x^{k+1/2}\|^2\right).$$

With $2\langle a, b\rangle = \|a+b\|^2 - \|a\|^2 - \|b\|^2$, we deduce

$$\|x^k - x^*\|^2 - \|x^{k+1} - x^*\|^2 - \|x^{k+1} - x^k\|^2$$
$$+ \|x^{k+1} - x^k\|^2 - \|x^{k+1/2} - x^k\|^2 - \|x^{k+1} - x^{k+1/2}\|^2$$
$$+ 2\gamma\langle g^k - B^{-1}\sum_{i=1}^{B} F(x^k, z_i^k), x^{k+1/2} - x^{k+1}\rangle + 2\gamma\langle g^k, x^* - x^{k+1/2}\rangle$$
$$\geq 2\gamma\left(r(x^{k+1/2}) - r(x^*) + \frac{\mu_r}{2}\|x^{k+1} - x^*\|^2 + \frac{\mu_r}{2}\|x^{k+1} - x^{k+1/2}\|^2\right).$$

After rewriting in a slightly different way,

$$\|x^{k+1} - x^*\|^2 + \|x^{k+1/2} - x^{k+1}\|^2 \leq \|x^k - x^*\|^2 - 2\gamma\langle g^k, x^{k+1/2} - x^*\rangle$$
$$- 2\gamma\langle B^{-1}\sum_{i=1}^{B} F(x^k, z_i^k) - g^k, x^{k+1/2} - x^{k+1}\rangle$$
$$- \|x^{k+1/2} - x^k\|^2 - 2\gamma(r(x^{k+1/2}) - r(x^*))$$
$$- \mu_r\gamma\|x^{k+1} - x^*\|^2 - \mu_r\gamma\|x^{k+1} - x^{k+1/2}\|^2$$
$$\leq \|x^k - x^*\|^2 - 2\gamma\langle g^k, x^{k+1/2} - x^*\rangle$$
$$+ \gamma^2\left\|B^{-1}\sum_{i=1}^{B} F(x^k, z_i^k) - g^k\right\|^2 + \|x^{k+1/2} - x^{k+1}\|^2$$
$$- \|x^{k+1/2} - x^k\|^2 - 2\gamma(r(x^{k+1/2}) - r(x^*))$$
$$- \mu_r\gamma\|x^{k+1} - x^*\|^2 - \mu_r\gamma\|x^{k+1} - x^{k+1/2}\|^2.$$

In the last step, we used Cauchy-Schwartz inequality (43). Subtracting $\|x^{k+1} - x^{k+1/2}\|^2$ from both parts, we get

$$\|x^{k+1} - x^*\|^2 \leq \|x^k - x^*\|^2 - 2\gamma\langle g^k, x^{k+1/2} - x^*\rangle + \gamma^2\|B^{-1}\sum_{i=1}^{B} F(x^k, z_i^k) - g^k\|^2$$

$$-\|x^{k+1/2} - x^k\|^2 - 2\gamma(r(x^{k+1/2}) - r(x^*))$$
$$-\mu_r\gamma\|x^{k+1} - x^*\|^2 - \mu_r\gamma\|x^{k+1} - x^{k+1/2}\|^2$$
$$=\|x^k - x^*\|^2 - 2\gamma\langle F(x^{k+1/2}), x^{k+1/2} - x^*\rangle$$
$$-2\gamma\langle \mathbb{E}_{k+1/2}[g^k] - F(x^{k+1/2}), x^{k+1/2} - x^*\rangle$$
$$-2\gamma\langle g^k - \mathbb{E}_{k+1/2}[g^k], x^{k+1/2} - x^*\rangle$$
$$+\gamma^2\|F(x^k) - F(x^{k+1/2}) + F(x^k) - B^{-1}\sum_{i=1}^{B} F(x^k, z_i^k) + F(x^{k+1/2}) - g^k\|^2$$
$$-\|x^{k+1/2} - x^k\|^2 - 2\gamma(r(x^{k+1/2}) - r(x^*))$$
$$-\mu_r\gamma\|x^{k+1} - x^*\|^2 - \mu_r\gamma\|x^{k+1} - x^{k+1/2}\|^2.$$

Again with Cauchy-Schwartz inequality (45), we conduct

$$\|x^{k+1} - x^*\|^2 \leq \|x^k - x^*\|^2 - 2\gamma\langle F(x^{k+1/2}), x^{k+1/2} - x^*\rangle$$
$$-2\gamma\langle \mathbb{E}_{k+1/2}[g^k] - F(x^{k+1/2}), x^{k+1/2} - x^*\rangle$$
$$-2\gamma\langle g^k - \mathbb{E}_{k+1/2}[g^k], x^{k+1/2} - x^*\rangle + 3\gamma^2\|B^{-1}\sum_{i=1}^{B} F(x^k, z_i^k) - F(x^k)\|^2$$
$$+3\gamma^2\|F(x^{k+1/2}) - g^k\|^2 + 3\gamma^2\|F(x^{k+1/2}) - F(x^k)\|^2 - \|x^{k+1/2} - x^k\|^2$$
$$-2\gamma(r(x^{k+1/2}) - r(x^*)) - \mu_r\gamma\|x^{k+1} - x^*\|^2 - \mu_r\gamma\|x^{k+1} - x^{k+1/2}\|^2. \quad (41)$$

A 5 and the property of the solution (9): $-(r(x^{k+1/2}) - r(x^*)) \leq \langle F(x^*), x^{k+1/2} - x^*\rangle$, together give

$$\|x^{k+1} - x^*\|^2 \leq \|x^k - x^*\|^2 - 2\gamma\langle F(x^{k+1/2}) - F(x^*), x^{k+1/2} - x^*\rangle$$
$$-2\gamma\langle \mathbb{E}_{k+1/2}[g^k] - F(x^{k+1/2}), x^{k+1/2} - x^*\rangle$$
$$-2\gamma\langle g^k - \mathbb{E}_{k+1/2}[g^k], x^{k+1/2} - x^*\rangle$$
$$+3\gamma^2\|B^{-1}\sum_{i=1}^{B} F(x^k, z_i^k) - F(x^k)\|^2 + 3\gamma^2\|F(x^{k+1/2}) - g^k\|^2$$
$$+3\gamma^2 L^2\|x^{k+1/2} - x^k\|^2 - \|x^{k+1/2} - x^k\|^2$$
$$-\mu_r\gamma\|x^{k+1} - x^*\|^2 - \mu_r\gamma\|x^{k+1} - x^{k+1/2}\|^2.$$

Next, one can apply A 6 and have

$$\|x^{k+1} - x^*\|^2 \leq \|x^k - x^*\|^2 - 2\mu_F\gamma\|x^{k+1/2} - x^*\|^2$$
$$-2\gamma\langle \mathbb{E}_{k+1/2}[g^k] - F(x^{k+1/2}), x^{k+1/2} - x^*\rangle$$
$$-2\gamma\langle g^k - \mathbb{E}_{k+1/2}[g^k], x^{k+1/2} - x^*\rangle + 3\gamma^2\|B^{-1}\sum_{i=1}^{B} F(x^k, z_i^k) - F(x^k)\|^2$$
$$+3\gamma^2\|F(x^{k+1/2}) - g^k\|^2 + 3\gamma^2 L^2\|x^{k+1/2} - x^k\|^2 - \|x^{k+1/2} - x^k\|^2$$
$$-\mu_r\gamma\|x^{k+1} - x^*\|^2 - \mu_r\gamma\|x^{k+1} - x^{k+1/2}\|^2.$$

Using Cauchy-Schwartz inequality (43) one more time, we get

$$\|x^{k+1} - x^*\|^2 \leq \|x^k - x^*\|^2 - 2\mu_F\gamma\|x^{k+1/2} - x^*\|^2$$
$$-\mu_r\gamma\|x^{k+1} - x^*\|^2 - \mu_r\gamma\|x^{k+1} - x^{k+1/2}\|^2$$
$$+\frac{4\gamma}{\mu_F + \mu_r}\|\mathbb{E}_{k+1/2}[g^k] - F(x^{k+1/2})\|^2 + \frac{(\mu_F + \mu_r)\gamma}{4}\|x^{k+1/2} - x^*\|^2$$
$$-2\gamma\langle g^k - \mathbb{E}_{k+1/2}[g^k], x^{k+1/2} - x^*\rangle + 3\gamma^2\|B^{-1}\sum_{i=1}^{B} F(x^k, z_i^k) - F(x^k)\|^2$$

$$+ 3\gamma^2 \|F(x^{k+1/2}) - g^k\|^2 + 3\gamma^2 L^2 \|x^{k+1/2} - x^k\|^2 - \|x^{k+1/2} - x^k\|^2$$

$$\leq \|x^k - x^*\|^2 - \frac{7\mu_F \gamma}{4} \|x^{k+1/2} - x^*\|^2$$

$$- \mu_r \gamma \|x^{k+1} - x^*\|^2 - \mu_r \gamma \|x^{k+1} - x^{k+1/2}\|^2$$

$$+ \frac{\mu_r \gamma}{4} \|x^{k+1/2} - x^*\|^2 + \frac{4\gamma}{\mu_F + \mu_r} \|\mathbb{E}_{k+1/2}[g^k] - F(x^{k+1/2})\|^2$$

$$- 2\gamma \langle g^k - \mathbb{E}_{k+1/2}[g^k], x^{k+1/2} - x^* \rangle + 3\gamma^2 \|B^{-1} \sum_{i=1}^{B} F(x^k, z_i^k) - F(x^k)\|^2$$

$$+ 3\gamma^2 \|F(x^{k+1/2}) - g^k\|^2 + 3\gamma^2 L^2 \|x^{k+1/2} - x^k\|^2 - \|x^{k+1/2} - x^k\|^2.$$

With Cauchy-Schwartz inequality in the form: $-\mu_r \gamma \|x^{k+1} - x^*\|^2 \leq -\frac{\mu_r \gamma}{2} \|x^{k+1/2} - x^*\|^2 + \mu_r \gamma \|x^{k+1} - x^{k+1/2}\|^2$, one can deduce

$$\|x^{k+1} - x^*\|^2 \leq \|x^k - x^*\|^2 - \frac{(7\mu_F + \mu_r)\gamma}{4} \|x^{k+1/2} - x^*\|^2$$

$$+ \frac{4\gamma}{\mu_F + \mu_r} \|\mathbb{E}_{k+1/2}[g^k] - F(x^{k+1/2})\|^2$$

$$- 2\gamma \langle g^k - \mathbb{E}_{k+1/2}[g^k], x^{k+1/2} - x^* \rangle + 3\gamma^2 \|B^{-1} \sum_{i=1}^{B} F(x^k, z_i^k) - F(x^k)\|^2$$

$$+ 3\gamma^2 \|F(x^{k+1/2}) - g^k\|^2 + 3\gamma^2 L^2 \|x^{k+1/2} - x^k\|^2 - \|x^{k+1/2} - x^k\|^2.$$

Taking the expectation and using Lemma 3, Lemma 4 (with $\Delta^2 \|x - x^*\|^2$ instead of $\delta^2 \|\nabla f(x)\|^2$), we have

$$\mathbb{E}\left[\|x^{k+1} - x^*\|^2\right] \leq \mathbb{E}\left[\|x^k - x^*\|^2\right] - \frac{(7\mu_F + \mu_r)\gamma}{4} \mathbb{E}\left[\|x^{k+1/2} - x^*\|^2\right]$$

$$+ \frac{4\gamma}{\mu_F + \mu_r} \mathbb{E}\left[\|\mathbb{E}_{k+1/2}[g^k] - F(x^{k+1/2})\|^2\right]$$

$$+ 3\gamma^2 \mathbb{E}\left[\mathbb{E}_k\left[\|B^{-1} \sum_{i=1}^{B} F(x^k, z_i^k) - F(x^k)\|^2\right]\right]$$

$$+ 3\gamma^2 \mathbb{E}\left[\mathbb{E}_{k+1/2}\left[\|F(x^{k+1/2}) - g^k\|^2\right]\right]$$

$$+ 3\gamma^2 L^2 \mathbb{E}\left[\|x^{k+1/2} - x^k\|^2\right] - \mathbb{E}\left[\|x^{k+1/2} - x^k\|^2\right]$$

$$\leq \mathbb{E}\left[\|x^k - x^*\|^2\right] - \frac{(7\mu_F + \mu_r)\gamma}{4} \mathbb{E}\left[\|x^{k+1/2} - x^*\|^2\right]$$

$$+ \frac{4\gamma}{\mu_F + \mu_r} \cdot C_2 \tau^2 M^{-2} B^{-2} \left(\sigma^2 + \Delta^2 \mathbb{E}\left[\|x^{k+1/2} - x^*\|^2\right]\right)$$

$$+ 3\gamma^2 \cdot C_1 \tau B^{-1} \left(\sigma^2 + \Delta^2 \mathbb{E}\left[\|x^k - x^*\|^2\right]\right)$$

$$+ 3\gamma^2 \cdot \left(4C_1 \tau B^{-1} \log_2 M + (4C_1 + 2)\tau^2 B^{-2}\right) \left(\sigma^2 + \Delta^2 \|x^{k+1/2} - x^*\|^2\right)$$

$$+ 3\gamma^2 L^2 \mathbb{E}\left[\|x^{k+1/2} - x^k\|^2\right] - \mathbb{E}\left[\|x^{k+1/2} - x^k\|^2\right].$$

With $M \geq \sqrt{C_2 \gamma^{-1} (\mu_F + \mu_r)^{-1}}$, we have

$$\mathbb{E}\left[\|x^{k+1} - x^*\|^2\right] \leq \mathbb{E}\left[\|x^k - x^*\|^2\right] - \frac{(7\mu_F + \mu_r)\gamma}{4} \mathbb{E}\left[\|x^{k+1/2} - x^*\|^2\right]$$

$$+ 4\gamma^2 \cdot \tau^2 B^{-2} \left(\sigma^2 + \Delta^2 \mathbb{E}\left[\|x^{k+1/2} - x^*\|^2\right]\right)$$

$$+ 3\gamma^2 \cdot C_1 \tau B^{-1} \left(\sigma^2 + \Delta^2 \mathbb{E}\left[\|x^k - x^*\|^2\right]\right)$$

$$+ 3\gamma^2 \cdot \left(4C_1 \tau B^{-1} \log_2 M + (4C_1 + 2)\tau^2 B^{-2}\right) \left(\sigma^2 + \Delta^2 \mathbb{E}\left[\|x^{k+1/2} - x^*\|^2\right]\right)$$

$$+ 3\gamma^2 L^2 \mathbb{E}\left[\|x^{k+1/2} - x^k\|^2\right] - \mathbb{E}\left[\|x^{k+1/2} - x^k\|^2\right]$$

$$\leq \mathbb{E}\left[\|x^k - x^*\|^2\right] - \frac{(7\mu_F + \mu_r)\gamma}{4}\mathbb{E}\left[\|x^{k+1/2} - x^*\|^2\right]$$
$$+ 3\gamma^2 \cdot C_1 \tau B^{-1} \Delta^2 \mathbb{E}\left[\|x^k - x^*\|^2\right]$$
$$+ 12\gamma^2 \cdot \left(C_1 \tau B^{-1} \log_2 M + (C_1 + 1)\tau^2 B^{-2}\right)\Delta^2 \mathbb{E}\left[\|x^{k+1/2} - x^*\|^2\right]$$
$$+ 15\gamma^2 \cdot \left(C_1 \tau B^{-1} \log_2 M + (C_1 + 1)\tau^2 B^{-2}\right)\sigma^2$$
$$+ 3\gamma^2 L^2 \mathbb{E}\left[\|x^{k+1/2} - x^k\|^2\right] - \mathbb{E}\left[\|x^{k+1/2} - x^k\|^2\right].$$

Cauchy-Schwartz inequality (44) gives

$$\mathbb{E}\left[\|x^{k+1} - x^*\|^2\right] \leq \mathbb{E}\left[\|x^k - x^*\|^2\right] - \frac{(7\mu_F + \mu_r)\gamma}{4}\mathbb{E}\left[\|x^{k+1/2} - x^*\|^2\right]$$
$$+ 15\gamma^2 \cdot \left(C_1 \tau B^{-1} \log_2 M + (C_1 + 1)\tau^2 B^{-2}\right)\Delta^2 \mathbb{E}\left[\|x^{k+1/2} - x^*\|^2\right]$$
$$+ 15\gamma^2 \cdot \left(C_1 \tau B^{-1} \log_2 M + (C_1 + 1)\tau^2 B^{-2}\right)\sigma^2$$
$$+ 6\gamma^2 \cdot C_1 \tau B^{-1} \Delta^2 \mathbb{E}\left[\|x^{k+1/2} - x^k\|^2\right]$$
$$+ 3\gamma^2 L^2 \mathbb{E}\left[\|x^{k+1/2} - x^k\|^2\right] - \mathbb{E}\left[\|x^{k+1/2} - x^k\|^2\right].$$

Since $\gamma \leq (7\mu_F + \mu_r) \cdot \left[120\left(C_1 \tau b^{-1} + (C_1 + 1)\tau^2 b^{-2}\right)\Delta^2\right]^{-1}$, $B = \lceil b \log_2 M \rceil$ and $M \geq 2$, one can obtain

$$\gamma \leq (7\mu_F + \mu_r) \cdot \left[120\left(C_1 \tau b^{-1} + (C_1 + 1)\tau^2 b^{-2}\right)\Delta^2\right]^{-1}$$
$$\leq (7\mu_F + \mu_r) \cdot \left[120\left(C_1 \tau B^{-1} \log_2 M + (C_1 + 1)\tau^2 B^{-2}\right)\Delta^2\right]^{-1},$$

and then,

$$\mathbb{E}\left[\|x^{k+1} - x^*\|^2\right] \leq \mathbb{E}\left[\|x^k - x^*\|^2\right] - \frac{(7\mu_F + \mu_r)\gamma}{8}\mathbb{E}\left[\|x^{k+1/2} - x^*\|^2\right]$$
$$+ 15\gamma^2 \cdot \left(C_1 \tau B^{-1} \log_2 M + (C_1 + 1)\tau^2 B^{-2}\right)\sigma^2$$
$$+ 6\gamma^2 \cdot C_1 \tau B^{-1} \Delta^2 \mathbb{E}\left[\|x^{k+1/2} - x^k\|^2\right]$$
$$+ 3\gamma^2 L^2 \mathbb{E}\left[\|x^{k+1/2} - x^k\|^2\right] - \mathbb{E}\left[\|x^{k+1/2} - x^k\|^2\right].$$

With Cauchy-Schwartz inequality in the form: $-\|x^{k+1/2} - x^*\|^2 \leq -\frac{1}{2}\|x^k - x^*\|^2 + \|x^k - x^{k+1/2}\|^2$, we have

$$\mathbb{E}\left[\|x^{k+1} - x^*\|^2\right] \leq \left(1 - \frac{(7\mu_F + \mu_r)\gamma}{16}\right)\mathbb{E}\left[\|x^k - x^*\|^2\right]$$
$$- \left(1 - (\mu_F + \mu_r)\gamma - 3\gamma^2 L^2 - 6\gamma^2 \cdot C_1 \tau B^{-1}\Delta^2\right)\mathbb{E}\left[\|x^{k+1/2} - x^k\|^2\right]$$
$$+ 15\gamma^2 \cdot \left(C_1 \tau B^{-1} \log_2 M + (C_1 + 1)\tau^2 B^{-2}\right)\sigma^2.$$

Since $\gamma \leq \min\left\{(3\mu_F + 3\mu_r)^{-1}; (3L)^{-1}; \sqrt{(18C_1)^{-1}\tau^{-1}b\Delta^{-2}}\right\}$, we get

$$\mathbb{E}\left[\|x^{k+1} - x^*\|^2\right] \leq \left(1 - \frac{(7\mu_F + \mu_r)\gamma}{16}\right)\mathbb{E}\left[\|x^k - x^*\|^2\right]$$
$$+ 15\gamma^2 \cdot \left(C_1 \tau B^{-1} \log_2 M + (C_1 + 1)\tau^2 B^{-2}\right)\sigma^2.$$

Next, we perform the recursion

$$\mathbb{E}\left[\|x^N - x^*\|^2\right] \leq \left(1 - \frac{(7\mu_F + \mu_r)\gamma}{16}\right)^N \|x^0 - x^*\|^2$$
$$+ \frac{240\gamma}{(\mu_F + \mu_r)} \cdot \left(C_1 \tau B^{-1} \log_2 M + (C_1 + 1)\tau^2 B^{-2}\right)\sigma^2$$

$$\leq \exp\left(-\frac{(\mu_F + \mu_r)\gamma N}{16}\right)\|x^0 - x^*\|^2$$
$$+ \frac{240\gamma}{(\mu_F + \mu_r)} \cdot \left(C_1 \tau B^{-1}\log_2 M + (C_1 + 1)\tau^2 B^{-2}\right)\sigma^2.$$

Substituting $B = \lceil b\log_2 M\rceil$ and using $M \geq 2$ finish the proof. $\qquad\square$

## B.9 Proof of Theorem 5

**Theorem 11** (Theorem 5). *Assume A 5, A 6 with $\mu_F + \mu_r = 0$, A 8, A 3, A 7. Let problem (9) be solved by Algorithm 3. Then for any $B \in \mathbb{N}^*$, and $\gamma$, $M$ satisfying $\gamma \lesssim L^{-1}$, $M = \sqrt{N}$, it holds that*

$$\mathbb{E}\left[Gap(\bar{x}^N)\right] \lesssim \frac{D^2}{\gamma N} + \gamma(\tau B^{-1}\log_2 N + \tau^2 B^{-2})(\sigma^2 + \Delta^2 D^2),$$

*where $\bar{x}^N = \frac{1}{N}\sum_{k=0}^{N-1}x^{k+1/2}$.*

*Proof.* We start from (41) with arbitrary $x \in \mathcal{X}$ instead of $x^*$ and $\mu_r = 0$:

$$\|x^{k+1} - x\|^2 \leq \|x^k - x\|^2 - 2\gamma\langle F(x^{k+1/2}), x^{k+1/2} - x\rangle$$
$$- 2\gamma\langle\mathbb{E}_{k+1/2}[g^k] - F(x^{k+1/2}), x^{k+1/2} - x\rangle$$
$$- 2\gamma\langle g^k - \mathbb{E}_{k+1/2}[g^k], x^{k+1/2} - x\rangle + 3\gamma^2\|B^{-1}\sum_{i=1}^{B}F(x^k, z_i^k) - F(x^k)\|^2$$
$$+ 3\gamma^2\|F(x^{k+1/2}) - g^k\|^2 + 3\gamma^2\|F(x^{k+1/2}) - F(x^k)\|^2 - \|x^{k+1/2} - x^k\|^2$$
$$- 2\gamma(r(x^{k+1/2}) - r(x)).$$

After small rearrangements, we get

$$2\gamma\left(\langle F(x^{k+1/2}), x^{k+1/2} - x\rangle + r(x^{k+1/2}) - r(x)\right)$$
$$\leq \|x^k - x\|^2 - \|x^{k+1} - x\|^2 - 2\gamma\langle\mathbb{E}_{k+1/2}[g^k] - F(x^{k+1/2}), x^{k+1/2} - x\rangle$$
$$- 2\gamma\langle g^k - \mathbb{E}_{k+1/2}[g^k], x^{k+1/2} - x\rangle + 3\gamma^2\|B^{-1}\sum_{i=1}^{B}F(x^k, z_i^k) - F(x^k)\|^2$$
$$+ 3\gamma^2\|F(x^{k+1/2}) - g^k\|^2 + 3\gamma^2\|F(x^{k+1/2}) - F(x^k)\|^2 - \|x^{k+1/2} - x^k\|^2.$$

Applying Cauchy-Schwartz inequality and making more rearrangements, we get

$$2\gamma\left(\langle F(x^{k+1/2}), x^{k+1/2} - x\rangle + r(x^{k+1/2}) - r(x)\right)$$
$$\leq \|x^k - x\|^2 - \|x^{k+1} - x\|^2 + \gamma^2 N\|\mathbb{E}_{k+1/2}[g^k] - F(x^{k+1/2})\|^2 + \frac{1}{N}\|x^{k+1/2} - x\|^2$$
$$- 2\gamma\langle g^k - \mathbb{E}_{k+1/2}[g^k], x^{k+1/2} - x^0\rangle - 2\gamma\langle g^k - \mathbb{E}_{k+1/2}[g^k], x^0 - x\rangle$$
$$+ 3\gamma^2\|B^{-1}\sum_{i=1}^{B}F(x^k, z_i^k) - F(x^k)\|^2 + 3\gamma^2\|F(x^{k+1/2}) - g^k\|^2$$
$$+ 3\gamma^2\|F(x^{k+1/2}) - F(x^k)\|^2 - \|x^{k+1/2} - x^k\|^2.$$

Summing over all $k$ from 0 to $N-1$ and dividing by $N$, we have

$$2\gamma \cdot \frac{1}{N}\sum_{k=0}^{N-1}\left(\langle F(x^{k+1/2}), x^{k+1/2} - x\rangle + r(x^{k+1/2}) - r(x)\right)$$
$$\leq \frac{\|x^0 - x\|^2 - \|x^K - x\|^2}{N} + \gamma^2\sum_{k=0}^{N-1}\|\mathbb{E}_{k+1/2}[g^k] - F(x^{k+1/2})\|^2 + \frac{1}{N^2}\sum_{k=0}^{N-1}\|x^{k+1/2} - x\|^2$$

$$-2\gamma \cdot \frac{1}{N} \sum_{k=0}^{N-1} \langle g^k - \mathbb{E}_{k+1/2}[g^k], x^{k+1/2} - x^0 \rangle - 2\gamma \langle N^{-1} \sum_{k=0}^{N-1} \left[ g^k - \mathbb{E}_{k+1/2}[g^k] \right], x^0 - x \rangle$$

$$+ 3\gamma^2 \cdot \frac{1}{N} \sum_{k=0}^{N-1} \| B^{-1} \sum_{i=1}^{B} F(x^k, z_i^k) - F(x^k) \|^2 + 3\gamma^2 \cdot \frac{1}{N} \sum_{k=0}^{N-1} \| F(x^{k+1/2}) - g^k \|^2$$

$$+ 3\gamma^2 \cdot \frac{1}{N} \sum_{k=0}^{N-1} \| F(x^{k+1/2}) - F(x^k) \|^2 - \frac{1}{N} \sum_{k=0}^{N-1} \| x^{k+1/2} - x^k \|^2.$$

Using monotonicity and Jensen's inequality (46) for convex function $r$, we get (with notation $\bar{x}^N = \frac{1}{N} \sum_{k=0}^{N-1} x^{k+1/2}$)

$$2\gamma \big( \langle F(x), \bar{x}^N - x \rangle + r(\bar{x}^N) - r(x) \big)$$

$$\leq \frac{\|x^0 - x\|^2 - \|x^N - x\|^2}{N} + \gamma^2 \sum_{k=0}^{N-1} \| \mathbb{E}_{k+1/2}[g^k] - F(x^{k+1/2}) \|^2 + \frac{1}{N^2} \sum_{k=0}^{N-1} \| x^{k+1/2} - x \|^2$$

$$- 2\gamma \cdot \frac{1}{N} \sum_{k=0}^{N-1} \langle g^k - \mathbb{E}_{k+1/2}[g^k], x^{k+1/2} - x^0 \rangle - 2\gamma \langle N^{-1} \sum_{k=0}^{N-1} \left[ g^k - \mathbb{E}_{k+1/2}[g^k] \right], x^0 - x \rangle$$

$$+ 3\gamma^2 \cdot \frac{1}{N} \sum_{k=0}^{N-1} \| B^{-1} \sum_{i=1}^{B} F(x^k, z_i^k) - F(x^k) \|^2 + 3\gamma^2 \cdot \frac{1}{N} \sum_{k=0}^{N-1} \| F(x^{k+1/2}) - g^k \|^2$$

$$+ 3\gamma^2 \cdot \frac{1}{N} \sum_{k=0}^{N-1} \| F(x^{k+1/2}) - F(x^k) \|^2 - \frac{1}{N} \sum_{k=0}^{N-1} \| x^{k+1/2} - x^k \|^2.$$

Applying Cauchy-Schwartz inequality (43) one more time,

$$2\gamma \big( \langle F(x), \bar{x}^N - x \rangle + r(\bar{x}^N) - r(x) \big)$$

$$\leq \frac{2\|x^0 - x\|^2}{N} + \gamma^2 \sum_{k=0}^{N-1} \| \mathbb{E}_{k+1/2}[g^k] - F(x^{k+1/2}) \|^2 + \frac{1}{N^2} \sum_{k=0}^{N-1} \| x^{k+1/2} - x \|^2$$

$$- 2\gamma \cdot \frac{1}{N} \sum_{k=0}^{N-1} \langle g^k - \mathbb{E}_{k+1/2}[g^k], x^{k+1/2} - x^0 \rangle + \gamma^2 \| N^{-1} \sum_{k=0}^{N-1} \left[ g^k - \mathbb{E}_{k+1/2}[g^k] \right] \|^2$$

$$+ 3\gamma^2 \cdot \frac{1}{N} \sum_{k=0}^{N-1} \| B^{-1} \sum_{i=1}^{B} F(x^k, z_i^k) - F(x^k) \|^2 + 3\gamma^2 \cdot \frac{1}{N} \sum_{k=0}^{N-1} \| F(x^{k+1/2}) - g^k \|^2$$

$$+ 3\gamma^2 \cdot \frac{1}{N} \sum_{k=0}^{N-1} \| F(x^{k+1/2}) - F(x^k) \|^2 - \frac{1}{N} \sum_{k=0}^{N-1} \| x^{k+1/2} - x^k \|^2.$$

Taking supermom on $x$ from $\mathcal{X}$ and then the full expectation, we get

$$2\gamma \mathbb{E} \left[ \text{Gap}(\bar{x}^N) \right] \leq \frac{2 \max_{x \in \mathcal{X}} \|x^0 - x\|^2}{N} + \frac{1}{N^2} \sum_{k=0}^{N-1} \mathbb{E} \left[ \max_{x \in \mathcal{X}} \| x^{k+1/2} - x \|^2 \right]$$

$$- 2\gamma \cdot \frac{1}{N} \sum_{k=0}^{N-1} \mathbb{E} \left[ \langle g^k - \mathbb{E}_{k+1/2}[g^k], x^{k+1/2} - x^0 \rangle \right]$$

$$+ \gamma^2 \sum_{k=0}^{N-1} \mathbb{E} \left[ \| \mathbb{E}_{k+1/2}[g^k] - F(x^{k+1/2}) \|^2 \right]$$

$$+ \gamma^2 \mathbb{E} \left[ \| N^{-1} \sum_{k=0}^{N-1} \left[ g^k - \mathbb{E}_{k+1/2}[g^k] \right] \|^2 \right]$$

$$+ 3\gamma^2 \cdot \frac{1}{N} \sum_{k=0}^{N-1} \mathbb{E}\left[\|B^{-1}\sum_{i=1}^{B} F(x^k, z_i^k) - F(x^k)\|^2\right]$$

$$+ 3\gamma^2 \cdot \frac{1}{N} \sum_{k=0}^{N-1} \mathbb{E}\left[\|F(x^{k+1/2}) - g^k\|^2\right]$$

$$+ 3\gamma^2 \cdot \frac{1}{N} \sum_{k=0}^{N-1} \mathbb{E}\left[\|F(x^{k+1/2}) - F(x^k)\|^2\right] - \frac{1}{N} \sum_{k=0}^{N-1} \mathbb{E}\left[\|x^{k+1/2} - x^k\|^2\right].$$

One can note that
$$\mathbb{E}\left[\langle g^k - \mathbb{E}_{k+1/2}[g^k], x^{k+1/2} - x^0\rangle\right] = \mathbb{E}\left[\mathbb{E}_{k+1/2}[\langle g^k - \mathbb{E}_{k+1/2}[g^k], x^{k+1/2} - x^0\rangle]\right]$$

$$= \mathbb{E}\left[\langle \mathbb{E}_{k+1/2}[g^k - \mathbb{E}_{k+1/2}[g^k]], x^{k+1/2} - x^0\rangle\right]$$

$$= 0,$$

and (here we also need Cauchy-Schwartz inequality (44))
$$\mathbb{E}\left[\|N^{-1}\sum_{k=0}^{N-1}[g^k - \mathbb{E}_{k+1/2}[g^k]]\|^2\right] = \frac{1}{N^2} \sum_{k=0}^{N-1} \mathbb{E}\left[\|g^k - \mathbb{E}_{k+1/2}[g^k]\|^2\right]$$

$$+ \frac{1}{N^2} \sum_{k\neq j} \mathbb{E}\left[\langle g^k - \mathbb{E}_{k+1/2}[g^k], g^j - \mathbb{E}_{j+1/2}[g^j]\rangle\right]$$

$$= \frac{1}{N^2} \sum_{k=0}^{N-1} \mathbb{E}\left[\|g^k - \mathbb{E}_{k+1/2}[g^k]\|^2\right]$$

$$+ \frac{2}{N^2} \sum_{k>j} \mathbb{E}\left[\langle \mathbb{E}_{k+1/2}[g^k - \mathbb{E}_{k+1/2}[g^k]], g^j - \mathbb{E}_{j+1/2}[g^j]\rangle\right]$$

$$= \frac{1}{N^2} \sum_{k=0}^{N-1} \mathbb{E}\left[\|g^k - \mathbb{E}_{k+1/2}[g^k]\|^2\right]$$

$$\leq \frac{2}{N^2} \sum_{k=0}^{N-1} \mathbb{E}\left[\|g^k - F(x^{k+1/2})\|^2\right]$$

$$+ \frac{2}{N^2} \sum_{k=0}^{N-1} \mathbb{E}\left[\|F(x^{k+1/2}) - \mathbb{E}_{k+1/2}[g^k]\|^2\right].$$

Then, we have
$$2\gamma\mathbb{E}\left[\text{Gap}(\bar{x}^N)\right] \leq \frac{2\max_{x\in\mathcal{X}}\|x^0 - x\|^2}{N} + \frac{1}{N^2} \sum_{k=0}^{N-1} \mathbb{E}\left[\max_{x\in\mathcal{X}}\|x^{k+1/2} - x\|^2\right]$$

$$+ 2\gamma^2 \sum_{k=0}^{N-1} \mathbb{E}\left[\|\mathbb{E}_{k+1/2}[g^k] - F(x^{k+1/2})\|^2\right]$$

$$+ 3\gamma^2 \cdot \frac{1}{N} \sum_{k=0}^{N-1} \mathbb{E}\left[\|B^{-1}\sum_{i=1}^{B} F(x^k, z_i^k) - F(x^k)\|^2\right]$$

$$+ 5\gamma^2 \cdot \frac{1}{N} \sum_{k=0}^{N-1} \mathbb{E}\left[\|F(x^{k+1/2}) - g^k\|^2\right]$$

$$+ 3\gamma^2 \cdot \frac{1}{N} \sum_{k=0}^{N-1} \mathbb{E}\left[\|F(x^{k+1/2}) - F(x^k)\|^2\right] - \frac{1}{N} \sum_{k=0}^{N-1} \mathbb{E}\left[\|x^{k+1/2} - x^k\|^2\right].$$

With A 6 and A 8, we obtain
$$2\gamma\mathbb{E}\left[\text{Gap}(\bar{x}^N)\right] \leq \frac{3D^2}{N} + 2\gamma^2 \sum_{k=0}^{N-1} \mathbb{E}\left[\|\mathbb{E}_{k+1/2}[g^k] - F(x^{k+1/2})\|^2\right]$$

$$+ 3\gamma^2 \cdot \frac{1}{N} \sum_{k=0}^{N-1} \mathbb{E}\left[\|B^{-1}\sum_{i=1}^{B} F(x^k, z_i^k) - F(x^k)\|^2\right]$$

$$+ 5\gamma^2 \cdot \frac{1}{N} \sum_{k=0}^{N-1} \mathbb{E}\left[\|F(x^{k+1/2}) - g^k\|^2\right]$$

$$- (1 - 3\gamma^2 L^2)\frac{1}{N} \sum_{k=0}^{N-1} \mathbb{E}\left[\|x^{k+1/2} - x^k\|^2\right].$$

Using Lemma 3 and Lemma 4, we have

$$2\gamma\mathbb{E}\left[\mathrm{Gap}(\bar{x}^N)\right] \leq \frac{3D^2}{N} + 2\gamma^2 C_2 \tau^2 M^{-2}B^{-2} \sum_{k=0}^{N-1} \left(\sigma^2 + \Delta^2\mathbb{E}\left[\|x^{k+1/2} - x^*\|^2\right]\right)$$

$$+ 3\gamma^2 C_1 \tau B^{-1} \cdot \frac{1}{N} \sum_{k=0}^{N-1} \left(\sigma^2 + \Delta^2\mathbb{E}\left[\|x^k - x^*\|^2\right]\right)$$

$$+ 20\gamma^2(C_1\tau B^{-1}\log_2 M + (C_1+1)\tau^2 B^{-2}) \cdot \frac{1}{N} \sum_{k=0}^{N-1} \left(\sigma^2 + \Delta^2\mathbb{E}\left[\|x^{k+1/2} - x^*\|^2\right]\right)$$

$$- (1 - 3\gamma^2 L^2)\frac{1}{N} \sum_{k=0}^{N-1} \mathbb{E}\left[\|x^{k+1/2} - x^k\|^2\right].$$

Again with A 8, we get

$$2\gamma\mathbb{E}\left[\mathrm{Gap}(\bar{x}^N)\right] \leq \frac{3D^2}{N} + 2\gamma^2 C_2 \tau^2 M^{-2}B^{-2}N\left(\sigma^2 + \Delta^2 D^2\right)$$
$$+ 3\gamma^2 C_1 \tau B^{-1}\left(\sigma^2 + \Delta^2 D^2\right)$$
$$+ 20\gamma^2(C_1\tau B^{-1}\log_2 M + (C_1+1)\tau^2 B^{-2}) \cdot \left(\sigma^2 + \Delta^2 D^2\right)$$
$$- (1 - 3\gamma^2 L^2)\frac{1}{N} \sum_{k=0}^{N-1} \mathbb{E}\left[\|x^{k+1/2} - x^k\|^2\right].$$

With $M = \sqrt{N}$ and $\gamma \leq (3L)^{-1}$, one can deduce

$$2\gamma\mathbb{E}\left[\mathrm{Gap}(\bar{x}^N)\right] \leq \frac{3D^2}{N} + 25\gamma^2(C_1\tau B^{-1}\log_2 M + (C_1 + C_2 + 1)\tau^2 B^{-2}) \cdot \left(\sigma^2 + \Delta^2 D^2\right).$$

Substituting $M = \sqrt{N}$ finishes the proof. $\qquad\square$

## C   Basic Facts

**Lemma 11** (see Lemma 1.2.3 and Theorem 2.1.5 from [76])**.** *If $f$ is L-smooth in $\mathbb{R}^d$, then for any $x, y \in \mathbb{R}^d$*

$$f(x) - f(y) - \langle\nabla f(y), x - y\rangle \leq \frac{L}{2}\|x - y\|^2. \tag{42}$$

**Lemma 12** (Cauchy Schwartz inequality)**.** *For any $a, b, x_1, \ldots, x_n \in \mathbb{R}^d$ and $c > 0$ the following inequalities hold:*

$$2\langle a, b\rangle \leq \frac{\|a\|^2}{c} + c\|b\|^2, \tag{43}$$

$$\|a + b\|^2 \leq \left(1 + \frac{1}{c}\right)\|a\|^2 + (1 + c)\|b\|^2, \tag{44}$$

$$\left\|\sum_{i=1}^{n} x_i\right\|^2 \leq n \cdot \sum_{i=1}^{n}\|x_i\|^2. \tag{45}$$

**Lemma 13** (Jensen's inequality). *If $f$ is a convex function, then for any $n \in \mathbb{N}^*$ and $x_1, \ldots, x_n \in \mathbb{R}^d$ the following inequality holds:*

$$f\left(\frac{1}{n}\sum_{i=1}^{n}x_i\right) \leq \frac{1}{n}\sum_{i=1}^{n}f(x_i). \tag{46}$$