# OpenReview forum: "First Order Methods with Markovian Noise: from Acceleration to Variational Inequalities"
_NeurIPS.cc/2023/Conference — NeurIPS 2023 poster_

### Official Review · Reviewer_8XQi · 2023-06-26

**Soundness:** 4 excellent
**Presentation:** 4 excellent
**Contribution:** 3 good
**Rating:** 7
**Confidence:** 4

**Summary:**

The authors study first order stochastic algorithms under a Markovian noise: the noise is no longer i.i.d. as for vanilla SGD, but follows a Markov chain, of finite mixing time $\tau$. The proposed algorithms are SGD-like algorithms, with batchsizes that can vary. In particular, they use a randomized batchsize strategy, whose aim is to trade-off between the noise variance of the stochastic gradients, and their bias (due to non-iid sampling). Batches are typically taken equal to the mixing time of the Markov chain.
An accelerated SGD algorithm is first introduced, and is proved to converge in the smooth strongly convex regime, with a rate that depends  on the mixing time, a maximal gradient noise (max in terms of Markov chain state space and variables), condition number and strong convexity parameter. Various provided lower bounds show that this rate, although not optimal, has an optimal noise dependency.
The SGD algorithm is then studied in the smooth non-convex setting, to obtain typical rates for this algorithm under this type of noise assumption.
Finally, the authors study variational inequalities under Markovian noise, by introducing Randomized Extra Gradient (with also their batchsize strategy).

**Strengths:**

Overall, the paper is well written, and very easy to follow, on a very interesting topic (stochastic approximation/optimization, under Markovian noise), that has re-gained attention lately.

In this line of work, the proposed algorithm and analysis of accelerated SGD is novel, this is the first accelerated rate for SGD under Markov chain noise. The batch size technique is also nice, and well explained. Together, these two techniques give a nice algorithm with a near-optimal rate, as shown by the lower bounds, that are more general than previous ones in this setting.

Although I am not very familiar of variational inequalities, the topic of VA with MC noise seems to be not very well studied in the literature, and provide the first bound with complexity proportional to the mixing time.

**Weaknesses:**

There are two major concerns. Questions related to these are below in the dedicated section.

1) **The noise assumption**. The noise parameter $\sigma^2$ is defined as (forgetting about the $\delta^2$): $\sigma^2\geq \sup_{z,x} \|\nabla_x F(x,z)-f(x)\|^2$. Here, taking the sup over $x\in\mathbb R^d$ is not problematic (and $\delta^2$ is usually there to compensate for this), but I worry about this sup over $z\in Z$. Indeed, for the very simple setting of gaussian additive noise, the noise term here is equal to infinity. Previous works (e.g. [28] in the paper) instead consider $\bar \sigma^2$ for noise assumption (in the smooth non-convex or smooth-PL settings), where a mean is taken over $z$ instead of a sup. $\sigma^2$ here is thus not a noise variance, but an almost sure bound, which can be confusing.
Table 1 should also discuss this noise assumption.

2) **Non-convex rate, Table, and missing comparison**. *(a)* The same non-convex rate is present in [28], under milder noise assumptions. *(b)* The strongly convex rate in this same reference has no mixing time in the transient regime (i.e. in the linearly converging term), and its noise is taken at the *optimum* (and is thus often null). *(c)* *We note that the complexity
55 bounds which scale linearly with the mixing time of the underlying Markov chain are currently
56 available only for non-convex minimization problems*: this assertion is false, since rates exists for convex problems, PL problems [23,28].
These three points *(a,b,c)* lead the reviewer to have some concerns with comparison related works that are cited in this paper (and that I would not have read if not cited, so that this leads me to believe, are there some other existing works like this in the literature ?). Indeed, the non-convex rates should clearly be compared with those of [28], that seem stronger, if I understand well. Also, their strongly convex rates should be discussed: the rate of Theorem 1 can be faster in some cases but not all I think. These rates should thus be in table 1 also.

**Questions:**

Questions related to the two major weaknesses/concerns above:

1. Is the accelerated rate in Theorem 1 obtained by adding stronger assumptions than previous works (i.e. a sup over the state space instead of a mean) ? I would like the authors to discuss this assumption, and its relation with other works.

2. Could the author provide a clearer comparison of their work to the related ones ? I think this requires adding some discussions in several parts of the paper.

3. The lower bound in [28] is in term of hitting time, and the authors of this paper provide an algorithm and analysis without noise assumption; could this be related to the current submission ?


Overall, I vote for acceptance, despite the concerns noted above, due to the contributions in variational inequalities, for the clarity of the paper, and the accelerated rate with batch randomization, while waiting for the authors' answer in order to provide a final decision.

**Limitations:**

No such limitation noted.

---

> ### Author Rebuttal · Authors · 2023-08-06
>
> We thank Reviewer __8XQi__ for the work and for the positive feedback! Next, we answer the issues raised.
>
> __W1__ and __Q1__(noise):
>
> Thank you! The noise assumptions are indeed an important part that needs to be covered. We analyze in detail the assumptions on noise from [28]. In Section 5.1, the author uses $\bar \sigma$, as the reviewer noted, but in addition to $\bar \sigma$, the estimates include $\sigma_{\max} = \max_v \sigma_v$, which is the analog of the supremum in the infinite case we consider. In particular, in Theorem 2, $\Delta = \sigma_{\max}^2/L + f(x^0) - f(x^*)$. In general, the boundedness assumption w.r.t. $z$ comes alongside with the mixing assumption in terms of the total variation distance. Using the unbounded (w.r.t. $z$) noisy oracles $\nabla_{x} F(x,z)$ requires another types of mixing, in particular, in terms of weighted Wassertein metrics, which makes the respective analysis more complicated. Moreover, in case of unbounded gradient oracles it might be complicated to obtain a direct counterpart of Lemma 1 and Lemma 2. For example, a counterparts of Lemma~1 would contain dependence upon initial condition $\xi$ in the r.h.s., which makes the standard recursion argument for the $k$-th iteration error much more involved.
>
> The authors in [28] also assume that $f_{\xi}$ are $L$-smooth for all $\xi$. In the finite sum setup considered in [28], such assumptions are reasonable. But $L$ for all $f_{\xi}$ can be larger than $L$ for $f$. Moreover, if we pass to the more general case (infinite) that we consider, the assumptions raise questions. In particular, if $f (x, \xi) = f(x) + \xi \cdot x^2 / 2$ and  $\nabla_x f (x, \xi) = \nabla f(x) + \xi x$. Let us check the $L$-smoothness of $f(\cdot, \xi)$ for all $\xi$:
> $$
> ||\nabla f (x, \xi) - \nabla f (y, \xi)||^2 = ||\nabla f(x) - \nabla f(y) + \xi (x-y)||^2 \leq 2 ||\nabla f(x) - \nabla f(y) ||^2 + 2\xi^2|| x-y||^2 \leq 2 (L_f^2 + \xi^2) || x-y||^2.
> $$
> In this case, $2L^2 = 2 \sup_\xi [L_f^2 + \xi^2]$. Sections 5.2, 5.3 use noise in the optimal point, but additionally assume that all $f_{\xi}$ are $\mu$-strongly convex. This discussion shows that all assumptions must be clearly compared. In particular for [28], we added two rows to Table 1 as well as comments in footnote:
>
> MC SGD [28] (Sec. 5.1) (7) |  + | - | - | - | $O\left( \frac{\tau (L(f(x^0) - f(x^*)) + \sigma^2)}{\varepsilon^2} + \frac{\tau (L(f(x^0) - f(x^*)) + \sigma^2) \bar \sigma^2}{\varepsilon^4}\right)$ | $O \left( \frac{\tau L}{\mu} \log \frac{(f(x^0) - f(x^*))/\mu + \sigma^2/(\mu L)}{\varepsilon} + \frac{\tau \bar \sigma^2}{\varepsilon \mu^2}\right)$
>
> MC SGD [28] (Sec. 5.2) (8) |  + | + | - | - | - | $O \left( \frac{L}{\mu} \log \frac{||x^0 - x^* ||^2}{\varepsilon} + \frac{L \tau \sigma^2_*}{\mu^3 \varepsilon}\right)$
>
> (7)  considers Markov noise with finite state space, uniformly bounds noise with $\sigma$, but use $\bar \sigma^2$ as a variance of noise. Also the author additionally assumes that all stochastic realization $f_\xi$ are $L$-smooth
>
> (8)  considers Markov noise with finite state space, $\sigma_*$ bounds noise only in $x^*$, but additionally assumes that all stochastic realization $f_\xi$ are $L$-smooth and $\mu$-strongly convex
>
> In "Comparison" paragraphs (Sections 2.1 and 2.3) we added a comparison of noise assumptions not only for our paper and [28], but also for others from Table 1. In particular, we note that in [86], [19], [20], [21] the authors use a uniform bound (supremum-like). But [23] deals with mean/expectation bound for the stochastic gradient.
>
> __W2__ and __Q2__(rate):
>
> We revised the comparison part, Table 1 and the paragraphs at the end of the subsections. As mentioned in the response to the __Q1__, we added a comparison of assumptions, and at the request of Reviewer __9UP4__ we added a comparison of steps that are considered in different algorithms. Regarding points (a)-(c):
>
> a) Our estimate for oracle complexity in the non-convex setting: $\mathcal{\tilde O} \left( \tau \cdot \left[\frac{(1 + \delta^2) L (f(x^0) - f^*)}{\varepsilon^2}+ \frac{L(f(x^0) - f^*)\sigma^2}{ \varepsilon^4} \right] \right)$. The estimate from [28]: $\mathcal{O} \left( \frac{\tau (L(f(x^0) - f(x^*)) + \sigma^2)}{\varepsilon^2} + \frac{\tau (L(f(x^0) - f(x^*)) + \sigma^2) \bar \sigma^2}{\varepsilon^4}\right)$.
> In particular, if $\delta = 0$, we are mainly interested in the second term of both estimates. And here the answer to which estimate is better depends on whether it is smaller  $L(f(x^0) - f^*)\sigma^2$ or $\bar \sigma^2 \sigma^2$. In the other words, if  $L(f(x^0) - f^*) \leq \bar \sigma^2$, then our estimate is better and vice versa.
>
> b) See the answer to the reviewer's previous question. We added two results from [28] to Table 1. We added to the comparison that an effect with independence of $\tau$ was found in the paper [28], but the price of this effect is an additional assumption, as well as an additional factor of $L/\mu$ in the estimate.
>
> c) We rewrote this sentence as follows: "We note that the complexity bounds which scale linearly with the mixing time of the underlying Markov chain are currently available only for general convex and non-convex minimization problems. Namely, [23] has investigated a version of the ergodic mirror descent algorithm that yields optimal convergence rates for Lipschitz, general convex and nonconvex problems." We also added to lines 78-79: "Meanwhile, (28) also achieved linear scaling by mixing time in the non-convex (general and PL) as well as the strongly convex cases."
>
> __Q3__:
>
> The noise assumption in Theorem~$2$ is contained in Assumption A3 and is expressed in terms of the mixing time $\tau$, which is naturally related to the hitting time from [28] in case of finite state-space Markov chains. While in such setup hitting time is a natural quantity, for general Markov noise the lower bounds in terms of the mixing time $\tau$ are more suitable. We hope we have understood the question correctly.

---

> > ### Comment · Reviewer_8XQi · 2023-08-17
> >
> > I thank the authors for all their clarifications and suggestions that are welcomed, and keep my good opinion of this paper unchanged

---

> > > ### Author Response · Authors · 2023-08-18
> > >
> > > Thanks so much for the reply! Thanks again for the detailed review!

---

### Official Review · Reviewer_RVr5 · 2023-07-04

**Soundness:** 4 excellent
**Presentation:** 3 good
**Contribution:** 3 good
**Rating:** 6
**Confidence:** 4

**Summary:**

The authors design and analyze first-order methods for stochastic minimization and stochastic variational inequalities (VIs), with Markovian noise. Typically in stochastic minimization, the target function is only accessible at point x, defined as $f(x)$, and its gradient $\nabla f(x)$, through the sampling of a seed $z \in Z$ using an oracle $F(x,z)$ and its gradient $\nabla F(x,z)$. Under Markovian noise, the seed $z_t$ sampled at time $t$, is a state in a time-homogeneous Markov chain with a unique invariant distribution $\pi$. Under this assumption, the goal is to minimize the expected value over the invariant distribution, that is $\min_x f(x)=E_{z\sim \pi}[F(x,z)]$.

The authors make an assumption about the mixing time of the Markov chain. Beyond this, they make no other fundamental assumptions about the Markov chain. Moreover, for every point $x$ and seed $z$, and for some positive $\sigma,\delta$, the noisy gradient $\nabla F(\cdot,\cdot)$ satisfies the following inequality:
$$
\|\nabla f(x) - \nabla F(x,z) \|^2 \leq \sigma^2 + \delta^2 \|\nabla f(x)\|^2,
$$

The authors prove the following results:
1. For strongly-convex minimization, they design and analyze an accelerated algorithm. This algorithm builds upon the accelerated stochastic gradient descent by Nesterov, and they further use randomized batch sizes and momentum. Additionally, they show that their convergence rate is tight in all parameters, except for $\delta$. They also show that the dependence on $\delta$ and the parameter that controls the Markov chain's mixing is necessary.
2. The authors extend their results to non-convex minimization as well as to stochastic variational inequalities (VIs) with Markovian noise. They design and analyze algorithms for both settings by modifying existing algorithms, specifically, the Stochastic Gradient Descent and Stochastic Extra Gradient. They use the randomized batching technique, previously used in their algorithm for strongly-convex minimization.



After the rebuttal, the authors effectively addressed my concerns about the real-world applications of their noise model. Furthermore, in my view, they have successfully addressed the main concerns raised by the other reviewers. Consequently, I maintained my original score.

**Strengths:**

The assumptions about the Markovian noise make sense especially in setting where the value of the noise sample is only influenced by factors exogenous of the picked action. Moreover, I think the paper is well-written, and the comparison with related work seems extensive. The authors seem to improve either in terms of assumptions, or in terms of convergence in all their settings. The results look reasonable, I checked the statements in the main body.

**Weaknesses:**

Given that this is a theory paper, the only minor weaknesses is that for the stochastic strongly-convex minimization, the bound is not tight for parameter $\delta$, and the fact that they are missing lower bounds for the non-convex setting, as well as the stochastic variational inequality setting. I do not consider this a big weakness.

**Questions:**

Could the authors provide more examples of stochastic minimization or stochastic VIs with Markovian noise?

**Limitations:**

I think that the authors have addressed the limitations of their work, by listing the necessary assumptions for their theorems. Moreover, I do not believe that this work will have negative societal impact.

---

> ### Author Rebuttal · Authors · 2023-08-06
>
> We thank the Reviewer __RVr5__ for the work and for the positive feedback! Next, we answer the issues raised.
>
> __Q1__(examples of Markovian noise):
>
> This is a very interesting and important question. Next, we give a number of examples which we will try to fit into the revised paper:
>
> _Coordinate._ Assumptions similar to A4 often arise in stochastic optimization. For example, in coordinate methods, when we use only part of coordinates instead of the full gradient. The existing literature assumes that we choose coordinates randomly and independently at each iteration, see e.g. [R1, R2]. Choosing the updated indices according to the Markovian dependency extends the possibilities of the existing methods. The idea of taking into account the history of the index updates seems natural. For example, if we used a coordinate in a previous iteration, it may be worth reducing the probability of its selection in the current iteration. Such family of methods is a very interesting area for future research.
>
> _Compression._ Compression is in some sense a generalization of the coordinate approach when we compress the transmitted information for the fastest communication between computing units in a distributed environment. Often compression operators utilize randomness [R3, R4]. As in the case of coordinate methods - the use of prehistory can be useful.
>
> _Mini-batching._ Assumption A4 often performs as well when we consider the gradient not over the whole sample, but over a batch. Here too, Markovian stochasticity can help, because a random choice of a batch number may lose to a non-random choice [R5, R6].
>
> _RL._ Reinforcement learning setting naturally falls into the Markovian stochasticity framework, since the gathered state-action sequence typically forms a Markov chain. The usual MDP setting falls naturally inside this paradigm. The Markovian noise here naturally comes into play from observing the triplets $(s_t,a_t,s_t'), t \in \mathbb{N}$, where $s_t$ is the current state of the environment, $a_t \sim \pi(\cdot | s_t)$ is an action sampled on the $t$-th iteration and $s_t'$ is the next state of the environment, which is typically assumed to depend only on history only through $s_t, a_t$, that is, $s_t' \sim P(\cdot | s_t, a_t)$ for an appropriate Markov kernel $P$. Moreover, the analysis of non-tabular RL problems requires to deal with the general state-space Markov noise. Potential applications include the policy evaluation methods, such as the celebrated temporal difference (TD(0), TD($\lambda$)) with or without functional approximation, see [R7]. We can apply the same methodology to the methods of policy gradient family, e.g. REINFORCE [R8], both in on-policy or off-policy settings.
>
> [R1] Nesterov. Efficiency of coordinate descent methods on huge-scale optimization problems
>
> [R2] Richtárik et al. Iteration complexity of randomized block-coordinate descent methods for minimizing a composite function
>
> [R3] Alistarh et al. QSGD: Communication-efficient SGD via gradient quantization and encoding
>
> [R4] Mishchenko et al. Distributed learning with compressed gradient differences
>
> [R5] Mishchenko et al. Random reshuffling: Simple analysis with vast improvements
>
> [R6] Koloskova et al. Shuffle SGD is Always Better than SGD: Improved Analysis of SGD with Arbitrary Data Orders
>
> [R7] Sutton. Learning to predict by the methods of temporal differences
>
> [R8] Sutton et al. Reinforcement learning: An introduction
>
> __W1__(bound is not tight for parameter $\delta$):
>
> Unfortunately yes, that’s what we're thinking about right now. In this paper we are clearly reflecting that we have this problem at the moment. You might also find interesting the related discussion with Reviewer __9UP4__.
>
> __W1__(lower bounds for VIs and non-convex):
>
> Lower bounds for variational inequalities can be obtained via lower bounds for saddle point problems, which are a special case of variational inequalities. The method for obtaining lower bounds for saddle point problems is reduced to obtaining estimates for the strongly convex minimization problem (see [R1, R2] for deterministic lower bounds), which we provide in our paper. Therefore, it is possible to obtain lower bounds for saddle point problems and hence for variational inequalities. For non-convex problems, lower bounds can also be obtained (see [R3, R4] for deterministic lower bounds).  But these bounds will have the same issue (dependence of $\delta$ – “the bound is not tight for parameter $\delta$”). Thus we think it is more important to focus on the lower bounds for the minimization problem and understand the dependence on $\delta$ to the end, and then move on to other cases. To clarify the reviewer's question, we added a remark to the paper about how lower bounds for variational inequalities and nonconvex minimization can be obtained, in this remark we also noted that the bounds will have the same open question as lower bounds for strongly convex minimization.
>
> [R1] Zhang et al. On lower iteration complexity bounds for the saddle point problems
>
> [R2] Han et al. Lower complexity bounds of finite-sum optimization problems: The results and construction
>
> [R3] Carmon et al. Lower bounds for finding stationary points i
>
> [R4] Zhou et al. Lower bounds for smooth nonconvex finite-sum optimization.

---

> > ### Comment · Reviewer_RVr5 · 2023-08-14
> >
> > Thank you for your comprehensive rebuttal. I found the applications you highlighted particularly interesting. I'm also grateful that you addressed my concerns regarding lower bounds for other settings. I agree with your reasoning regarding the importance of lower bounds for the minimization setting.
> >
> > After examining feedback and rebuttals from other reviewers, I was particularly drawn to the discussions with reviewer 9UP4 regarding mixing time and the decaying step-size, as well as the conversation with reviewer 8XQi about the related work. Your rebuttal and proof sketch have addressed these newfound concerns. Given these reasons, I've opted to maintain my score.

---

> > > ### Author Response · Authors · 2023-08-15
> > >
> > > Thank you so much for the detailed response! We thank you again for the work and time spent on the review and the discussion process!

---

### Official Review · Reviewer_9UP4 · 2023-07-05

**Soundness:** 3 good
**Presentation:** 4 excellent
**Contribution:** 3 good
**Rating:** 6
**Confidence:** 4

**Summary:**

This paper presents an optimization study focusing on stochastic optimization with Markovian noise as opposed to i.i.d. noise. The authors explore three distinct scenarios: strongly convex, non-convex, and Variational Inequalities (VI). In comparison to previous research, they relax the boundedness assumptions related to the domain or gradient magnitude for cases 1 and 2. The newly proposed convergence bounds exhibit linear dependence on the mixing time, whereas previous works relied on quadratic or exponential dependence. The authors establish a lower bound to demonstrate the optimality of this linear dependence. Notably, this work is the first to demonstrate the convergence of the stochastic EG algorithm for VI inequalities under Markovian noise.

**Strengths:**

This paper introduces a relaxation of the restrictive assumptions found in previous analyses by adopting common and mild assumptions. It explores three distinct settings and presents a comprehensive framework for their convenient analysis. The exposition of the paper is accessible and the proofs are clearly articulated.

**Weaknesses:**

One major limitation of the method is its lack of obliviousness to the mixing time. This paper addresses this weakness by achieving linear dependence on the mixing time; however, this is accomplished through a dependence on multilevel mini-batching, where the mini-batch sizes are contingent on the mixing time. This particular dependence was absent in some previous works, which may explain their quadratic dependence on the mixing time.

Another limitation is the comparison of convergence with existing results. While the paper relaxes certain restrictive assumptions, its convergence is limited to the neighborhood. In contrast, for example, [21] achieves asymptotic convergence to the minima. Moreover, when analyzing convergence with fixed step size or momentum parameters, the feasible parameter values in this work rely on σ and other unknown values to achieve an ε error.

Finally, it would be beneficial to augment the paper with experimental results that validate the upper bound. Specifically, it would be valuable to observe the linear dependence on the mixing time and also include numerical results for the Variational Inequalities (VI) setting

**Questions:**

1- There are some typos in your proofs in supplementary materials e.g. in the proof of Lemma 2, in the equation after line 601, it should be Log M instead of log N and x^k_g instead of x^k.

2- There is no discussion in the paper about a range of \delta^2 appearing in the accelerated rate. It could be large as the condition number and in that case, the rate given in the paper is worse than SGD in terms of dependence on the condition number. Why the accelerated rate doesn’t have backward compatibility in this case?

3- Your convergence proofs are based on fixed momentum and step size parameters. And due to that, you get a convergence to the neighborhood. Could you analyze in the decaying step size regime?

**Limitations:**

Yes

---

> ### Author Rebuttal · Authors · 2023-08-06
>
> We thank the Reviewer __9UP4__ for the work and for the positive feedback! Next, we answer the issues raised.
>
> __W1__(mixing time):
>
> We agree that the demand of knowing mixing time in advance is a significant limitation of the method. Estimating mixing time $\tau$ from a single trajectory of the running Markov chain is known to be computationally hard problem, see e.g. [R1] and references therein. At the same time, methods, which share the same (optimal) linear scaling of the sample complexity w.r.t. the mixing time also share the same drawback as our method. In particular, it holds true for the EMD algorithm [R2], SGD-DD algorithm [R3], and usual SGD with Markovian data [R4]. The reviewer is right that the sample complexity of vanilla SGD is indeed quadratic in mixing time, provided that the learning rate is agnostic to $\tau$. At the same time, the sample complexity rates  which are linear w.r.t. $\tau$ for accelerated SGD do not automatically follow from the known techniques [R5] with rescaling step size by $\tau$. Moreover, as stated in the main paper, the corresponding results of [R5] are incomplete.
>
> The only work which is known to us, which is truly oblivious to the mixing time, is the paper [R6]. It also relies on the randomized batch size techniques, yet it allows to obtain the optimal sample complexity rates in a non-convex setting with AdaGrad learning rate. Thus it is an interesting direction for the future work to suggest a procedure that would allow to generalize the results of [R6] to accelerated SGD setting.
>
> [R1] Wolfer et al. Estimating the mixing time of ergodic markov chains.
>
> [R2] Duchi et al. Ergodic mirror descent.
>
> [R3] Nagaraj et al. Least squares regression with markovian data: Fundamental limits and algorithms.
>
> [R4] Even. Stochastic gradient descent under Markovian sampling schemes.
>
> [R5] Doan et al. Convergence rates of accelerated markov gradient descent with applications in reinforcement learning.
>
> [R6] Dorfman et al. Adapting to mixing time in stochastic optimization with markovian data.
>
> __W2__ and __Q3__(stepsize):
>
> We provide a sketch of the proof for the diminishing step size in our general reply. We allow for a decreasing step size, and the expression for the step depends only on $L$, $\mu$, which seems more practical (as requested by the reviewer).  We added this reasoning to the text of the paper as a separate corollary of Theorem 1, and also made similar ones for VIs and the non-convex case. We agree with the reviewer that the results of some other papers are sometimes less demanding on the step selection, and we also added this to the text of the paper in the "Comparison" paragraphs at the end of Sections 2.1 and 2.3.
>
> __W3__(experiments):
>
> Unfortunately in a week to respond, we couldn't make the final version of the experiments, but we promise to add a small set of experiments to vary the theory in the new version of the paper (the final one, if the paper is accepted). We aim to illustrate the benefits of the proposed approaches on the policy evaluation problems on the discrete type environments solved within the TD ($\lambda$)-type algorithms.
>
> __Q1__(typos):
>
> Thank you very much for careful reading of the paper. We also made an additional proofreading of the supplemental materials to catch typos.
>
> __Q2__($\delta$ discussion):
>
> Thank you for your important comment. Indeed, it is possible that the quantity $\delta^2$ scales as $L/\mu$, and the resulting sample complexity bound scales as $(L/\mu)^{3/2}$. At the same time, this drawback is shared by the classical results on learning under the so-called strong growth condition, see e.g. [R1]. As it is shown in [R2], the respective rates can be worse than the ones obtained by usual SGD even under the i.i.d. noise setting, see Appendix F.3 in [R2]. Under particular noise settings the acceleration over SGD might be not possible, see e.g. [R3].
>
> Making the analysis of accelerated SGD ‘backward compatible’ w.r.t. the rates of usual SGD requires to perform analysis in terms of additional problem-specific quantities, which are different from the ones coming from the assumptions A1-A4. For specific problem instances, such as online regression, there exist solutions e.g. in terms of stochastic condition number, see [R4,R2]. To the best of our knowledge, there is no recipe for such “backward compatible” bounds for the general minimization setup. We will add the underlying discussion and references to the revised version of the paper.
>
> [R1] Vaswani et al. Fast and faster convergence of SGD for over-parameterized models and an accelerated perceptron.
>
> [R2] Liu et al. Accelerating SGD with momentum for over-parameterized learning.
>
> [R3] Nagaraj et al. Least squares regression with Markovian data: Fundamental limits and algorithms.
>
> [R4] Prateek et al. Accelerating stochastic gradient descent.

---

> > ### Comment · Reviewer_9UP4 · 2023-08-13
> >
> > I'd like to thank the authors for their response! I maintain my score.

---

> > > ### Author Response · Authors · 2023-08-14
> > >
> > > Glad to hear that! Thanks again for the time spent on the review!

---

### Official Review · Reviewer_11kd · 2023-07-15

**Soundness:** 3 good
**Presentation:** 3 good
**Contribution:** 3 good
**Rating:** 7
**Confidence:** 4

**Summary:**

This paper studies the convergence of first-order methods with Markovian noise for solving stochastic optimization problems and variational inequalities. The authors establish finite-time complexity bounds which scale linearly with the mixing time of the underlying Markov chain.

**Strengths:**

This paper analyzes SGD (with and without Nesterov acceleration) with Markovian noise without the bounded domain and unifomly bounded stochastic gradient estimate assumptions, for both strongly convex and nonconvex scenarios. Matching lower bounds are also provided for the strongly convex case. Finally, an analysis for variational inequalities with general stochastic Markov oracle, arbitrary optimization set and arbitrary composite term is also provided. The results are interesting and appear to be novel.

**Weaknesses:**

Purely theoretical paper. No numerical experiments are provided.


**Questions:**

It is suggested that the authors should provide some simple numerical experiments to demonstrate the effectiveness of the proposed algorithms.


**Limitations:**

Yes.

---

> ### Author Rebuttal · Authors · 2023-08-06
>
> We thank Reviewer __11kd__ for the work and for the positive feedback! Next, we answer the issues raised.
>
> __W1__ and __Q1__(experiments):
>
> Unfortunately in a week to respond, we couldn't make the final version of the experiments, but we promise to add a set of experiments to vary the theory in the new version of the paper (the final one, if the paper is accepted). In particular, now we aim at practical verification of some theoretical dependencies. For example, Reviewer __9UP4__ asked to observe the linear dependence on the mixing time and also include numerical results for the Variational Inequalities (VI) setting. We aim to illustrate the benefits of the proposed approaches on the policy evaluation problems on the discrete type environments solved within the TD ($\lambda$)-type algorithms.

---

> > ### Comment · Reviewer_11kd · 2023-08-16
> > **Response**
> >
> > Adding experiments is just a suggestion. Also as a more theoretical researcher, I personally think that the theoretical contribution of the paper is sufficient for me to champion its acceptance. With numerical experiments, I would say the goal is to reach a broader audience and raise the impact of the paper, especially for large conferences like NeurIPS.

---

> > > ### Author Response · Authors · 2023-08-16
> > >
> > > Thanks so much for the reply! We understand your position (we absolutely agree!). Since another reviewer also asked for experiments, we decided to add them.

---

### Author Rebuttal · Authors · 2023-08-06

Reviewer __9UP4__ requested to make the analysis with stepsize, which is decreasing  and independent of a large set of unknown parameters. In part because of the size limitations of a personal reply to Reviewer, we put this in a general reply.

Here we give a sketch of how to make a decreasing step for the strongly convex case. The first detail is to make the parameters of Algorithm 1 depend on the iteration number $k$: $\gamma, p, \beta, \eta, M, B \to \gamma_k, p_k, \beta_k, \eta_k, M_k, B_k$, but the expressions for $\gamma_k, p_k, \beta_k, \eta_k, M_k, B_k$ from Theorem 1 are kept. We start the proof after line 643:
\begin{align}
    \mathbb{E_k} [||x^{k+1} - x^*||^2 &+ 2\gamma_k \eta^2_k (f(x_f^{k+1}) - f(x^*)) ]
\\\\
\leq&
    \left( 1 - \beta_k / 2\right) \left[|| x^k - x^*||^2 + 2\gamma_k \eta^2_k (f(x_f^k) - f(x^*)) \right]
    + 36 \beta^2_k \mu^{-2} \left(C_{1} \tau b^{-1} + (C_{1} + 1) \tau^2 b^{-2}\right) \sigma^2
\end{align}
With $\beta_k = \sqrt{\frac{4 p^2_k \mu \gamma_k}{3}}, ~~\eta_k = \sqrt{\frac{3}{\mu \gamma_k}}$, we get
\begin{align}
    \mathbb{E_k} [||x^{k+1} - x^*||^2 &+ \frac{6}{\mu} (f(x_f^{k+1}) - f(x^*))]
    \\\\
    \leq&
    \left( 1 - \sqrt{\frac{p_k^2 \mu \gamma_k}{3}} \right) \left[|| x^k - x^*||^2 + \frac{6}{\mu} (f(x_f^k) - f(x^*)) \right]
    + \frac{48 p_k^2 \gamma_k}{\mu} \left(C_{1} \tau b^{-1} + (C_{1} + 1) \tau^2 b^{-2}\right) \sigma^2.
\end{align}
Since $p_k = \left[1 + 2\left(  1 +  \gamma_k L \right) \left(1 + 4 \left[C_{1} \tau b^{-1} + (C_{1} + 1) \tau^2 b^{-2}\right] \delta^2 \right)\right]^{-1}$ and $\gamma_k \in (0; \frac{3}{4L})$, then $p_k \in [p_l; p_u]$, where $p_l, p_u \sim (1 + (1 + \tau b^{-1} + \tau b^{-2})\delta^2)$. Taking the full expectation, we have
\begin{align}
    \mathbb{E}[||x^{k+1} - x^*||^2 &+ \frac{6}{\mu} (f(x_f^{k+1}) - f(x^*))]
    \\\\
    \leq&
    \left( 1 - \sqrt{\frac{p_l^2 \mu \gamma_k}{3}} \right) \mathbb{E}\left[|| x^k - x^*||^2 + \frac{6}{\mu} (f(x_f^k) - f(x^*)) \right]
    + \frac{48 p_k^2 \gamma_k}{\mu} \left(C_{1} \tau b^{-1} + (C_{1} + 1) \tau^2 b^{-2}\right) \sigma^2.
\end{align}
With notation $r_{k} = \mathbb{E}\left[|| x^k - x^*||^2 + \frac{6}{\mu} (f(x^k_f) - f(x^*)) \right]$, $a = \sqrt{p^2_l \mu / 3}$, $\omega_k = \sqrt{\gamma_k}$ and $C = \frac{48 p_u^2}{\mu} \left(C_{1} \tau b^{-1} + (C_{1} + 1) \tau^2 b^{-2}\right) \sigma^2$, we obtain
\begin{align}
    r_{k+1}
    \leq
    \left( 1 - a \omega_k \right) r_{k} + \omega_k^2 C,
\end{align}
where $0 < \omega_k \leq d = \sqrt{3/(4L)}$. For this kind of recursion, we can use the results of Lemma 3 of [Stich. Unified optimal analysis of the (stochastic) gradient method]. In particular, we can choose $\gamma_k$ as follows
\begin{align*}
    \text{if } N \leq \frac{d}{a}, && \gamma_k = \frac{1}{d}
    \\\\
    \text{if } N > \frac{d}{a}, \text{ and } k < \lceil \frac{N}{2 }\rceil && \gamma_k = \frac{1}{d}
    \\\\
    \text{if } N > \frac{d}{a}, \text{ and } k \geq \lceil \frac{N}{2 }\rceil && \gamma_k = \frac{2}{a(k + \frac{2d}{a} + \lceil \frac{N}{2 }\rceil)}
\end{align*}
and get
$$
r_N = \mathcal{O}\left( \frac{d r_0}{a} \exp\left( - \frac{aN}{2d} \right) + \frac{C}{a^2 N}  \right).
$$
The step choice still depends on the horizon of iterations $N$. To fix it, we can the following restart procedure. We construct a sequence of the iteration budgets $N_t=2^t$ for $t\geq 0$. For each restart $t$ we set the stepsize $\gamma(N_t)$, run the algorithm for $N_t$ basic iterations and use the obtained point as a warm-start for the next restart. For simplicity, we can also use the same starting $x^0$ point for all the restarts. Let us now assume that the algorithm made $N$ iterations. This means that it made at least $T = \lfloor \log_2(N)\rfloor-1$ restarts. Since at the end of the last restart it made $N_T$ basic iterations with the stepsize $\gamma(N_T)$, we can guarantee that
$$
r_{N} = r_{N_T} = \mathcal{O}\left( \frac{d r_0}{a} \exp\left( - \frac{aN_T}{2d} \right) + \frac{C}{a^2 N_T}  \right).
$$
One can note that $N_T \sim N$, then
$$
r_{N} = r_{N_T} = \mathcal{O}\left( \frac{d r_0}{a} \exp\left( - \frac{aN}{2d} \right) + \frac{C}{a^2 N}  \right).
$$
This algorithm does not require to fix the number of basic steps $N$ in advance. If we want to have $\varepsilon$-solution in terms of $r_N$, then we have the following estimate on the number of iterations:
$$
N = \mathcal{O}\left( \frac{d}{a} \log \varepsilon^{-1} + \frac{C}{a^2 \varepsilon}\right) = \mathcal{O}\left( \left[1 + (1 + \tau b^{-1} + \tau^2 b^{-2} ) \delta^2\right] \sqrt{\frac{L}{\mu}} \log \varepsilon^{-1} + \frac{\sigma^2}{\mu^2 \varepsilon}\right).
$$
It remains to note that now $B_k = b \log_2 M_k = b \log_2 M_k \sim b \log_2 N \sim b \log_2 \varepsilon^{-1}$ - it gives additional log factor in the estimate for oracle complexity. This factor doesn't really affect the final bound.

---

### Decision · Program_Chairs · 2023-09-21

**Decision:**

Accept (poster)

**Comment:**

This is a solid paper on learning with Markovian noise, focusing on convergence bounds that are linear in the Markov chain's mixing time, across a number of assumption settings (strong convexity, non-convexity, and VIs). Four expert reviewers all recommend acceptance, underscoring the result, the clarity of exposition and analysis, and the coverage of related work. Basic work on learning from non-iid noise is inherently interesting today and to the NeurIPS audience, and this paper extends knowledge on that front well.

The authors' interaction with the reviewers was productive and brought about some revisions that will improve the final paper further. Among these, I'll underscore the discussions clarifying the dependencies on parameters in the convergence bounds (e.g. the range of the delta parameter and the dependence on mini-batch sizes, from the threads with reviewers 9UP4 and RVr5, and the noise parameter from the exchange with reviewer 8XQi). The list of examples from discussion with reviewer RVr5 will also be a nice addition.

Two reviewers also brought up numerical experiments -- more as a suggestion than a necessary addition. The author indicated progress on this feedback that should help strengthen the final draft further as well.